# A genetic framework for RNAi inheritance in *Caenorhabditis elegans*

Jan Schreier [ID][1], Lizaveta Pshanichnaya [ID][1,2], Fridolin Kielisch [ID][3] & René F Ketting [ID][1,4✉]

## Abstract

**Gene regulation by RNA interference (RNAi) is a conserved process driven by double-stranded RNA (dsRNA). It responds to exogenous cues and drives endogenous gene regulation. In *Caenorhabditis elegans*, RNAi can be inherited from parents to offspring. While a number of factors have been implicated in this inheritance process, we do not understand how and when they function. Using a new inheritance assay, we establish a hierarchy amongst previously identified inheritance factors. We show that the nuclear Argonaute protein HRDE-1 is required for RNAi establishment in parents and offspring, but not for the inheritance process. In contrast, the cytoplasmic Argonaute protein WAGO-3 is the only factor essential for inheritance, via sperm and oocyte, while not affecting establishment in either parent or offspring. We propose a cycle in which nuclear and cytoplasmic Argonaute proteins interact to generate both a silencing response and a cytoplasmic factor that transmits the silencing between parent and offspring, WAGO-3. Finally, we implicate the RNA helicase ZNFX-1 as a factor that allows the inherited WAGO-3 protein to trigger silencing in the offspring.**

**Keywords** RNAi; Paternal Inheritance; Maternal Inheritance; *C. elegans*; Argonaute
**Subject Category** RNA Biology

## Introduction

RNA interference (RNAi) is a process in which genes are silenced through double-stranded RNA (dsRNA) (Fire et al, 1998). After processing the dsRNA into smaller fragments by the enzyme Dicer, these so-called siRNAs are bound by a protein of the Argonaute family. Guided by siRNAs, these proteins can induce the destabilization of mRNAs by direct cleavage or through the recruitment of additional factors. Additionally, nuclear Argonaute proteins can induce transcriptional gene silencing. These processes, reviewed by (Ketting, 2011; Meister, 2013; Ozata et al, 2019; Luteijn

and Ketting, 2013), or similar processes are deeply conserved and they play a role in various aspects of gene activity control.

In *Caenorhabditis elegans*, the RNAi process is driven through a two-step process (reviewed in (Ketting and Cochella, 2021)). First, the siRNAs made from the dsRNA are bound by an Argonaute protein named RDE-1. Instead of directly silencing mRNAs with complementary sequences, RDE-1 induces the addition of pUG tails to targeted mRNA (Shukla et al, 2020), which in turn triggers the activity of the RNA-dependent RNA polymerase (RdRP) RRF-1 on the targeted mRNA. RRF-1 generates secondary siRNAs, also known as 22G RNAs, which are bound by worm-specific Argonaute proteins, or WAGOs. These then drive the silencing in an as yet poorly understood manner. This amplification step also allows for another feature: the inheritance of RNAi-driven silencing across generations (Grishok et al, 2000; Fire et al, 1998; Alcazar et al, 2008; Vastenhouw et al, 2006). In many cases, silencing up to six generations after the dsRNA-exposed generation can still be observed, and in some cases, virtually indefinitely stable silencing can be induced (Luteijn et al, 2012; Shirayama et al, 2012; Ashe et al, 2012; Shukla et al, 2021; Lev et al, 2017; Devanapally et al, 2021). Normally, RNAi inheritance is restricted in duration, and specific mechanisms may be in place to ensure that inheritance is finite (Houri-Ze'evi et al, 2016).

The combination of environmental effects on gene expression through dsRNA and the inheritance of this silencing over generations has led to studies probing whether environmental cues, in the form of gene-regulatory information, may be transmitted to offspring. Potentially, this may better prepare animals to face specific environmental conditions that were already met by their parents. Indeed, links between nutrients (Rechavi et al, 2014), pathogens (Rechavi et al, 2011; Sengupta et al, 2024; Kaletsky et al, 2020) and animal behavior (Posner et al, 2019; Toker et al, 2022) have been coupled to RNAi inheritance, although in some cases there is dispute on reproducibility (Gainey et al, 2024; Kaletsky et al, 2025). The spontaneous creation of so-called epi-alleles through RNAi-related mechanisms has also been described, albeit that these were not found to be long-lived (Wilson et al, 2023). In order to firmly test the possibilities and limitations of RNAi inheritance, a strong framework of the responsible molecular mechanisms is required. Unfortunately, at the molecular level the RNAi inheritance process is not well understood. A number of factors have been implicated in the process, including for instance

[1]Biology of Non-coding RNA Group, Institute of Molecular Biology, Ackermannweg 4, 55128 Mainz, Germany. [2]International PhD Programme on Gene Regulation, Epigenetics & Genome Stability, Mainz, Germany. [3]Bioinformatics Core Facility, Institute of Molecular Biology, Ackermannweg 4, 55128 Mainz, Germany. [4]Institute of Developmental Biology and Neurobiology, Johannes Gutenberg University, 55099 Mainz, Germany. ✉E-mail: r.ketting@imb-mainz.de

RNA helicases (Dai et al, 2022; Wan et al, 2018; Ishidate et al, 2018) and potential scaffold proteins of germ granules (Wan et al, 2021; Placentino et al, 2021; Schreier et al, 2022; Ouyang et al, 2019; Lev et al, 2019). Also, Argonaute proteins have been implicated in the RNAi inheritance process. First, the nuclear Argonaute protein HRDE-1 drives heterochromatin formation and was one of the first RNAi inheritance factors identified (Buckley et al, 2012). In addition, the cytoplasmic Argonautes WAGO-3 and WAGO-4 have been linked to inheritance (Schreier et al, 2022; Wan et al, 2018; Conine et al, 2013; Liu et al, 2023). However, none of these factors have been woven into a coherent framework of activities that explain what is happening at the molecular level. An important aspect to note is that these studies made use of self-fertilizing hermaphrodites to study inheritance. While this is a valid approach, it does not allow conclusions on whether a given factor acts in the parents or in the offspring or both. As this is an important first step towards dissecting inheritance mechanisms, we developed an assay in which inheritance through male and female germlines can be separately studied, by using crosses between males and hermaphrodites. We then used this assay to study factors that have been implicated in RNAi inheritance: HRDE-1, WAGO-1, WAGO-3, WAGO-4, ZNFX-1, PEI-1 and PEI-2. Our results lead us to propose a novel framework for RNAi inheritance in *C. elegans*.

## Results

### Exogenous RNAi effects are heritable via both oocyte and sperm

Although RNAi through the feeding of dsRNA-expressing bacteria has been shown to work robustly for *C. elegans* hermaphrodites, its effectiveness is thought to be very low in males (Bezler et al, 2019). Nevertheless, to further dissect male, female and zygotic components of the RNAi inheritance pathway, we aimed to develop a feeding-based RNAi protocol for males and hermaphrodites. We generated a strain that specifically expressed a *gfp::histone-H4* transgene as an RNAi target throughout male and female germline development, and in mature gametes. The transgene contained a *his-67* promoter, driving a *gfp* coding sequence containing three introns fused to *his-67* followed by the 3'UTR from the *tbb-2* gene. This strain also contained the *him-5(e1490)* mutation to increase the frequency of males in the culture and expressed an endogenously tagged PGL-1::mTagRFP-T fusion protein. The latter served for automated germline detection during image processing and normalization of GFP::H4 fluorescence. We treated animals for one generation (from hatched L1 until adulthood) with either *control* RNAi or *gfp* RNAi via feeding, and calculated the relative fluorescence intensity (RFI; see Methods) of GFP::H4 in adult hermaphrodites and adult males using microscopy (Fig. 1A). Even though RNAi in males has been described to be inefficient (Ma et al, 2014), we found that males were as sensitive to the *gfp* RNAi treatment as hermaphrodites, given that no GFP::H4 signal could be detected with either fluorescence-based or immuno-based approaches (Fig. 1B–F). In addition to protein levels, we also performed smFISH experiments to assess the abundance and distribution of *gfp* mRNA in males and hermaphrodites treated with either *control* RNAi or *gfp* RNAi (Fig. 1E,F). In *control* RNAi-treated hermaphrodites, we detected diffuse cytoplasmic *gfp* mRNA

signal throughout the entire germline, with an enrichment in the distal gonad preceding weak perinuclear foci in the pachytene region. Upon *gfp* RNAi treatment, cytoplasmic *gfp* mRNA signal became undetectable, albeit some weak perinuclear signal in the pachytene region remained (Fig. 1E,F). In *control* RNAi-treated males, *gfp* mRNA was also cytoplasmic, but reduced to a few weak perinuclear foci in spermatocytes and absent in spermatids. No *gfp* mRNA signal was detected in *gfp* RNAi-treated males (Fig. 1F).

Next, we set up crosses between differently treated animals to test and quantify the individual contribution of oocytes and sperm to *gfp* RNAi inheritance. In order to identify progeny produced by allogamy, only males expressed both GFP::H4 and PGL-1::mTagRFP-T (hereafter referred to as P0 males), while hermaphrodites only expressed GFP::H4 and were wild-type for *him-5* and *pgl-1* (hereafter referred to as P0 hermaphrodites). Thus, only F1 animals that were produced by mating expressed PGL-1::mTagRFP-T. Using these two strains, we set up crosses in which *gfp* RNAi was inherited to the F1 through sperm, oocyte or both (Fig. 1G). We found that exogenous RNAi effects were faithfully inherited via both oocyte and sperm (Fig. 1H–L). Furthermore, our data shows that *gfp* RNAi inheritance via the oocyte was slightly more effective than via sperm, but less effective compared to inheritance via both gametes. These findings were independent on the sex of the progeny, as they were observed for both F1 hermaphrodites and F1 males (Fig. 1H–J). We note that the *gfp* RNAi effects triggered in the P0 generation can also be transgenerationally inherited by following autogamy of P0 hermaphrodites, just as described in previous studies (Fig. EV1A–C) (Buckley et al, 2012). However, we could not detect silencing following two consecutive paternal inheritance events (Fig. EV1D,E).

### WAGO-4 and HRDE-1 affect germline *gfp* RNAi establishment in hermaphrodites

Having established this exogenous RNAi inheritance assay that allows us to individually investigate maternal or paternal contributions, we sought to examine the requirement of proteins that were previously shown to act in RNAi inheritance: WAGO-1, WAGO-3, WAGO-4, HRDE-1 and ZNFX-1 (Conine et al, 2010; Schreier et al, 2022; Wan et al, 2018; Buckley et al, 2012; Liu et al, 2023). Throughout the study, we used deletion alleles that are most likely null alleles (see Reagents and Tools Table). First, we investigated their role in maternal *gfp* RNAi inheritance and started by assessing the effect of each mutant on germline *gfp* RNAi sensitivity in adult hermaphrodites (Fig. 2A–K). Along with their respective wild-type controls, we quantified the fluorescence intensity of GFP::H4 in mutant animals that were treated with either *control* RNAi or *gfp* RNAi. We found that *wago-1*, *wago-3* and *znfx-1* mutants did not show any defects in germline GFP::H4 silencing (Fig. 2B–D,G–I). In contrast, germline-wide RNAi defects in *wago-4* mutant hermaphrodites became occasionally apparent after a few mutant generations and became fully penetrant after 11 generations (Generation 8: Fig. 2E,J, Generations 4, 8, and 11: Fig. EV2A–F). *hrde-1* mutants also showed impaired RNAi sensitivity, as consistently both GFP::H4 protein and *gfp* mRNA remained detectable in the distal region of the gonad (Figs. 2F,K and EV3A–C). In the more proximal zone of the gonad, including the oocytes, silencing in *hrde-1* mutants seemed unperturbed. This

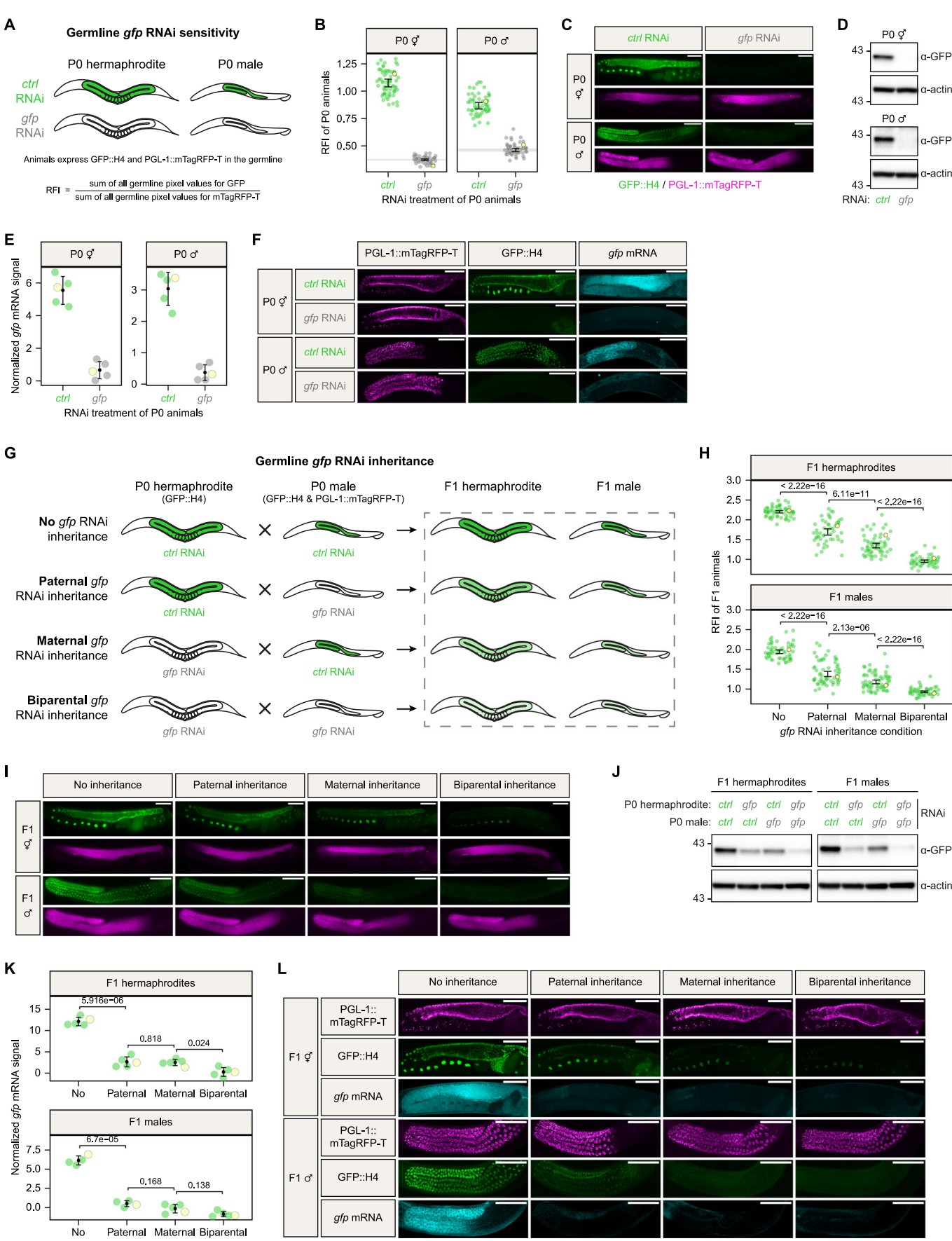

**Figure 1. Exogenous RNAi effects are heritable via both oocyte and sperm.**

(A) Schematic representation summarizing the *gfp* RNAi effect on P0 hermaphrodites and P0 males expressing GFP::H4 and PGL-1::mTagRFP-T in the germline. Germline color illustrates GFP::H4 expression: green—expressed, white—silenced. (B) Relative fluorescence intensity (RFI) of GFP::H4 in P0 hermaphrodites and P0 males treated with either *control* RNAi (green) or *gfp* RNAi (gray). Each dot represents an individual animal, with yellow dots referring to representative micrographs shown in (C). 95% confidence intervals of the median are shown as black error bars for all samples, and as gray lines for *gfp* RNAi treatments. Sample size = ~60 animals per condition. (C) Widefield fluorescence micrographs of representative P0 hermaphrodites and P0 males treated with either *control* RNAi or *gfp* RNAi, as indicated in (B). GFP::H4 and PGL-1::mTagRFP-T appear in green and magenta, respectively. Scale bars: 50 µm. (D) Representative Western blots comparing levels of GFP::H4 with respect to β-Actin loading control in P0 hermaphrodites and P0 males treated with either *control* RNAi or *gfp* RNAi. Sample size = 50 animals per condition. (E) Normalized smFISH signal of *gfp* mRNA in P0 hermaphrodites and P0 males treated with either *control* RNAi (green) or *gfp* RNAi (gray). Each colored dot represents an individual animal, with yellow dots referring to representative micrographs shown in (F). The black dot and error bars represent the mean and standard deviation, respectively. Sample size = 5 animals per condition. (F) Confocal maximum intensity projections of representative P0 hermaphrodites and P0 males treated with either *control* RNAi or *gfp* RNAi, as indicated in (E). GFP::H4, PGL-1::mTagRFP-T and *gfp* mRNA appear in green, magenta and cyan, respectively. Scale bars: 50 µm. (G) Crossing schemes summarizing the effect of *gfp* RNAi inheritance via male and/or female gametes on germline GFP::H4 expression in F1 animals. Germline color illustrates GFP::H4 expression: green—expressed, shades of light green—expressed at lower levels, white—silenced. (H) Relative fluorescence intensity (RFI) of GFP::H4 in F1 hermaphrodites and F1 males after *gfp* RNAi inheritance via male and/or female gametes. Each dot represents an individual animal, with yellow dots referring to representative micrographs shown in (I). 95% confidence intervals of the mean are shown as black error bars. The *P* values are from *t* tests for linear contrasts in Gaussian models and have been corrected for multiple testing per sex. Sample size = ~60 F1 animals from 5 P0 founders per condition. (I) Widefield fluorescence micrographs of representative F1 hermaphrodites and F1 males after *gfp* RNAi inheritance via male and/or female gametes, as indicated in (H). GFP::H4 and PGL-1::mTagRFP-T appear in green and magenta, respectively. Scale bars: 50 µm. (J) Representative Western blots comparing levels of GFP::H4 with respect to β-Actin loading control in F1 hermaphrodites and F1 males after *gfp* RNAi inheritance via male and/or female gametes. Sample size = 50 F1 animals from 5 P0 founders per condition. (K) Normalized smFISH signal of *gfp* mRNA in F1 hermaphrodites and F1 males after *gfp* RNAi inheritance via male and/or female gametes. Each colored dot represents an individual animal, with yellow dots referring to representative micrographs shown in (L). The black dot and error bars represent the mean and standard deviation, respectively. *P* values were calculated using an unpaired two-tailed Student's *t* test. Sample size = 5 F1 animals from 5 P0 founders per condition. (L) Confocal maximum intensity projections of representative F1 hermaphrodites and F1 males after *gfp* RNAi inheritance via male and/or female gametes, as indicated in (K). GFP::H4, PGL-1::mTagRFP-T and *gfp* mRNA appear in green, magenta, and cyan, respectively. Scale bars: 50 µm. Source data are available online for this figure.

matches earlier observations that reported functional RNAi in the oocytes of *hrde-1* mutants (Buckley et al, 2012). We hypothesize that this proximal silencing activity is post-transcriptional. These data highlight functions for WAGO-4 and HRDE-1 in the establishment of germline RNAi. While WAGO-4 is seemingly required to maintain germline RNAi sensitivity over generations, HRDE-1 is required for exogenous RNAi-mediated gene silencing in the distal germline. We also note that the basal expression of the transgene appeared to be affected by specific mutations. This aspect will be covered in more detail later.

## Maternal WAGO-3 and WAGO-4, but not HRDE-1 and ZNFX-1 are required for RNAi inheritance via the oocyte

As all investigated mutants succeeded to silence GFP::H4 during oogenesis, we continued by examining their effect on maternal *gfp* RNAi inheritance (Fig. 3A). To do this, we crossed wild-type and mutant P0 hermaphrodites treated with either *control* RNAi or *gfp* RNAi with P0 males that were wild-type for the respective genes and treated with *control* RNAi, taking care that RNAi-triggering bacteria were transferred as little as possible (see Methods). Both males and hermaphrodites were homozygous for the GFP::H4 transgene. F1 animals of all four crosses were grown until adulthood, and GFP::H4 fluorescence was quantified using microscopy. We then calculated the relative GFP::H4 fluorescence reductions and compared them between F1 animals sired by wild-type and mutant P0 hermaphrodites. In Fig. 3B,C, we present a statistical summary of the data, as described in more detail in the Methods section. The data behind these plots is presented in the other panels (Fig. 3D-M). Surprisingly, we found that maternal loss of ZNFX-1 caused the maternally inherited *gfp* RNAi effect to be significantly stronger, as we measured a greater GFP::H4 fluorescence reduction in both F1 hermaphrodites and F1 males (Fig. 3B,D,I). Also contrary to our expectations, we found that *hrde-1* mutant P0 hermaphrodites still passed on *gfp* RNAi, to both F1 hermaphrodites and males (Fig. 3B,F,K),

and also *wago-1* mutants behaved like wild-type (Fig. 3B,E,J). Finally, F1 animals sired by *wago-3* or *wago-4* mutant hermaphrodites revealed clear RNAi inheritance defects, independent of the filial sex (P0 *wago-4* generation 8; Fig. 3B,C,G,H,L,M). Given the observation that *wago-4* mutant hermaphrodites become resistant to germline RNAi over generations (Fig. EV2A–F), we also tested whether the duration of *wago-4* mutant homozygosity affected maternal *gfp* RNAi inheritance. To achieve this, we compared P0 hermaphrodites carrying the *wago-4* mutation for either four or eight generations for their ability to inherit *gfp* RNAi effects. We found that the duration of *wago-4* mutant homozygosity did not mostly affect RNAi inheritance to the F1 (Fig. EV2G–J). Inheritance to F1 males was marginally better after four generations than after eight, but this effect may be driven by the increased GFP::H4 expression in the control strain, particularly after four generations (Fig. EV2G–J). Independent of the duration of mutant homozygosity and filial sex, we noticed that the maternal inheritance defects were not complete and some *gfp* RNAi effects were still transferred (Fig. EV2K). We conclude that maternal inheritance of *gfp* RNAi requires WAGO-3 and WAGO-4, and is independent of HRDE-1. Furthermore, our data show that maternal ZNFX-1 restricts the maternal inheritance of RNAi.

## WAGO-4 and HRDE-1 are required for germline *gfp* RNAi in males

We proceeded by investigating the paternal contribution of exogenous RNAi inheritance, for which we also included the PEI granule-localizing proteins PEI-1 and PEI-2 (Schreier et al, 2022) in our analyses. Firstly, we assessed the effect of each mutant on germline *gfp* RNAi sensitivity in adult males (Fig. 4A–O). Along with their respective wild-type controls, we quantified the fluorescence intensity of GFP::H4 in mutant males that were treated with either *control* RNAi or *gfp* RNAi. We found that *wago-1*, *wago-3*, *znfx-1*, *pei-1*, and *pei-2* mutants did not show any defects

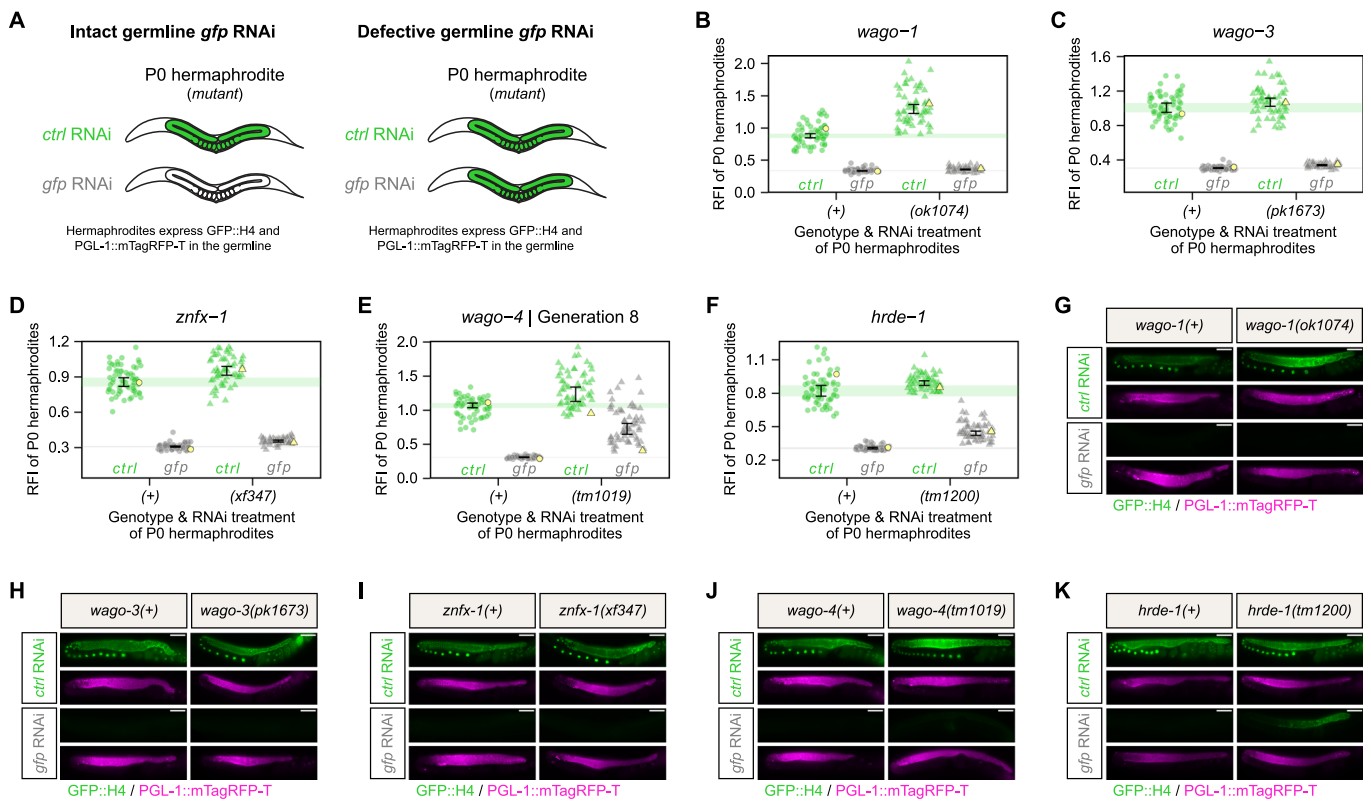

**Figure 2. WAGO-4 and HRDE-1 are required for germline RNAi establishment in hermaphrodites.**

(**A**) Schematic representations summarizing the *gfp* RNAi effect on mutant P0 hermaphrodites expressing GFP::H4 and PGL-1::mTagRFP-T in the germline. Germline color illustrates GFP::H4 expression: green—expressed, white—silenced. (**B–F**) Relative fluorescence intensity (RFI) of GFP::H4 in P0 hermaphrodites treated with either *control* RNAi (green) or *gfp* RNAi (gray). P0 hermaphrodites were either wild-type (circle) or mutant (triangle) for *wago-1* (**B**), *wago-3* (**C**), *znfx-1* (**D**), *wago-4* (**E**), or *hrde-1* (**F**). Each dot represents an individual animal, with yellow dots referring to representative micrographs shown in (**G–K**). 95% confidence intervals of the median are shown as black error bars for all samples as well as green (*control* RNAi) and gray (*gfp* RNAi) lines for wild-type conditions. Sample size = ~60 animals per condition. (**G–K**) Widefield fluorescence micrographs of representative P0 hermaphrodites treated with either *control* RNAi or *gfp* RNAi, as indicated in (**B–F**). P0 hermaphrodites were either wild-type or mutant for *wago-1* (**G**), *wago-3* (**H**), *znfx-1* (**I**), *wago-4* (**J**), or *hrde-1* (**K**). GFP::H4 and PGL-1::mTagRFP-T appear in green and magenta, respectively. Scale bars: 50 µm. Note that the plots and images of (**E, J**) are also displayed in Fig. EV2B,D,E, for ease of comparison within Fig. EV2. Source data are available online for this figure.

in germline GFP::H4 silencing in males (Fig. 4B–F,I–M). In contrast, *wago-4* and *hrde-1* mutant males showed profound germline RNAi defects, as GFP::H4 fluorescence was still detectable and close to wild-type levels throughout the male germline after *gfp* RNAi treatment (P0 *wago-4* generation 8; Fig. 4G,H,N,O). The *wago-4* RNAi defect did not display the transgenerational effect as we noted in hermaphrodites. We conclude that WAGO-4 and HRDE-1 are crucial for the establishment of germline RNAi in males.

## Paternal WAGO-3 and PEI-1 are required for RNAi inheritance via sperm

Next, we investigated the effect of each mutant on paternal *gfp* RNAi inheritance (Fig. 5A). We again set up four crosses per investigated gene, where either wild-type or mutant P0 males treated with either *control* RNAi or *gfp* RNAi were crossed with *control* RNAi-treated P0 hermaphrodites that were wild-type for respective genes. GFP::H4 fluorescence of adult F1 animals was quantified using microscopy, and used to calculate the relative

GFP::H4 fluorescence reduction after paternal RNAi inheritance (Fig. 5B–Q). Consistent with the inability to establish germline RNAi (Fig. 4G,H), we found that *wago-4* and *hrde-1* mutant males sired offspring that failed to repress GFP::H4 expression after paternal *gfp* RNAi treatment (P0 *wago-4* generation 8; Fig. 5B–E,K,L). We also found that *wago-1*, *znfx-1* and *pei-2* mutant P0 males still inherited *gfp* RNAi effects as efficiently as their respective wild-type controls (Fig. 5B,F–H,M–O). In contrast, F1 animals sired by *wago-3* or *pei-1* mutant P0 males revealed clear defects in paternal *gfp* RNAi inheritance. Their relative GFP::H4 fluorescence reduction was significantly lower compared to wild-type controls, with *wago-3* mutant P0 males eliciting a stronger defect than *pei-1* mutant P0 males (Fig. 5B,I,J,P,Q). We conclude that paternal inheritance of *gfp* RNAi requires WAGO-3 and PEI-1.

## HRDE-1 and ZNFX-1 are required zygotically to re-establish gene silencing

Our finding that *hrde-1* and *znfx-1* mutant hermaphrodites were able to faithfully inherit *gfp* RNAi contrasts with previous studies

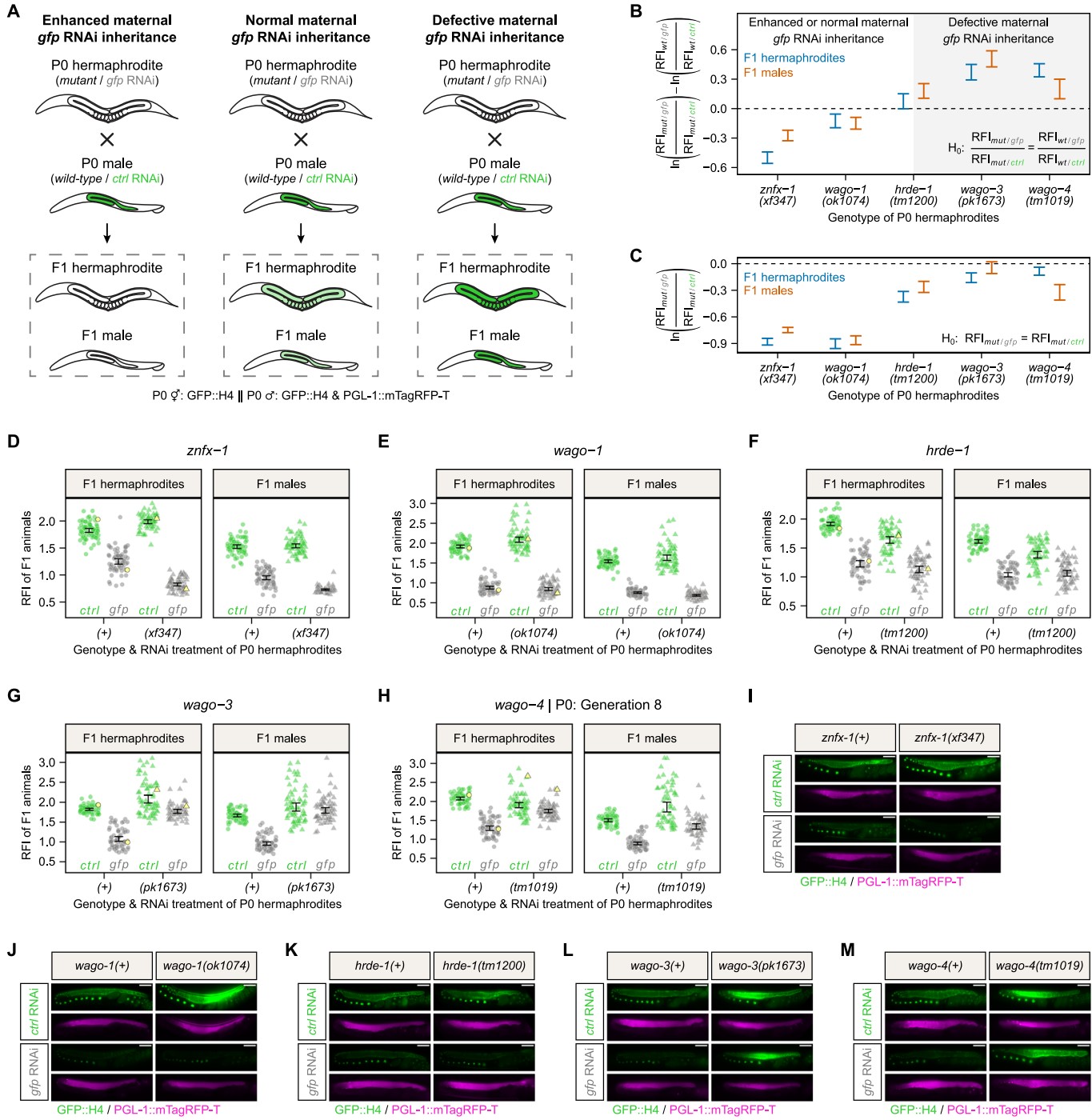

that reported profound RNAi inheritance defects for these mutants (Buckley et al, 2012; Wan et al, 2018). These findings, however, were ascribed by a *gfp* RNAi inheritance assay that scored GFP expression during clonal propagation. As both RNAi-treated animals and non-treated offspring were mutants, defects in either the parental or the filial generation could have caused the defective phenotype. We thus asked whether the strong RNAi inheritance defect described for *hrde-1* and *znfx-1* mutants was based on defective re-establishment of RNAi in F1 animals rather than defective transmission of RNAi effects from the oocyte to the

embryo. For clarity, we define transmission as the transfer of parentally derived factors that mediate *gfp* silencing to the offspring, e.g., small RNA-loaded Argonaute proteins. While these factors are required to initialize the re-establishment of *gfp* silencing in F1 animals, other parentally expressed factors are likely to act upstream of this, and zygotically expressed factors are likely to act downstream in this process without themselves being directly involved in the transmission process itself. To test this experimentally, we conducted maternal RNAi inheritance assays, in which we crossed mutant P0 hermaphrodites with either

Figure 3. Maternal WAGO-3 and WAGO-4 are required for RNAi inheritance via the oocyte.

(A) Crossing schemes summarizing how mutations in P0 hermaphrodites affect germline GFP::H4 expression in F1 animals after maternal *gfp* RNAi inheritance. Germline color illustrates GFP::H4 expression: green—expressed, light green—expressed at lower level, white—silenced. (B) Comparison of the relative GFP::H4 fluorescence reduction after maternal *gfp* RNAi inheritance between F1 animals sired by wild-type and mutant P0 hermaphrodites for the indicated genes. The plot summarizes the Gaussian models fitted in (D–H) and depicts 95% confidence intervals (CIs) of mean differences of log fold changes. The mean differences in log fold changes are in the center of the CIs. The null hypothesis (H$_0$) expresses equality of relative GFP::H4 fluorescence reduction between wild-type and mutant condition, meaning that the mutation does not cause an enhanced or defective maternal *gfp* RNAi inheritance. A 95% CI not including zero is equivalent to a rejection of the null hypothesis at the 5% significance level and indicates that mutations caused either enhanced (95% CI < 0) or defective (95% CI > 0) maternal *gfp* RNAi inheritance. Color of CIs indicates sex of F1 animals: blue—hermaphrodite, red—male. Sample size = ~60 F1 animals from 5 P0 founders per condition. (C) Comparison of the relative GFP::H4 fluorescence intensity after maternal RNAi inheritance between F1 animals sired by mutant P0 hermaphrodites treated with either *control* RNAi or *gfp* RNAi. The plot summarizes the Gaussian models fitted in (D–H) and depicts 95% confidence intervals (CIs) of mean RFI log fold changes. The mean log fold changes are in the center of the CIs. The null hypothesis (H$_0$) expresses equality of relative GFP::H4 fluorescence intensity between *control* RNAi and *gfp* RNAi treatments, meaning that the mutation causes a completely defective maternal *gfp* RNAi inheritance. A 95% CI not including zero is equivalent to a rejection of the null hypothesis at the 5% significance level and indicates that mutations do not cause a completely defective maternal *gfp* RNAi inheritance. Color of CIs indicates sex of F1 animals: blue—hermaphrodite, red—male. Sample size = ~60 F1 animals from 5 P0 founders per condition. (D–H) Relative fluorescence intensity (RFI) of GFP::H4 in F1 hermaphrodites and F1 males after maternal RNAi inheritance. P0 hermaphrodites were treated with either *control* RNAi (green) or *gfp* RNAi (gray), and were either wild-type (circle) or mutant (triangle) for *znfx-1* (D), *wago-1* (E), *hrde-1* (F), *wago-3* (G), or *wago-4* (H). P0 males were always wild-type for these genes and treated with *control* RNAi. Each dot represents an individual animal, with yellow dots referring to representative micrographs shown in (I–M). 95% confidence intervals of the mean are shown as black error bars. Sample size = ~60 F1 animals from 5 P0 founders per condition. (I–M) Widefield fluorescence micrographs of representative F1 hermaphrodites after maternal RNAi inheritance, as indicated in (D–H). Indicated genotypes and RNAi treatments refer to P0 hermaphrodites, which were either wild-type or mutant for *znfx-1* (I), *wago-1* (J), *hrde-1* (K), *wago-3* (L), or *wago-4* (M). GFP::H4 and PGL-1::mTagRFP-T appear in green and magenta, respectively. Scale bars: 50 μm. Source data are available online for this figure.

homozygous wild-type or mutant P0 males (Fig. 6A). GFP::H4 fluorescence was quantified in adult F1 animals and compared between heterozygous and homozygous mutant offspring. We found that only F1 animals whose parents were both homozygous mutant for *hrde-1* or *znfx-1* showed RNAi inheritance defects (Fig. 6B–D). Notably, the GFP::H4 signals in these animals were comparable with offspring sired by *control* RNAi-treated P0 hermaphrodites, revealing a complete lack of inheritance. Intrigued by these findings, we performed a second RNAi inheritance experiment, in which we aimed to exclude possible effects arising from mutant P0 animals. To achieve this, we set up crosses with heterozygous P0 animals that were treated with *gfp* RNAi. We manually divided the sired F1 animals based on visible GFP expression in the germline (Set 1: GFP negative, Set 2: GFP positive), and used subsets to quantify GFP::H4 fluorescence and occurrence of homozygous mutant F1 animals, respectively (Fig. 6E). We found that the group of GFP positive offspring showed a high percentage of homozygous mutants (*hrde-1(tm1200)*: 100%, *znfx-1(xf347)*: 83.8%), while GFP negative F1 animals were mostly wild-type or heterozygous for respective genes (*hrde-1(tm1200)*: 1.75%, *znfx-1(xf347)*: 4.09%) (Fig. 6F,G). We conclude that HRDE-1 and ZNFX-1 are crucial for the re-establishment of gene silencing in the embryo, following the transmission of RNAi effects from the parents.

## Spatiotemporal regulation of basal GFP::H4 expression in the hermaphroditic germline

We noticed that *wago-1* and *wago-4* mutant P0 hermaphrodites displayed an increased GFP::H4 fluorescence signal in the early/mid pachytene region of the adult germline (Fig. 2G,J). In order to describe this unexpected GFP::H4 dysregulation in more detail, we compared mean fluorescence intensities (MFIs) and coefficients of variations (CVs) of GFP::H4 and PGL-1::mTagRFP-T between wild-type and mutant P0 hermaphrodites treated with *control* RNAi (Fig. EV4A–C). While the MFIs describe overall changes in fluorescence intensity, CVs (ratio of the standard deviation to the mean) describe the dispersion of fluorescence intensities and hence

help to determine local alterations. We mainly found minor MFI and CV changes for both GFP::H4 and PGL-1::mTagRFP-T in *wago-3*, *hrde-1* and *znfx-1* mutants. *wago-1* and *wago-4* mutants, however, showed a tendency for elevated MFIs of GFP:H4, but not for PGL-1::mTagRFP-T (Fig. EV4B). Both mutants also showed a strong and specific increase in CVs for GFP::H4 (Fig. EV4C), indicating that they indeed expressed GFP::H4 with greater variation compared to their wild-type controls. Our data indicate that the increased fluorescence intensity in the early/mid pachytene region was specific to GFP::H4 in *wago-1* and *wago-4* mutants. Notably, we also found that this altered GFP::H4 pattern is maternally inherited to heterozygous hermaphroditic offspring (Fig. 3J,M), but not male offspring (Fig. EV4D,E). Interestingly, similar increased expression was detected in offspring sired by *wago-3* mutant hermaphrodites and wild-type males, even though this was not detected in the parental generation (Figs. 2H, 3L and EV4D,E). Finally, we also asked whether these increases in transgene expression were specific to the maternal germline and performed the same analyses with P0 males (Fig. EV5A–E). This only revealed slight increases in MFIs and CVs for GFP::H4 in *wago-1* mutant males (Fig. EV5B,C). We conclude that *wago-1* and *wago-4* affect the basal expression of the GFP::H4 transgene, indicating that it is under RNAi-related control also in the absence of dsRNA targeting GFP. These effects can be inherited, which is affected by *wago-3*.

## Discussion

We developed an exogenous RNAi inheritance assay that enables the specific and unbiased investigation of the maternal, paternal and filial contribution to RNAi inheritance via the combination of mating experiments and fluorescence quantification of a transgenic GFP::H4 reporter. The assay also allows the simultaneous examination of germline RNAi sensitivity across the various regions of the male and hermaphroditic germlines, including mature gametes. We used this assay to further characterize established RNAi mutants, and successfully dissected formerly

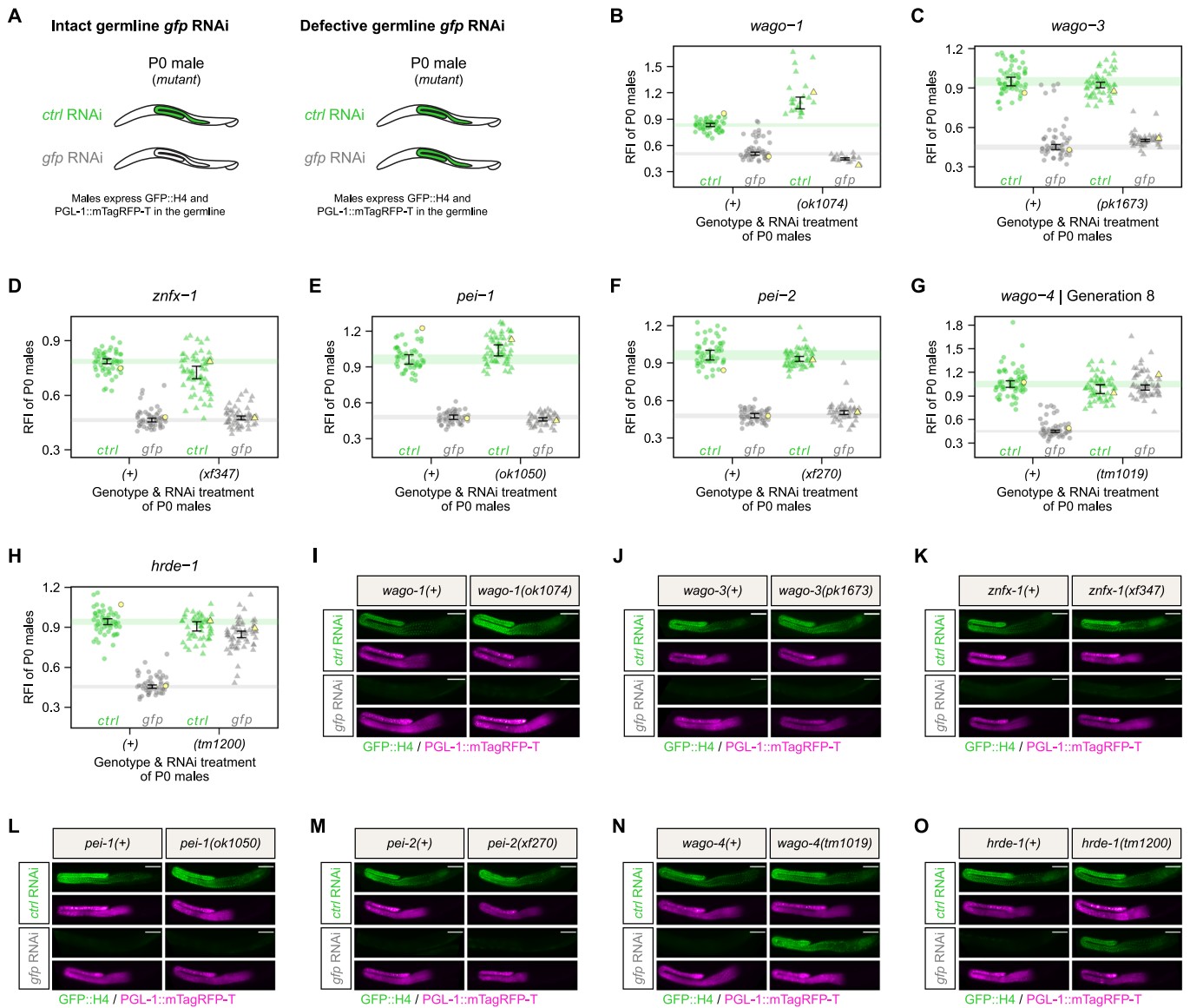

**Figure 4. WAGO-4 and HRDE-1 are required for germline RNAi establishment in males.**

(A) Schematic representations summarizing the *gfp* RNAi effect on mutant P0 males expressing GFP::H4 and PGL-1::mTagRFP-T in the germline. Germline color illustrates GFP::H4 expression: green—expressed, white— silenced. (B–H) Relative fluorescence intensity (RFI) for GFP::H4 in P0 males treated with either *control* RNAi (green) or *gfp* RNAi (gray). P0 males were either wild-type (circle) or mutant (triangle) for *wago-1* (B), *wago-3* (C), *znfx-1* (D), *pei-1* (E), *pei-2* (F), *wago-4* (G), or *hrde-1* (H). Each dot represents an individual animal, with yellow dots referring to representative micrographs shown in (I–O). 95% confidence intervals of the median are shown as black error bars for all samples as well as green (*control* RNAi) and gray (*gfp* RNAi) lines for wild-type conditions. Sample size = ~60 animals per condition (exception: ~23 *wago-1(ok1074)* animals). (I–O) Widefield fluorescence micrographs of representative P0 males treated with either *control* RNAi or *gfp* RNAi, as indicated in (B–H). P0 males were either wild-type or mutant for *wago-1* (I), *wago-3* (J), *znfx-1* (K), *pei-1* (L), *pei-2* (M), *wago-4* (N), or *hrde-1* (O). GFP::H4 and PGL-1::mTagRFP-T appear in green and magenta, respectively. Scale bars: 50 μm. Source data are available online for this figure.

reported inheritance defects by attributing specific roles during: (i) different germ cell stages in the adult, (ii) sperm-mediated transmission of RNAi effects, (iii) oocyte-mediated transmission of RNAi effects, and (iv) re-establishment of silencing in offspring of RNAi-treated animals. A model consistent with these results is depicted in Fig. 7A,B. Starting with the male inheritance data we obtained, we propose a loop between HRDE-1 and WAGO-3, where HRDE-1 mediates silencing and drives the loading of WAGO-3, and WAGO-3 mediates the transmission of the silencing

signal (22G RNAs) from sperm to offspring. Being in PEI granules, WAGO-3 perfectly fits this role. However, WAGO-3 loading in response to HRDE-1 has not been experimentally demonstrated. Yet, the following observations support that idea. First, HRDE-1 has been shown to drive the formation of new 22G RNAs, named tertiary 22G RNAs (Sapetschnig et al, 2015). Second, direct loading of WAGO-3 in response to dsRNA, independent of HRDE-1, would not explain the defect of *hrde-1* mutants in transmission of the silencing. Next, in the F1 ZNFX-1 and HRDE-1 are both needed

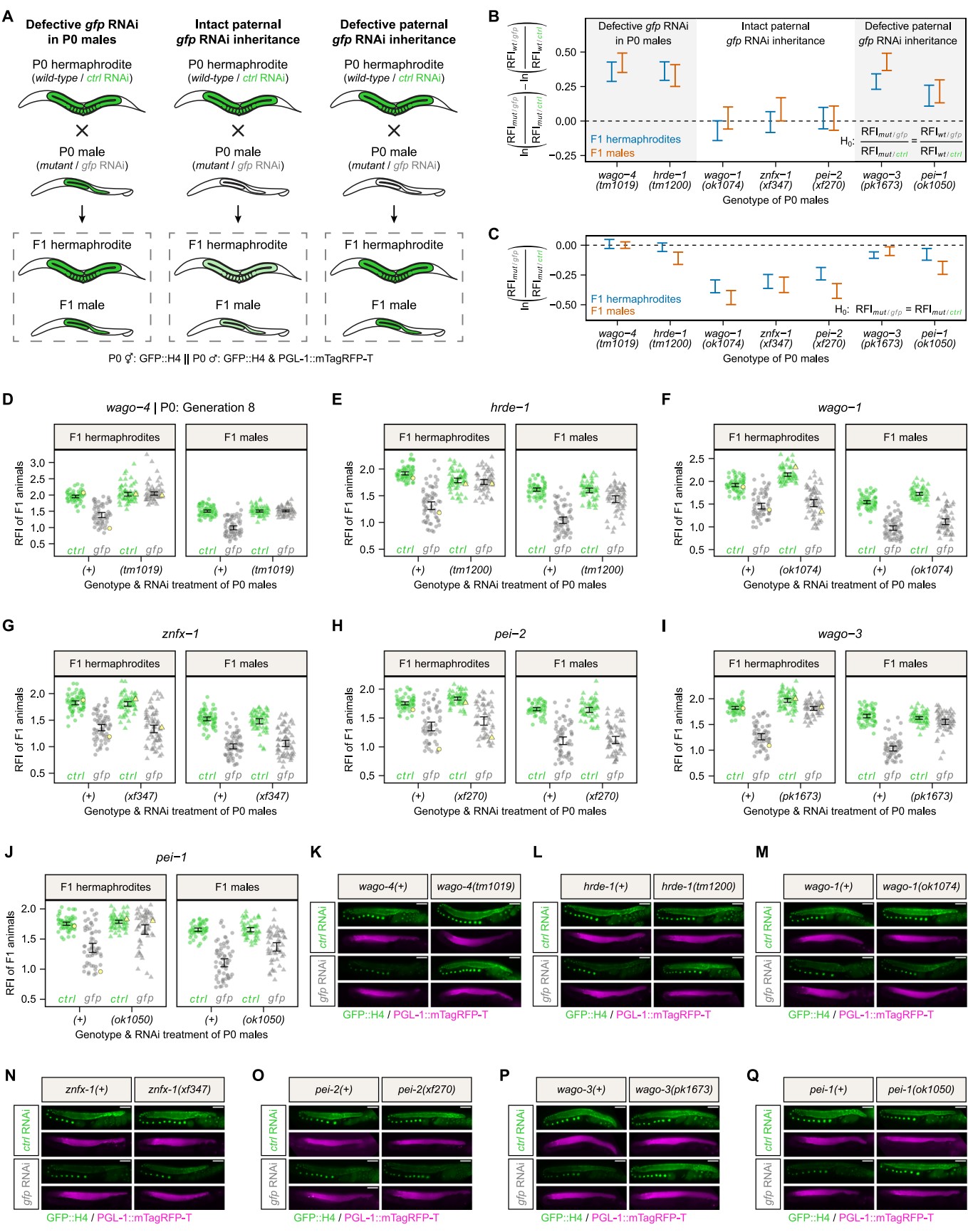

**Figure 5. Paternal WAGO-3 and PEI-1 are required for RNAi inheritance via sperm.**

(A) Crossing schemes summarizing how mutations in P0 males affect germline GFP::H4 expression in F1 animals after paternal *gfp* RNAi inheritance. Germline color illustrates GFP::H4 expression: green—expressed, light green—expressed at lower level, white—silenced. (B) Comparison of the relative GFP::H4 fluorescence reduction after paternal *gfp* RNAi inheritance between F1 animals sired by wild-type and mutant P0 males for the indicated genes. The plot summarizes the Gaussian models fitted in (D–J) and depicts 95% confidence intervals (CIs) of differences of mean log fold changes. The mean differences in log fold changes are in the center of the CIs. The null hypothesis (H₀) expresses equality of relative GFP::H4 fluorescence reduction between wild-type and mutant condition, meaning that the mutation does not cause a defective paternal *gfp* RNAi inheritance. A 95% CI not including zero, is equivalent to a rejection of the null hypothesis at the 5% significance level and indicates that mutations caused defective (95% CI > 0) paternal *gfp* RNAi inheritance. We note that *wago-4* and *hrde-1* mutant P0 males did not inherit the *gfp* RNAi effect due to defective *gfp* RNAi sensitivity (see Fig. 4). Color of CIs indicates sex of F1 animals: blue—hermaphrodite, red—male. Sample size = ~60 F1 animals from 5 P0 founders per condition. (C) Comparison of the relative GFP::H4 fluorescence intensity after paternal RNAi inheritance between F1 animals sired by mutant P0 hermaphrodites treated with either *control* RNAi or *gfp* RNAi. The plot summarizes the Gaussian models fitted in (D–J) and depicts 95% confidence intervals (CIs) of mean RFI log fold changes. The mean log fold changes are in the center of the CIs. The null hypothesis (H₀) expresses equality of relative GFP::H4 fluorescence intensity between *control* RNAi and *gfp* RNAi treatments, meaning that the mutation causes a completely defective paternal *gfp* RNAi inheritance. A 95% CI not including zero is equivalent to the rejection of the null hypothesis at the 5% significance level and indicates that mutations did not cause a completely defective paternal *gfp* RNAi inheritance. Color of CIs indicates sex of F1 animals: blue—hermaphrodite, red—male. Sample size = ~60 F1 animals from 5 P0 founders per condition. (D–J) Relative fluorescence intensity (RFI) of GFP::H4 in F1 hermaphrodites and F1 males after paternal RNAi inheritance. P0 males were treated with either *control* RNAi (green) or *gfp* RNAi (gray), and were either wild-type (circle) or mutant (triangle) for *wago-4* (D), *hrde-1* (E), *wago-1* (F), *znfx-1* (G), *pei-2* (H), *wago-3* (I), or *pei-1* (J). P0 hermaphrodites were always wild-type for these genes and treated with *control* RNAi. Each dot represents an individual animal, with yellow dots referring to representative micrographs shown in (K–Q). 95% confidence intervals of the mean are shown as black error bars. Sample size = ~60 F1 animals from 5 P0 founders per condition. (K–Q) Widefield fluorescence micrographs of representative F1 hermaphrodites after paternal RNAi inheritance, as indicated in (D–J). Indicated genotypes and RNAi treatments refer to P0 males, which were either wild-type or mutant for *wago-4* (K), *hrde-1* (L), *wago-1* (M), *znfx-1* (N), *pei-2* (O), *wago-3* (P), or *pei-1* (Q). GFP::H4, and PGL-1::mTagRFP-T appear in green and magenta, respectively. Scale bars: 50 μm. Source data are available online for this figure.

to re-establish the silencing. As ZNFX-1 has been coupled to 22G RNA biogenesis intermediates (Ouyang et al, 2022), we propose that ZNFX-1 may help WAGO-3 to trigger 22G RNA biogenesis that will be loaded into HRDE-1, and possibly other Argonaute proteins. It is unclear at which stage of germ cell development this step would occur. In our system, the silencing, however, mostly depends on HRDE-1, but this could differ for different targets. Effectively, this establishes an intergenerational loop between HRDE-1 and WAGO-3, where HRDE-1 establishes silencing in parent and offspring, while WAGO-3 provides the physical transmission of 22G RNAs from parent to offspring. We anticipate that this loop also acts during inheritance via the mother. However, oocytes contain an HRDE-1 independent RNAi response, acting at the post-transcriptional level. As inheritance via the oocyte also required WAGO-3, our model poses that this post-transcriptional silencing mechanism should also trigger the loading of WAGO-3. Which WAGO protein, besides WAGO-3, is involved here is not clear at present, but WAGO-4 is an interesting candidate (Wan et al, 2018; Xu et al, 2018). Notably, ZNFX-1 could play a role here as well (Ishidate et al, 2018; Ouyang et al, 2022; Wan et al, 2018), by allowing cytoplasmic WAGO protein to trigger tertiary 22G RNA biogenesis for loading into WAGO-3. After fertilization, oocyte-derived WAGO-3 could act the same as sperm-derived WAGO-3. The role of ZNFX-1 in these steps can be conceptualized as follows: ZNFX-1 may stimulate the production of tertiary 22G RNAs by cytoplasmic WAGO proteins. Uncontrolled re-triggering of 22G RNA biogenesis would result in a runaway amplification response, and thus needs to be controlled. We propose ZNFX-1 plays a role in this. Finally, how WAGO-4 fits into this framework is unclear. Given the transgenerational build-up of a phenotype in the hermaphrodite, it is possible that any RNAi-inheritance phenotype of *wago-4* mutants is rather indirect; for instance, genes involved in the mechanisms may drop in expression, or substrates of the RNAi mechanisms may slowly accumulate and compete for limited resources. For this reason, we leave WAGO-4 out of our model at present. Below, we will discuss our results on the establishment,

transmission and re-establishment of silencing in more detail in light of this model.

## Basal expression of the GFP::H4 transgene

Expression levels of the transgene in absence of RNAi were affected by some of the mutants we tested. In particular WAGO-1 appears to play a major role in this, in both the male and female germline. Strikingly, *wago-1* mutants did not display any defects in RNAi establishment or its inheritance in our assays, suggesting that WAGO-1 may be dedicated to endogenous silencing cues, and may dampen the expression of our transgene. While *wago-4* also affected this aspect, its effects were weaker, and it is feasible that these stem from indirect effects, as discussed above. Interestingly, we also observed an effect on basal transgene expression in heterozygous offspring from *wago-3* mutants, but not in *wago-3* mutants themselves. How may this be explained? We propose that this effect stems from the introduction of parental small RNAs into an RNAi system that was lacking such parental contributions, as WAGO-3 seems to be the main conveyor of small RNAs between generations. The absence of such parental small RNAs may lead to a re-equilibration of the overall RNAi network and that may be de-stabilized by re-introduction of paternal or maternal small RNAs. Interestingly, WAGO-1 and WAGO-3 have been shown to interact in vivo (Schreier et al, 2022; Barucci et al, 2020), possibly reflecting "communication" between these two Argonaute proteins. We also note that these effects on basal expression imply that the transgene is under limited RNAi influence also in non-RNAi conditions. This may be related to the weak perinuclear foci for *gfp* mRNA we detected in absence of RNAi. This can of course affect the overall behavior of the transgene upon dsRNA administration. We would like to emphasize here that any individual gene may respond slightly different to RNAi, depending on how strongly endogenous RNAi pathways affect it. We would argue that such differences reflect relevant aspects of the RNAi mechanism, and one should not be valued above another. For instance, results from RNAi on

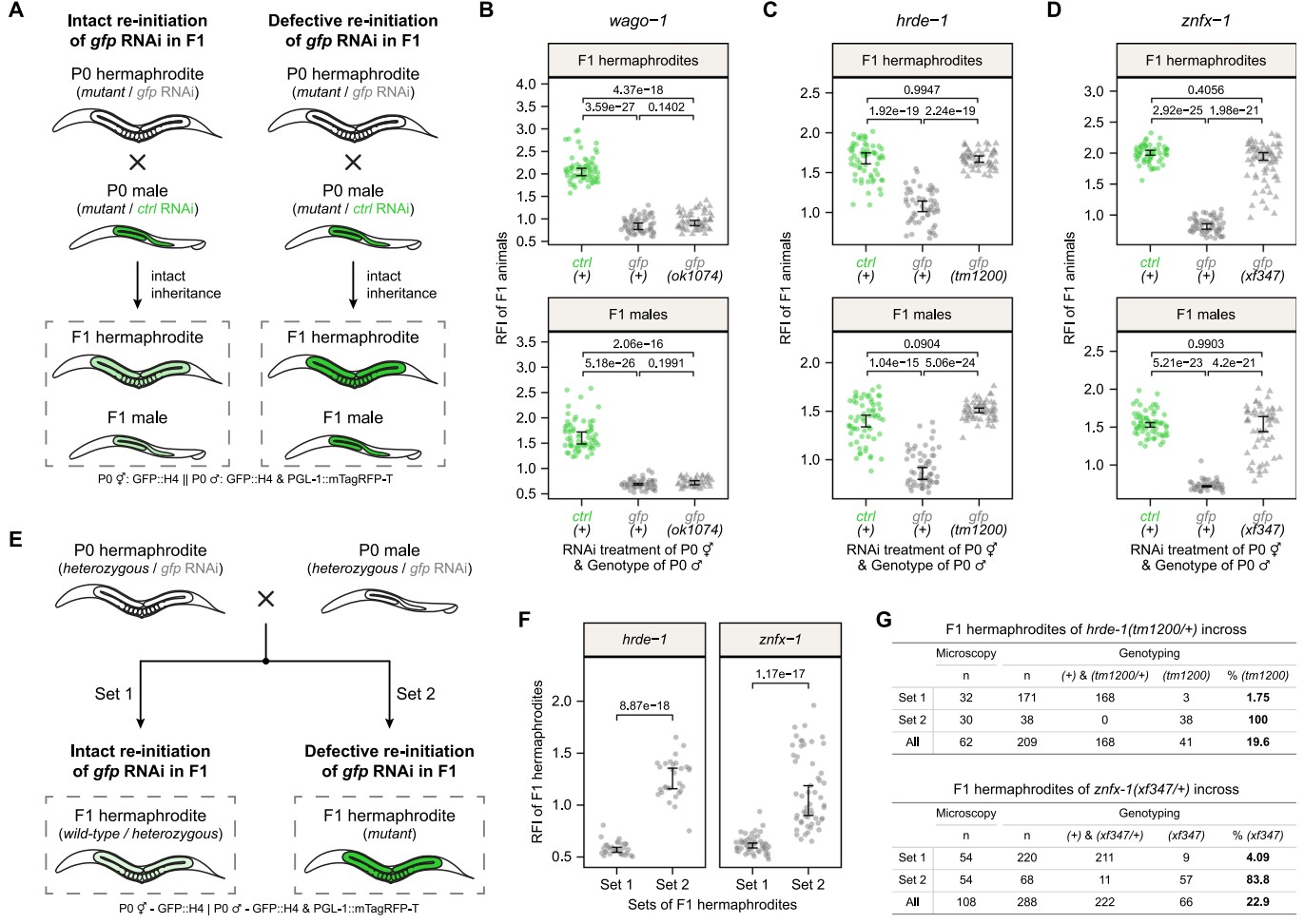

**Figure 6. Zygotic HRDE-1 and ZNFX-1 are required for re-initiation of RNAi.**

(A) Crossing schemes summarizing how mutations in F1 animals affect their germline GFP::H4 expression after maternal *gfp* RNAi inheritance. Germline color illustrates GFP::H4 expression: green—expressed, light green—expressed at a lower level, white—silenced. (B–D) Relative fluorescence intensity (RFI) of GFP::H4 in F1 hermaphrodites and F1 males after maternal RNAi inheritance. P0 hermaphrodites were treated with either *control* RNAi (green) or *gfp* RNAi (gray), and P0 males were either wild-type (circle) or mutant (triangle) for *wago-1* (B), *hrde-1* (C) or *znfx-1* (D). For all crosses, P0 hermaphrodites were mutant for the indicated genes, and P0 males were treated with *control* RNAi. Each dot represents an individual animal. 95% confidence intervals of the median are shown as black error bars. Statistically significant differences were determined using two-sided Dunn's tests. Sample size = ~60 F1 animals from five P0 founders per condition. (E) Crossing scheme depicting how the genotype of F1 animals affects their germline GFP::H4 expression after biparental *gfp* RNAi inheritance. Germline color illustrates GFP::H4 expression: green—expressed, light green—expressed at a lower level, white—silenced. (F) Relative fluorescence intensity (RFI) of GFP::H4 in F1 hermaphrodites after biparental *gfp* RNAi inheritance. P0 hermaphrodites and P0 males were treated with *gfp* RNAi and heterozygous for a mutation in either *hrde-1* or *znfx-1*. F1 hermaphrodites were divided in two sets based on visible GFP expression in the germline. Each dot represents an individual animal. 95% confidence intervals of the median are shown as black error bars. Statistically significant differences were determined using two-sided Wilcoxon rank-sum tests. Sample size = ~31 (*hrde-1*) or 54 (*znfx-1*) F1 animals from five P0 founders per condition. (G) Genotyping summary and percentage of homozygous mutant F1 animals in each set, as described in (E, F). Source data are available online for this figure.

---

transgenes may reflect better on RNAi's well-documented role on fighting foreign genetic material, while results on endogenous genes (which may have evolved to resist RNAi) may reflect more on the use of RNAi for controlling gene expression.

## Differences between male and female germline

We detected roles for HRDE-1 and WAGO-4 in the establishment of RNAi in the adult germline. HRDE-1 is required for RNAi establishment in the distal region of the hermaphroditic germline, as also observed before (Ouyang et al, 2022). This implies that in the mitotic and early meiotic stages nuclear RNAi is the main

driver of the silencing effect. During later meiotic stages, nuclear RNAi is apparently no longer required, as we observed silencing in these stages in *hrde-1* mutants. This is consistent with previous findings showing that RNAi in oocytes is functional in *hrde-1* mutants (Buckley et al, 2012). Possibly, this reflects a shift to overall more post-transcriptional gene regulation during the final steps of oogenesis. However, we cannot rule out that histone protein turnover in the female gonad may be responsible for this loss of GFP signal in the proximal region. The involvement of WAGO-4 in RNAi establishment in the ovary only became apparent after a number of homozygous mutants generations (>4). Notably, RNAi establishment in these later-generation *wago-4*

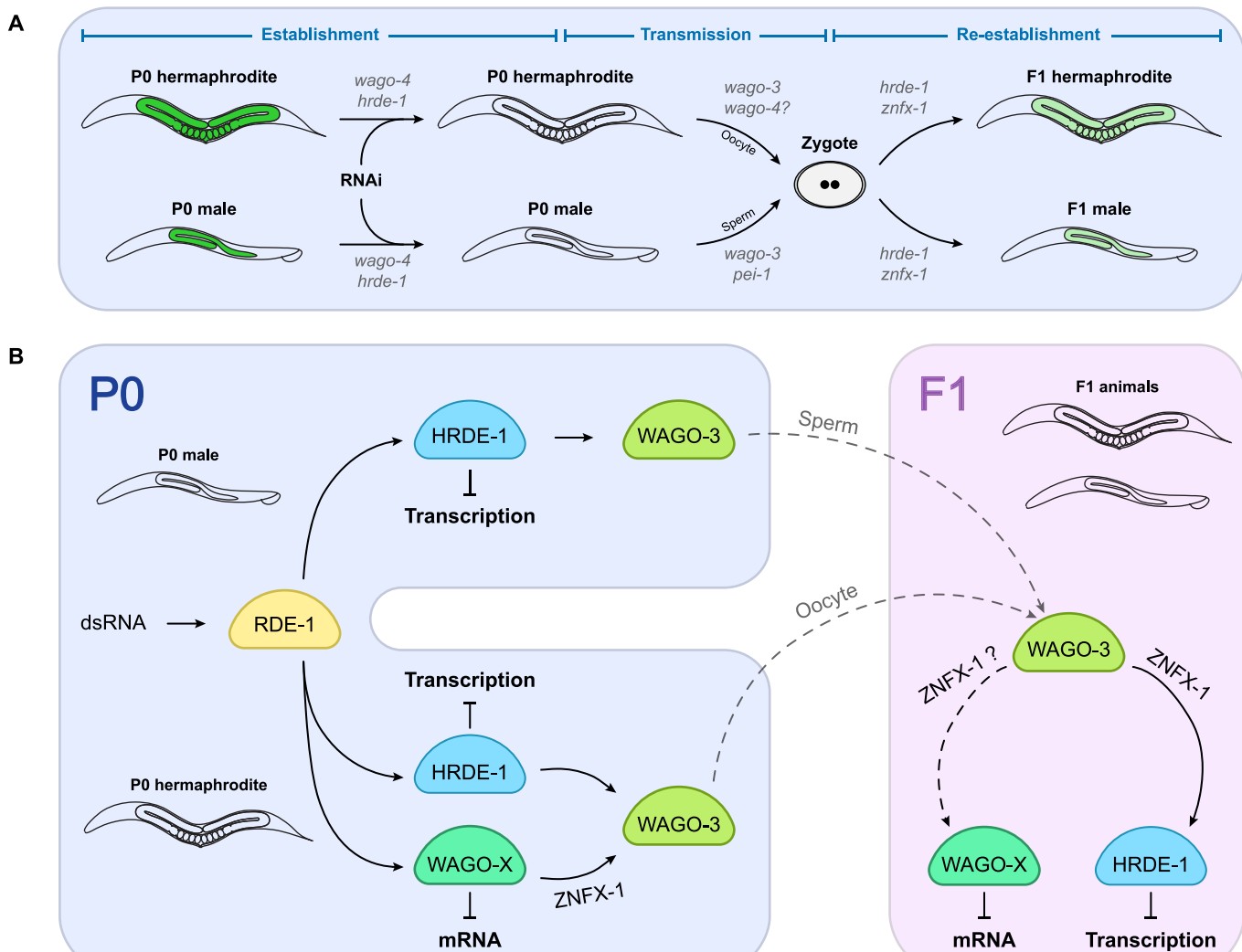

**Figure 7. Working model of the three major steps governing exogenous RNAi inheritance.**

(A) Schematic representation summarizing the three major steps during exogenous RNAi inheritance and their required factors. (1) Establishment: Upon exposure to double-stranded RNA, WAGO-4 and HRDE-1 are individually required to mediate gene silencing of the germline-expressed target transcript in both P0 hermaphrodites and P0 males. Interestingly, their impact differs between both sexes. While lack of each protein causes global and immediate germline RNAi sensitivity defects in P0 males, their impact in P0 hermaphrodites is less comprehensive. Here, HRDE-1 is seemingly required for gene silencing in mitotic and early meiotic germ cells, while germline-wide defects in *wago-4* mutants become only apparent after a couple of mutant generations. (2) Transmission: following the establishment of silencing in the germline, WAGO-3 is required for faithful transmission of RNAi effects to the embryo. Paternal contribution seems to be mainly driven by PEI granules, as only *wago-3* and *pei-1* mutants showed defects in paternal RNAi transmission. The role of WAGO-4 is less clear, as over generations, also an RNAi-establishment phenotype developed in the female germline. Nevertheless, in early generations, *wago-4* mutant females established RNAi normally, while they did not transmit silencing, suggesting WAGO-4 does play a role in transmission. (3) Re-establishment: Upon faithful transmission of RNAi effects from the gamete(s) to the embryo, HRDE-1 and ZNFX-1 are individually required for the re-establishment of RNAi in both hermaphroditic and male offspring. (B) Working model for *gfp* RNAi inheritance. We propose a cycle between the nuclear Argonaute protein HRDE-1 and the cytoplasmic Argonaute protein WAGO-3. While HRDE-1 drives transcriptional gene silencing and WAGO loading, WAGO-3 ensures faithful RNAi inheritance. After fertilization, the RNA helicase ZNFX-1 might function to link inherited WAGO-3 protein to HRDE-1-dependent nuclear RNAi activity in the offspring. Possibly, this can also re-establish a post-transcriptional silencing response.

mutants is defective all over the germline, also in the HRDE-1 dependent zone. Given the delay in phenotype establishment, the interaction may be rather indirect, for instance, via a destabilization of the overall RNAi network, leading to an overall loss of RNAi effectivity across the whole germline. The endogenous 22G RNAs bound by WAGO-4 are different from those of HRDE-1, further supporting this potential indirect effect of WAGO-4 on HRDE-1 functionality.

In the male germline, both HRDE-1 and WAGO-4 affected GFP expression following *gfp* RNAi across the entire gonad. Notably, in *wago-4* mutant males we did not observe the intermediate sensitivity we detected initially in the hermaphrodites, but simply full RNAi resistance. We hypothesize that the male germline may be more sensitive to the effects of WAGO-4 on the RNAi network compared to the female germline. Finally, the male germline did not have an HRDE-1 independent zone, suggesting the absence of post-transcriptional

silencing in response to dsRNA. However, slow histone turnover during spermatogenesis could also contribute to this observation.

## Sperm-mediated transmission of RNAi

WAGO-3 was implicated in paternal transmission of RNAi to the F1 generation, while it was not required for establishment. Consistent with our previous work on PEI granules that secure WAGO-3 in sperm (Schreier et al, 2022), PEI-1 was also found to be required. However, the effect of *pei-1* was weaker. Likely, this reflects the previously described loss of much, but not all of the WAGO-3 protein into the residual body during spermatogenesis (Schreier et al, 2022). We could not detect a role for the second structural PEI granule component: PEI-2. This may be explained by the fact that in *pei-2* mutants PEI granules are evenly distributed across spermatids and residual bodies, leading to even more WAGO-3 remaining in the sperm, and concentrated in PEI granules, compared to *pei-1* mutants (Schreier et al, 2022).

## Oocyte-mediated transmission of RNAi

Inheritance via the oocyte also depends on WAGO-3. Indeed, WAGO-3 is also expressed in the female germline (Schreier et al, 2022; Seroussi et al, 2023). However, in another assay, named *Mutator*-induced sterility (Mis-assay) (de Albuquerque et al, 2015), we could not detect a role for maternal WAGO-3 (Schreier et al, 2022). The main difference between the RNAi assay described here and the Mis-assay is that the latter is driven by endogenous small RNAs, while the RNAi assay is driven by exogenous RNAi. Also, the Mis-assay readout is a rather convoluted effect, much downstream of the original RNAi-related effects that are being inherited (de Albuquerque et al, 2015; Phillips et al, 2015). It is therefore feasible that, in the Mis-assay, loss of WAGO-3 may be compensated by other effects. These may be other Argonaute proteins or other gene-regulatory events. As the RNAi assay described here is more direct, it is likely that WAGO-3 is indeed involved in 22G RNA inheritance, both via sperm and via oocytes.

Maternal HRDE-1 was not required for inheritance. Our model proposes that this is due to the existence of a post-transcriptional mechanism in the oocytes. One major hurdle to better understand this is that we do not know which WAGO protein(s) is(are) required for driving RNAi during oogenesis. None of the tested WAGO proteins on their own affected silencing in the oocytes specifically, suggesting that this is driven by a combination of the tested WAGOs and/or by one that we did not test.

The effect of maternal ZNFX-1 on inheritance is also noteworthy: maternal ZNFX-1 limited inheritance via the oocyte (Fig. 3). Interestingly, an enhancing effect of ZNFX-1 was noted before by Ouyang et al (Ouyang et al, 2022), who noted earlier and stronger effects on *mex-6* transcripts in the nucleus upon *mex-6* RNAi in *znfx-1* mutant P0 animals. In the context of our model, loss of ZNFX-1 function could keep the initial RNAi response focused on HRDE-1, without diverging the response into other branches of the RNAi pathway. Such an increased nuclear RNAi effect may be associated with an enhanced response at the chromatin level, which has been proposed (Rieger et al, 2023) to be responsible for previously described ZNFX-1-independent silencing in the F1 (Ouyang et al, 2022).

## Comparison to previous work

We identified two factors that are essential for the transmission of the silenced information to the F1 generation: WAGO-3 and WAGO-4. These are both cytoplasmic. As such, our results are in line with the described nucleus-independent inheritance of RNAi (Rieger et al, 2023). On the other hand, our model, at first glance, may not match the results obtained by Ouyang et al (Ouyang et al, 2022). In this study, the individual loss of HRDE-1 or ZNFX-1 did not result in a complete loss of silencing in the F1 generation, and fully de-silenced F1 animals were only obtained with *hrde-1;znfx-1* double mutations. In contrast, we observe close to full de-silencing in *hrde-1* and *znfx-1* single mutants, even when both parent and offspring are mutant, as they are in the assays by Ouyang et al (Ouyang et al, 2022). Several explanations for this potential inconsistency can be considered. First, we note that the exposure of P0 animals differs significantly between our studies: in our system, the P0 is exposed from L1 to adulthood, while Ouyang et al (Ouyang et al, 2022) expose only for a few hours. Another important difference in experimental setup is that we dissect parental from filial effects using crosses, while the Ouyang et al's study (Ouyang et al, 2022) does not: P0 and F1 genotypes were the same. This prevents any assignment of F1 effects to either P0 or F1. Third, in our assays, we have not addressed mRNA levels in the F1. Even if GFP protein and *gfp* mRNA levels correlated well in the P0 generation, we cannot currently comment on effects on mRNA in the F1. Nevertheless, the model we propose (Fig. 7) does not contradict the model proposed by Ouyang et al (Ouyang et al, 2022). First, our model (Fig. 7) contains a completely HRDE-1-independent, but ZNFX-1-dependent inheritance cycle, assuming that in the embryo, a post-transcriptional response can be mounted, similar to what occurs in oocytes. Loss of HRDE-1 in the P0 may in fact enhance this mode of inheritance, as more transcripts may become available for this post-transcriptional process in the oocytes of *hrde-1* mutants. Second, we state that ZNFX-1 stimulates tertiary 22G RNA biogenesis by cytoplasmic WAGOs, but not that it is essential for it. Thus, some level of ZNFX-1-independent inheritance is compatible with our model. Interestingly, Ouyang et al (Ouyang et al, 2022) observed that in *znfx-1* mutants nuclear puncta accumulated more rapidly than in wild-type animals, suggesting that loss of ZNFX-1 enhances HRDE-1-mediated silencing dynamics. Intriguingly, we observed enhanced silencing in the F1 of *znfx-1* mutant mothers (Fig. 3), possibly a consequence of the effect described by Ouyang et al (Ouyang et al, 2022). Also, these observations are consistent with the ZNFX-1-independent silencing in the F1, as proposed by (Rieger et al, 2023). The precise molecular steps driven by ZNFX-1 remain presently unclear, but Ouyang et al,'s findings of ZNFX-1 association with pUG RNA (Shukla et al, 2020; Ouyang et al, 2022) is fully consistent with our model.

## Outlook

Genetic experiments to dissect the RNAi inheritance process have been, and still are, immensely valuable. However, these assays may be convoluted by aspects that are hard to control, such as redundancy and compensatory mechanisms. It will be essential to also take on biochemical approaches to dissect RNAi and its inheritance.

Importantly, these need to go beyond the identification of interactions. For instance, does ZNFX-1 stimulate RdRP activity? Is ZNFX-1 brought to RdRP substrates by WAGO proteins? How do transcripts from HRDE-1 targeted loci feed into the RNAi pathway? These and many more questions will also need biochemical approaches to be resolved.

# Methods

### Reagents and tools table

| Reagent/resource | Reference or source | Identifier or catalog number |
|---|---|---|
| **Experimental models** | | |
| *C. elegans* wild isolate var Bristol | *Caenorhabditis* Genetics Center | N2 |
| *wago-1(ok1074) I.* deletion; presumed null | *Caenorhabditis* Genetics Center | RB1096 |
| *wago-3(pk1673) I.* deletion; presumed null | *Caenorhabditis* Genetics Center | NL5117 |
| *wago-4(tm1019) II.* deletion; presumed null | Barstead et al, 2012 | RFK22 |
| *hrde-1(tm1200) III.* deletion; presumed null | *Caenorhabditis* Genetics Center | YY538 |
| *pei-1(ok1050) IV.* deletion; presumed null | *Caenorhabditis* Genetics Center | RB1083 |
| *pei-2(xf270) V.* deletion; presumed null | Schreier et al, 2022 | RFK1342 |
| *pgl-1(xf233[pgl-1::mTagRfp-t]) IV.* | Schreier et al, 2022 | RFK1086 |
| *ttTi5605 II; unc-119(ed3) III; oxEx1578[eft-3p::gfp + Cbr-unc-119(+)].* | Frøkjær-Jensen et al, 2012 | EG6699 |
| *xfSi255[his-67p::gfp::his-67::tbb-2 3'UTR + Cbr-unc-119(+)] II.* | This study | RFK1305 |
| *xfSi255[his-67p::gfp::his-67::tbb-2 3'UTR + Cbr-unc-119(+)] II; him-5(e1490) V.* | This study | RFK1622 |
| *xfSi255[his-67p::gfp::his-67::tbb-2 3'UTR + Cbr-unc-119(+)] II; pgl-1(xf233[pgl-1::mTagRfp-t]) IV.* | This study | RFK1527 |
| *xfSi255[his-67p::gfp::his-67::tbb-2 3'UTR + Cbr-unc-119(+)] II; wago-1(ok1074) I.* | This study | RFK1513 |
| *xfSi255[his-67p::gfp::his-67::tbb-2 3'UTR + Cbr-unc-119(+)] II; wago-3(pk1673) I.* | This study | RFK1520 |
| *xfSi255[his-67p::gfp::his-67::tbb-2 3'UTR + Cbr-unc-119(+)] II; wago-4(tm1019) II.* | This study | RFK1515 |
| *xfSi255[his-67p::gfp::his-67::tbb-2 3'UTR + Cbr-unc-119(+)] II; hrde-1(tm1200) III.* | This study | RFK1494 |
| *xfSi255[his-67p::gfp::his-67::tbb-2 3'UTR + Cbr-unc-119(+)] II; znfx-1(xf347) II.* | This study | RFK1615 |
| *xfSi255[his-67p::gfp::his-67::tbb-2 3'UTR + Cbr-unc-119(+)] II; pgl-1(xf233[pgl-1::mTagRfp-t]) IV; him-5(e1490) V.* | This study | RFK1405 |
| *xfSi255[his-67p::gfp::his-67::tbb-2 3'UTR + Cbr-unc-119(+)] II; pgl-1(xf233[pgl-1::mTagRfp-t]) IV; him-5(e1490) V; wago-1(ok1074) I.* | This study | RFK1512 |
| *xfSi255[his-67p::gfp::his-67::tbb-2 3'UTR + Cbr-unc-119(+)] II; pgl-1(xf233[pgl-1::mTagRfp-t]) IV; him-5(e1490) V; wago-3(pk1673) I.* | This study | RFK1470 |
| *xfSi255[his-67p::gfp::his-67::tbb-2 3'UTR + Cbr-unc-119(+)] II; pgl-1(xf233[pgl-1::mTagRfp-t]) IV; him-5(e1490) V; wago-4(tm1019) II.* | This study | RFK1514 |
| *xfSi255[his-67p::gfp::his-67::tbb-2 3'UTR + Cbr-unc-119(+)] II; pgl-1(xf233[pgl-1::mTagRfp-t]) IV; him-5(e1490) V; hrde-1(tm1200) III.* | This study | RFK1473 |
| *xfSi255[his-67p::gfp::his-67::tbb-2 3'UTR + Cbr-unc-119(+)] II; pgl-1(xf233[pgl-1::mTagRfp-t]) IV; him-5(e1490) V; znfx-1(xf347) II.* | This study | RFK1619 |
| *xfSi255[his-67p::gfp::his-67::tbb-2 3'UTR + Cbr-unc-119(+)] II; pgl-1(xf233[pgl-1::mTagRfp-t]) IV; him-5(e1490) V; pei-1(ok1050) IV.* | This study | RFK1471 |
| *xfSi255[his-67p::gfp::his-67::tbb-2 3'UTR + Cbr-unc-119(+)] II; pgl-1(xf233[pgl-1::mTagRfp-t]) IV; him-5(e1490) V; pei-2(xf270) V.* | This study | RFK1472 |
| **Recombinant DNA** | | |
| Plasmid—Induces Mos1 transposition by injection. | Addgene | pCFJ601 - #34874 |
| Plasmid—Negative selection, heat-shock inducible. | Addgene | pMA122 - #34873 |
| Plasmid—Visual extra-chromosomal array marker. Red, nervous system, cytosolic. | Addgene | pGH8 - #19359 |
| Plasmid—Visual extra-chromosomal array marker. Red, body wall muscle, cytosolic. | Addgene | pCFJ104 - #19328 |

| Reagent/resource | Reference or source | Identifier or catalog number |
|---|---|---|
| Plasmid—Visual extra-chromosomal array marker. Red, pharynx, cytosolic. | Addgene | pCFJ90 - #19327 |
| Plasmid—Multiple cloning site vector for MosSCI on locus ttTi5605 | Addgene | pCFJ350 - #34866 |
| Plasmid—gfp::SEC::3xflag parental vector for CRISPR/Cas9 insertions | Addgene | pDD282 - #66823 |
| Plasmid—[his-67p::gfp::his-67::tbb-2 3'UTR] insertion in pCFJ350 | This study | pRFK4197 |
| Plasmid—Expression of Cas9 and sgRNA(F + E) without protospacer | Schreier et al, 2022 | pRFK2411 |
| Plasmid—Expression of sgRNA(F + E) without protospacer | Schreier et al, 2022 | pRFK2412 |
| Plasmid—Expression of Cas9 and unc-58 sgRNA1(F + E) | Schreier et al, 2022 | pRFK2588 |
| Plasmid—Expression of znfx-1 sgRNA1(F + E) | This study | pRFK3358 |
| Plasmid—Expression of znfx-2 sgRNA1(F + E) | This study | pRFK3359 |
| Plasmid—Expression of znfx-3 sgRNA1(F + E) | This study | pRFK3360 |
| Plasmid—Expression of znfx-4 sgRNA1(F + E) | This study | pRFK3361 |
| Plasmid—Empty backbone for RNAi feeding experiments | Addgene | L4440 - #1654 |
| Plasmid—gfp sequence from pDD282 inserted in L4440 | This study | pRK4103 |
| **Antibodies** | | |
| Monoclonal mouse anti-GFP (B-2) | Santa Cruz Biotechnology | #sc-9996 |
| Monoclonal rabbit β-Actin (D6A8) | Cell Signaling Technology | #8457 |
| Anti-mouse IgG, HRP-linked antibody | Cell Signaling Technology | #7076 |
| Anti-rabbit IgG, HRP-linked antibody | Cell Signaling Technology | #7074 |
| **Oligonucleotides and other sequence-based reagents** | | |
| Protospacer sequence—ATC CAC GCA CAT GGT CAC TA | Arribere et al, 2014 | unc-58(e665) #1 |
| Protospacer sequence—CCA CCT CAA CGT GAG GTT GG | This study | znfx-1(xf347) #1 |
| Protospacer sequence—ATC CAT CGC GAT GCC CAT GG | This study | znfx-1(xf347) #2 |
| Protospacer sequence—TAA TCT CAC CAA TGC TTC GG | This study | znfx-1(xf347) #3 |

| Reagent/resource | Reference or source | Identifier or catalog number |
|---|---|---|
| Protospacer sequence—GAC ATC AGA TCG AAT GTT GG | This study | znfx-1(xf347) #4 |
| attcttctcctttactcat | LGC Biosearch Technologies | gfp smFISH probe #1 |
| gaggattgggacaactcca | LGC Biosearch Technologies | gfp smFISH probe #2 |
| cgttgacgtctccgtcgag | LGC Biosearch Technologies | gfp smFISH probe #3 |
| ccggagacggagaacttgt | LGC Biosearch Technologies | gfp smFISH probe #4 |
| gtgagctttccgtaggtgg | LGC Biosearch Technologies | gfp smFISH probe #5 |
| gtggtgcagatgaacttga | LGC Biosearch Technologies | gfp smFISH probe #6 |
| catgggactgggagctttc | LGC Biosearch Technologies | gfp smFISH probe #7 |
| aaggtggtgacgagggttg | LGC Biosearch Technologies | gfp smFISH probe #8 |
| agaagcattggactccgta | LGC Biosearch Technologies | gfp smFISH probe #9 |
| cttcatgtggtctgggtaa | LGC Biosearch Technologies | gfp smFISH probe #10 |
| ttgtagagctcgtccattc | LGC Biosearch Technologies | gfp smFISH probe #11 |
| ggcggacttgaagaagtcg | LGC Biosearch Technologies | gfp smFISH probe #12 |
| tcttggacgtatccctctg | LGC Biosearch Technologies | gfp smFISH probe #13 |
| cgtccttgaagaagatggt | LGC Biosearch Technologies | gfp smFISH probe #14 |
| gcacgggtcttgtagtttc | LGC Biosearch Technologies | gfp smFISH probe #15 |
| tgtctccctcgaacttgac | LGC Biosearch Technologies | gfp smFISH probe #16 |
| gctcgatacggttgacgag | LGC Biosearch Technologies | gfp smFISH probe #17 |
| tcctccttgaagtcgattc | LGC Biosearch Technologies | gfp smFISH probe #18 |
| ttgtgtccgaggatgtttc | LGC Biosearch Technologies | gfp smFISH probe #19 |
| gagttgtagttgtactcga | LGC Biosearch Technologies | gfp smFISH probe #20 |
| gccatgatgtagacgttgt | LGC Biosearch Technologies | gfp smFISH probe #21 |
| attccgttcttttgcttgt | LGC Biosearch Technologies | gfp smFISH probe #22 |
| cggatcttgaagttgacct | LGC Biosearch Technologies | gfp smFISH probe #23 |
| gatccgtcctcgatgttgt | LGC Biosearch Technologies | gfp smFISH probe #24 |
| tagtggtcggcgagttgga | LGC Biosearch Technologies | gfp smFISH probe #25 |

| Reagent/resource | Reference or source | Identifier or catalog number |
|---|---|---|
| ccgattggggtgttttgtt | LGC Biosearch Technologies | *gfp* smFISH probe #26 |
| ttgtctgggaggaggactg | LGC Biosearch Technologies | *gfp* smFISH probe #27 |
| gattgggtggagaggtagt | LGC Biosearch Technologies | *gfp* smFISH probe #28 |
| gttttgggtccttggagagg | LGC Biosearch Technologies | *gfp* smFISH probe #29 |
| gaccatgtggtcacgcttc | LGC Biosearch Technologies | *gfp* smFISH probe #30 |
| cggcggtgacgaactcgag | LGC Biosearch Technologies | *gfp* smFISH probe #31 |
| **Chemicals, enzymes, and other reagents** | | |
| 4% Paraformaldehyde | Sigma-Aldrich | #441244 |
| Deionized formamide | Thermo Fisher Scientific | #AM9342 |
| Dextran sulfate | Sigma-Aldrich | #D4911 |
| *E. coli* tRNA | Roche | #10109550001 |
| Vanadyl ribonucleoside complex | NEB BioLabs | #S1402 |
| **Software** | | |
| WormBase | https://wormbase.org/ Harris et al, 2020; Sternberg et al, 2024; Davis et al, 2022 | |
| CRISPOR | http://crispor.tefor.net Haeussler et al, 2016 | |
| **Other** | | |
| THUNDER Imager inverted widefield microscope | Leica | |
| BC43 spinning disk confocal microscope | Andor | |
| ChemiDoc XRS+ System | Bio-Rad | |

No blinding was applied in performing the described experiments. No ethics permits were required for the described work.

## *C. elegans* culture and strains

Unless otherwise stated, all worm strains were cultured according to standard laboratory conditions at 20°C on standard Nematode Growth Medium (NGM) plates seeded with *E. coli* OP50. All strains are in the Bristol N2 background. Every non-'RFK' strain was provided by the *Caenorhabditis* Genetics Center (CGC), which is funded by NIH Office of Research Infrastructure Programs (P40 OD01440). A list of all strains used in this study is provided in the Reagents and Tools Table. Wormbase (Harris et al, 2020; Sternberg et al, 2024; Davis et al, 2022) was extensively used in these studies.

## Mos1-mediated transgenesis

Mos1-mediated Single Copy Insertion (MosSCI) was used to generate the strain RFK1305; *xfSi255[his-67p::gfp::his-67::tbb-2 3'UTR + Cbr-unc-119(+)] II* (Frøkjær-Jensen et al, 2012, 2008). This transgene was targeted to the locus *ttTi5605* on chromosome II. The *his-67* promoter and *tbb-2* 3′ UTR sequence were used for ectopic expression in the whole germline. The *gfp* coding sequence including three introns was amplified from pDD282. An amplified DNA fragment containing the sequences of the *his-67* promoter, *gfp* with three introns, *his-67* CDS and *tbb-2* 3′ UTR was cloned into pCFJ350. All plasmids used for microinjection were purified from 4 ml bacterial culture using PureLink™ HiPure Plasmid Miniprep Kit (Art. No. K210011, Invitrogen™), eluted in sterile water and confirmed by enzymatic digestion and sequencing. A plasmid mix containing 50 ng/μl pCFJ601, 10 ng/μl pMA122, 10 ng/μl pGH8, 5 ng/μl pCFJ104, 2.5 ng/μl pCFJ90, and 50 ng/μl of pRFK4197 were injected in both gonads of 20 young adults of the EG6699 strain. The progeny was screened as previously described (Frøkjær-Jensen et al, 2012, 2008). Successful insertion events were confirmed by Sanger sequencing. The generated strain was outcrossed two times prior to any further crosses or analysis. A list of all plasmids used in this study is provided in the Reagents and Tools Table.

pDD282 was a gift from Bob Goldstein (Addgene plasmid # 66823; http://n2t.net/addgene:66823; RRID:Addgene_66823) (Dickinson et al, 2015). pCFJ601, pMA122, pGH8, pCFJ90, pCFJ104 and pCFJ350 were gifts from Erik Jorgensen (Addgene plasmid # 34874; http://n2t.net/addgene:34874; RRID:Addgene_34874, Addgene plasmid # 34873; http://n2t.net/addgene:34873; RRID:Addgene_34873, Addgene plasmid # 19359; http://n2t.net/addgene:19359; RRID:Addgene_19359, Addgene plasmid # 19327; http://n2t.net/addgene:19327; RRID:Addgene_19327, Addgene plasmid # 19328; http://n2t.net/addgene:19328; RRID:Addgene_19328, Addgene plasmid # 34866; http://n2t.net/addgene:34866; RRID:Addgene_34866).

## CRISPR/Cas9-mediated genome editing

CRISPR/Cas9-mediated genome editing was used to generate the *znfx-1(xf347)* indel allele (8517 bp deletion + 44 bp insertion; 27 bp downstream of endogenous *znfx-1* start codon) using the *unc-58(e665)* co-conversion strategy (Arribere et al, 2014). All protospacer sequences were chosen using CRISPOR (http://crispor.tefor.net) (Haeussler et al, 2016) and cloned in either pRFK2411 (plasmid expressing Cas9 + sgRNA(F + E) (Chen et al, 2013); derived from pDD162) or pRFK2412 (plasmid expressing only sgRNA(F + E) (Chen et al, 2013); derived from pRK2411) via site-directed, ligase-independent mutagenesis (SLIM) (Chiu et al, 2004, 2008; Schreier et al, 2022). pDD162 (*eft-3p::Cas9* + empty sgRNA) was a gift from Bob Goldstein (Addgene plasmid # 47549; http://n2t.net/addgene:47549; RRID:Addgene_47549) (Dickinson et al, 2013). SLIM reactions were transformed in Subcloning Efficiency™ DH5α™ Competent Cells (Art. No. 18265017, Invitrogen™) and plated on LB agar plates supplemented with 100 μg/ml ampicillin. All plasmids used for microinjection were purified from 4 ml bacterial culture using PureLink™ HiPure Plasmid Miniprep Kit (Art. No. K210011, Invitrogen™), eluted in sterile water and confirmed by enzymatic digestion and sequencing. A DNA mix containing 50 ng/μl pRFK2588, 30 ng/μl pRFK3358, 30 ng/μl pRFK3359, 30 ng/μl pRFK3360, 30 ng/μl pRFK3361, and 750 mM SJ763 was injected in both gonads of 20 young adults hermaphrodites maintained at 20°C (Schreier et al, 2022). Selected F1 *unc* progeny was screened for insertion or deletion by PCR. Successful editing events were confirmed by Sanger sequencing. The generated mutant strain was outcrossed two times prior to any further cross

or analysis. A list of all plasmids and protospacer sequences used in this study is provided in the Reagents and Tools Table.

## RNA interference

RNAi was performed via feeding *Escherichia coli* HT115(DE3) expressing double-stranded RNA (Kamath et al, 2003, 2001; Kamath and Ahringer, 2003). The plasmid L4440 served as a negative control and was a gift from Andrew Fire (Addgene plasmid # 1654; http://n2t.net/addgene:1654; RRID:Addgene_1654). A *gfp*-specific sequence was amplified from pDD282 and inserted in L4440 to generate pRFK4103, which was used to perform *gfp* RNAi experiments targeting the transgene *xfSi255[his-67p::gfp::his-67::tbb-2 3'UTR + Cbr-unc-119(+)]*. For each RNAi experiment, HT115(DE3) bacteria transformed with either L4440 or pRFK4103 were freshly grown on LB agar plates supplemented with 100 µg/ml ampicillin and 10 µg/ml tetracycline. Following bleaching of mixed-staged worm strains, embryos were directly transferred to RNAi NGM plates (diameter, 90 mm) and grown at 20°C for one generation. Crosses between L4440-staged hermaphrodites and young adult males were set up after 3 days. Imaging of adult animals was performed after 4 days.

## RNAi inheritance assays

RFK1305 (*xfSi255*) and RFK1405 (*xfSi255; pgl-1(xf233); him-5(e1490)*) served as standard P0 hermaphrodite strain and standard P0 male strain, respectively. *Pgl-1(xf233[pgl-1::mTagRfp-t])* served as a mating control to identify progeny produced by allogamy. A deletion allele of a gene of interest (*goi*) was crossed into both standard P0 strains. The resulting four strains were used to test the effect of the *goi* on specific steps during RNAi inheritance. Each P0 hermaphrodite strain was imaged twice, once in a *pgl-1(+)* and once in a *pgl-1(xf233[pgl-1::mTagRfp-t])* background. While the former was eventually used for mating, the latter was used to analyze fluorescence data according to the established image processing and statistical analysis protocols, which rely on PGL-1::mTagRFP-T. We note that we did not observe any differences between both *pgl-1* genotypes with regard to *gfp* RNAi sensitivity in any P0 hermaphrodite strain. However, we noticed that *wago-1(ok1074); pgl-1(xf233)* double mutants become sterile after three generations.

Paternal *gfp* RNAi inheritance was tested by crossing standard P0 hermaphrodites treated with *control* RNAi (*control* RNAi/*goi(+)*) with four different P0 males of the following conditions: (i) *control* RNAi/*goi(+)*, (ii) *gfp* RNAi/*goi(+)*, (iii) *control* RNAi/*goi(-)*, (iv) *gfp* RNAi/*goi(-)*. These four conditions were also used to assess the effect of the *goi* on *gfp* RNAi sensitivity in P0 males.

Maternal *gfp* RNAi inheritance was tested by crossing standard P0 males treated with *control* RNAi (*control* RNAi / *goi(+)*) with four different P0 hermaphrodites of the following conditions: (i) *control* RNAi / *goi(+)*, (ii) *gfp* RNAi/*goi(+)*, (iii) *control* RNAi/*goi(-)*, (iv) *gfp* RNAi/*goi(-)*. These four conditions were also used to assess the effect of the *goi* on *gfp* RNAi sensitivity in P0 hermaphrodites.

Re-initiation of *gfp* RNAi in F1 animals was tested by two different approaches. First, we analyzed GFP::H4 expression in homozygous mutant F1 animals (*goi(-)*) after maternal *gfp* RNAi inheritance. Therefore, we performed a maternal *gfp* RNAi inheritance assay, in which we combined conditions (iii) and (iv) as described above with

the following condition: (v) P0 hermaphrodites (*gfp* RNAi/*goi(-)*) crossed with P0 males (*control* RNAi/*goi(-)*). Second, we set up a single cross, in which both P0 hermaphrodites and P0 males were heterozygous for the *goi* and treated with *gfp* RNAi (*gfp* RNAi/*goi(+/-)*). F1 animals expressing PGL-1::mTagRFP-T were sorted into two groups based on the presence or absence of visible GFP::H4 expression in the germline. Both groups were split into two subgroups, with the first being used for genotyping and the second being used for microscopy.

For each cross, five L4-staged hermaphrodites and ten young adult males grown for 3 days on RNAi NGM plates were used. Only one male or two hermaphrodites per RNAi NGM plate were selected for mating. To minimize the potential transfer of RNAi-triggering HT115(DE3) bacteria, each animal was hand-picked into a drop of 100 µl M9 buffer, washed thoroughly, and subsequently transferred onto standard NGM plates seeded with *Escherichia coli* OP50 (diameter, 35 mm). After one hour, the RNAi effect was confirmed by microscopy, all animals were combined on a standard NGM mating plate (diameter, 35 mm) and grown overnight at 20°C. On the next day, adult males were hand-picked for single-worm-lysis and adult hermaphrodites were singled onto individual standard NGM plates (diameter, 35 mm) and grown at 20°C. We note that we never detected HT115(DE3) colonies on mating plates (these can be easily distinguished by the morphology of the colony from the OP50 lawn). After 3 days, P0 hermaphrodites and P0 males were genotyped for confirmation. On day later, 25 F1 hermaphrodites and 25 F1 males expressing mTagRFP-T in the germline were hand-picked from each plate, while combining all F1 animals of the same sex for subsequent microscopy (75 animals per slide) and Western blot (50 animals per sample).

Transgenerational *gfp* RNAi inheritance was performed using the standard P0 hermaphrodite strain (*xfSi255*). P0 animals were treated with either *control* RNAi or *gfp* RNAi for one generation as described above. After 4 days of RNAi treatment, F1 embryos were obtained by bleaching and transferred onto standard NGM plates seeded with *Escherichia coli* OP50 (diameter, 90 mm). For each filial generation, ten random hermaphrodites of the post-*control* RNAi condition and ten hermaphrodites with the lowest GFP::H4 expression of the post-*gfp* RNAi condition were singled onto fresh NGM plates seeded with *Escherichia coli* OP50 (diameter, 90 mm) to establish the next generation. Animals were grown at 20°C and 75 adult hermaphrodites per sample were used for microscopy. The experiment was stopped after six filial generations.

## Western blot

Per sample, 50 adult animals, either hermaphrodites or males, were hand-picked in 1× Novex™ NuPAGE™ LDS sample buffer (Art. No. NP0007, Invitrogen™) supplemented with 100 mM DTT and incubated for 30 min at 95°C. Following thorough mixing and centrifugation for 10 min at 21,000 × *g*, supernatants were transferred into fresh tubes and stored at −20°C until usage. Together with Color Prestained Protein Standard, Broad Range (10–250 kDa, Art. No. P7719S, New England BioLabs®), samples were separated on a Novex™ NuPAGE™ 10% Bis-Tris Mini-Protein-Gel (Art. No. NP0301, Invitrogen™) in 1× Novex™ NuPAGE™ MOPS SDS Running Buffer (Art. No. NP0001, Invitrogen™) at 50 mA. Afterward, proteins were transferred on an Immobilon™-P Membrane (PVDF, 0.45 µm, Art. No. IPVH00010, Merck Millipore) for 16 h at 15 V using a Mini Trans-Blot® Cell (Art.

No. 1703930, Bio-Rad) and 1× NuPAGE™ Transfer Buffer (Art. No NP0006, Invitrogen™) supplemented with 20% methanol. Following incubation in 1× PBS supplemented with 5% skim milk and 0.05% Tween®20 for 1 h, the PVDF membrane was incubated in 1× PBS supplemented with 0.5% skim milk, 0.05% Tween®20 and the primary antibody (1:1000 monoclonal mouse anti-GFP (B-2), Art. No. sc-9996, Santa Cruz Biotechnology®/1:5000 monoclonal rabbit β-Actin (D6A8), Art. No. 8457, Cell Signaling Technology®) for 1 h, followed by three washes with 1× PBS supplemented with 0.05% Tween®20 (hereinafter referred to as 0.05% PBS-T) for 10 min each, one hour incubation in 0.05% PBS-T supplemented with the secondary antibody (1:10,000 anti-mouse IgG, HRP-linked antibody, Art. No. 7076, Cell Signaling Technology®/1:10,000 anti-rabbit IgG, HRP-linked antibody, Art. No. 7074, Cell Signaling Technology®) and three final washes with 0.05% PBS-T for 10 min each. Chemiluminescence detection was performed using Amersham™ ECL Select™ Western Blotting Detection Reagent (Art. No. RPN2235, GE Healthcare) and a ChemiDoc™ XRS+ System (Art. No. 1708265, Bio-Rad).

## smFISH

Single-molecule FISH (smFISH) probes targeting *gfp* sequence (Reagents and Tools Table) with Quasar670 dye were designed using the Stellaris Probe Designer (v4.2). For smFISH, 60–80 animals were washed twice in M9 buffer and fixed in 4% paraformaldehyde (Art. No. 441244-1KG, Sigma-Aldrich) in 1× PBS at room temperature for 1 h. After fixation, worms were spun down at $3000 \times g$ in a table-top centrifuge and washed with 1 ml 1× PBS. For permeabilization, worms were stored at 4°C overnight in 1 ml 70% ethanol. On the next day, samples were washed once with 1 ml wash buffer (10% (v/v) deionized formamide (Art. No. AM9342, Thermo Fisher Scientific), 2× SSC) and resuspended in 100 μl hybridization buffer (100 mg/ml dextran sulfate (Art. No. D4911-50G, Sigma-Aldrich), 1 mg/ml *E. coli* tRNA (Art. No. 10109550001, Roche), 2 mM vanadyl ribonucleoside complex (Art. No. S1402S, NEB BioLabs), 0.2 mg/ml BSA ((Art. No. A7906-500G, Sigma-Aldrich), 10% (v/v) deionized formamide (Art. No. AM9342, Thermo Fisher Scientific)) together with 2.5 μl 5 μM probe suspended in TE buffer. Samples were incubated overnight at 30°C and washed twice with 1 ml wash buffer at 37°C for 30 min, the second wash was supplemented with 5 ng/ml DAPI. Additionally, samples were washed twice in 2× SSC buffer and twice in 1× PBS. Finally, animals were resuspended in ibidi mounting medium with DAPI (Art. No. 50011, ibidi) and placed on slides for microscopy.

## Microscopy

Per slide, 75 adult hermaphrodites or males were hand-picked into a drop of 100 μl M9 buffer, washed thoroughly and individually transferred to a drop of 50 μl M9 buffer supplemented with 40 mM sodium azide on a coverslip. After 20 min, excess buffer was removed, and a glass slide containing a freshly made 2% agarose (w/v) pad was placed on top of the coverslip. Animals were immediately imaged using a THUNDER Imager (Leica) inverted widefield microscope equipped with a 10x/0.32 dry objective for tile scan acquisition or 40x/0.95 dry objective for individual animal acquisition. All images were acquired in 16-bit format.

smFISH samples were imaged using BC43 spinning disk confocal microscope equipped with a 40×/0.75 air objective.

## Image processing

ImageJ v.1.54f was used to process all tile scans and images.

### GFP::H4 measurements

For each tile scan, the red emission channel (detecting PGL-1::mTagRFP-T) was duplicated twice. One duplicate was converted to an 8-bit color image, while the second duplicate was used to generate a binary file using the triangle threshold. The dilate function was applied to binary files showing hermaphrodites. If required, this process was repeated with other tile scans of the same microscopy slide, and duplicated worms were removed. Following merge of the 8-bit color image with the binary file, the pencil tool (set color picker to white for both for- and background, set pencil width = 2) was used to manually separate connected germlines from different animals, or germlines from air bubbles, in the binary file using the signal of the 8-bit color image as reference. After splitting channels, the adjusted binary file was used to identify individual germlines using the analyze particles function with a minimum size threshold to exclude speckles and embryos. Using the ROI manager, overlapping germlines were removed and individual gonads of the same animal were combined (for hermaphrodites only). All regions of interest (ROIs) were rechecked, saved and applied to both the green (GFP::H4) and red emission channels (PGL-1::mTagRFP-T) of the original tile scan file(s) to measure the following values: area, mean, standard deviation, modal, median, min, max and integrated density. For hermaphrodites that were not expressing PGL-1::mTagRFP-T (Fig. EV1), tile scans were processed as described above with the following changes: (i) mean threshold was used to generate binary files, (ii) pencil tool was used to manually separate connecting animals.

### gfp smFISH measurements

"Sum of slices" projections of confocal *z* stacks were used to analyze *gfp* smFISH experiments (41 *z* planes per animals; voxel size: 0.152 × 0.152 × 0.195 μm). PGL-1::mTagRFP-T was used to generate the region of interest (ROI) for one gonad per animal, and DAPI signal was used the generate the ROI for the whole animal. ROI(gonad) was subtracted from ROI(animal) to create ROI(autofluorescence), which only included somatic tissues. The inverse of ROI(animal) was used as ROI(background). All four ROIs were rechecked and applied to the blue (DAPI), green (GFP::H4), red (PGL-1::mTagRFP-T) and far-red (*gfp* smFISH) emission channels of the 'Sum of slices' confocal projections to measure the following values: area, mean, standard deviation, modal, median, min, max and integrated density.

## Statistical analyses

### Ratios of integrated fluorescence densities

The GFP::H4 fluorescence signal was of primary interest for the analysis. However, the integrated density (RawIntDen; sum of pixel values) was susceptible to fluctuations in individual body and gonad size, and the mean intensity was susceptible to positional effects. We therefore used the ratio of integrated fluorescence densities,

$$\text{RFI} = \text{RawIntDen}(\text{GFP} :: \text{H4}) / \text{RawIntDen}(\text{PGL}-1 :: \text{mTagRFPT})$$

which corrects for both biases simultaneously. Note that a ratio image generated by the microscopy software should not be used, as the fine morphological structure of the GFP and RFP signals is systematically different.

### F1 datasets were analyzed with Gaussian models in log scale (Figs. 1, 3, 5 and EV2)

We used Gaussian linear models with categorical predictors for RFI values in logarithmic scale. The normal distribution of the log(RFI) data was checked for each data set using Q-Q plots and Shapiro-Wilk tests. In a few cases, the normality assumption was borderline. Due to the large sample size ($n > 50$), the use of Gaussian models was also unproblematic in these cases. Since the homoscedasticity assumption was violated, we fitted models with inverse variance weights. The large sample size makes this an effective method of correcting for heteroskedasticity. One advantage of the analysis in log scale was that statistical hypothesis tests provided information on relative group differences (log fold changes). In the figures the data are presented in original scale, with the end points of the confidence intervals transformed to original scale. Details of the models are given in the following paragraphs (hereafter "Gaussian models"). Models were fitted using the "lm" function in R 4.3.1, and confidence intervals were calculated using the emmeans v1.8.7 package (Lenth, 2023).

The confidence intervals in Fig. 1H are from one-way Gaussian models with the predictor *inheritance type*. They were fitted separately for hermaphrodites and males. The p values belong to tests of linear contrasts

$$H_0 : \mathrm{E}\big[\log\big(\mathrm{RFI}_{\mathrm{group2}}\big)\big] - \mathrm{E}\big[\log\big(\mathrm{RFI}_{\mathrm{group1}}\big)\big]$$
$$= \mathrm{E}\left[\log\left(\frac{\mathrm{RFI}_{\mathrm{group2}}}{\mathrm{RFI}_{\mathrm{group1}}}\right)\right] = 0$$

and were corrected for multiple testing (emmeans method 'mvt', based on multivariate t-distribution).

The data in Figs. 3, 5, and EV2 were analyzed with two-way Gaussian models with interaction term (predictors *genotype* and *RNAi treatment*). Error bars in Figs. 3D–H, 5D–J, and EV2H,I represent 95% confidence intervals for the group means. Figures 3B,C, 5B,C, and EV2J,K show the results of group comparisons. We present the results of the hypothesis tests as confidence intervals of the interaction parameter

$$\beta_{\mathrm{interaction}} = E\left[\log\left(\frac{\mathrm{RFI}_{\mathrm{mut,gfp}}}{\mathrm{RFI}_{\mathrm{mut,ctrl}}}\right) - \log\left(\frac{\mathrm{RFI}_{\mathrm{wt,gfp}}}{\mathrm{RFI}_{\mathrm{wt,ctrl}}}\right)\right]$$

or linear contrasts. This presentation provides more information and is easier to interpret than p values. Note that if a confidence interval does not contain the value 0, this corresponds to a rejection of the null hypothesis stated in the panel.

### P0/F1/F2 datasets were analyzed with nonparametric methods (Figs. 1, 2, 4, EV1, EV2, and EV5)

For datasets where no group comparisons were made or where the assumption of normal distribution was violated, we calculated confidence intervals using the bootstrapping method. In each case, Bias-Corrected and Accelerated Confidence Intervals for the median were calculated using the R-package boot v1.3-28.1

(method 'BCa', $R = 5000$) (Canty and Ripley, 2022). The F1 data in Fig. 6 was also analyzed with a nonparametric model. For group comparisons, appropriate nonparametric tests were used. For datasets with two groups (Figs. 6F, EV1E, EV2B, EV4B–E, and EV5B–E), Wilcoxon rank-sum tests were used. For datasets with more than two groups (Fig. 6B–D), we used Dunn's test and corrected for multiple testing with Holm's method.

### smFISH data were analyzed with unpaired, two-sided Student's t tests

Normalized *gfp* smFISH signal was calculated by dividing the difference of (mean intensity$_{gfp\ \mathrm{smFISH}\ /\ \mathrm{gonad}}$ – mean intensity$_{gfp\ \mathrm{smFISH}\ /\ \mathrm{autofluorescence}}$) with the difference of (mean intensity$_{\mathrm{PGL-1::mTagRFP-T}\ /\ \mathrm{gonad}}$ – mean intensity$_{\mathrm{PGL-1::mTagRFP-T}\ /\ \mathrm{autofluorescence}}$). *P* values were calculated using unpaired, two-sided Student's *t* tests.

Replicates in this study reflect measurements on independently treated animals, hence representing biological replicates.

## Data availability

Primary data is available at BioImage Archive, accession number S-BIAD1922.

The source data of this paper are collected in the following database record: biostudies:S-SCDT-10_1038-S44319-025-00512-7.

## Peer review information

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

## Acknowledgements

The authors thank all members of the Ketting laboratory for critical reading of the manuscript. Joana Costa Pereirinha and Ida Josefine Isolehto are thanked for providing the MosSCI strain expressing GFP::H4 in the whole germline and generating the appropriate *gfp* RNAi clone, respectively. Aaron Noah Ottmann is acknowledged for his contribution to the early stages of this project. We thank the IMB Media Laboratory and Microscopy Core Facility for consumables and equipment. Some strains were provided by the *Caenorhabditis* Genetics Center (CGC), funded by the NIH Office of Research Infrastructure Programs (P40 OD010440). This work was funded by the Deutsche Forschungsgemeinschaft (DFG, German Research Foundation) project ID 252386272 to RFK.

## Author contributions

**Jan Schreier**: Conceptualization; Data curation; Investigation; Visualization; Methodology; Writing—original draft; Project administration. **Lizaveta Pshanichnaya**: Investigation; Methodology; Writing—review and editing. **Fridolin Kielisch**: Software; Visualization. **René F Ketting**: Conceptualization; Supervision; Funding acquisition; Writing—review and editing.

Source data underlying figure panels in this paper may have individual authorship assigned. Where available, figure panel/source data authorship is listed in the following database record: biostudies:S-SCDT-10_1038-S44319-025-00512-7.

## Disclosure and competing interests statement

The authors declare no competing interests.

# Expanded View Figures

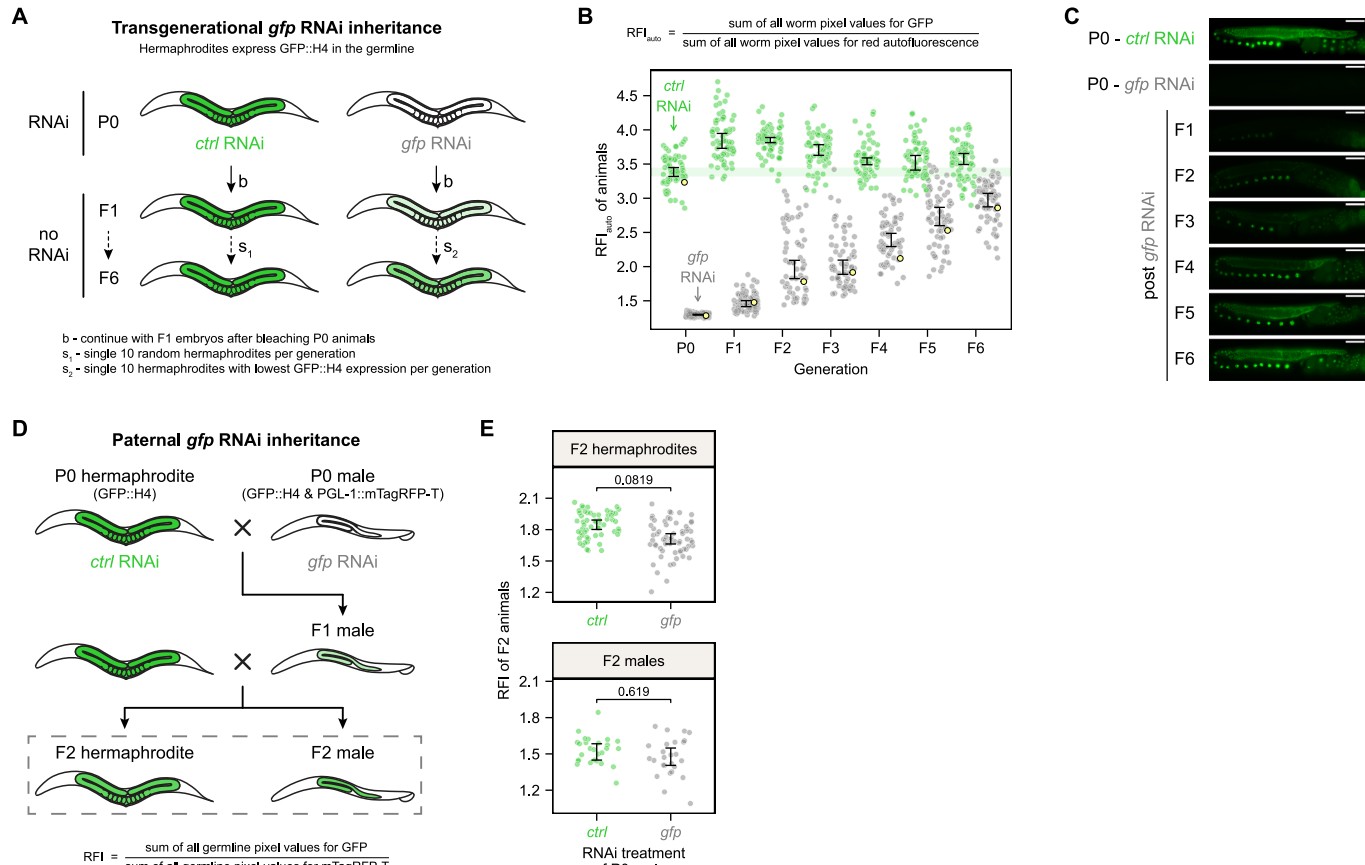

**Figure EV1. Transgenerational *gfp* RNAi inheritance of GFP::H4.**

(A) Schematic representation summarizing the effects of transgenerational *gfp* RNAi inheritance in hermaphrodites expressing GFP::H4 in the germline. Germline color illustrates GFP::H4 expression: green—expressed, shades of light green—expressed at lower levels, white—silenced. (B) Relative fluorescence intensity (RFI$_{auto}$) of GFP::H4 in hermaphrodites before and during transgenerational RNAi inheritance. Only P0 hermaphrodites were treated with either *control* RNAi (green) or *gfp* RNAi (gray). Each dot represents an individual animal, with yellow dots referring to representative micrographs shown in (C). 95% confidence intervals of the median are shown as black error bars for all samples as well as green line for the P0 generation treated with *control* RNAi. Sample sizes: P0 = ~60 animals per condition, F1–F6 = ~60 animals from 10 founders per condition. (C) Widefield fluorescence micrographs of representative hermaphrodites before and during transgenerational *gfp* RNAi inheritance, as indicated in (B). GFP::H4 appears in green. Generations and RNAi conditions are indicated to the left of the micrographs. Scale bars: 50 μm. (D) Crossing scheme summarizing the effect of paternal *gfp* RNAi inheritance on germline GFP::H4 expression in F2 animals. Germline color illustrates GFP::H4 expression: green—expressed, light green—expressed at lower level, white—silenced. (E) Relative fluorescence intensity (RFI) of GFP::H4 in F2 hermaphrodites and F2 males after paternal RNAi inheritance. P0 males were treated with either *control* RNAi (green) or *gfp* RNAi (gray). P0 hermaphrodites were treated with *control* RNAi. Each dot represents an individual animal. 95% confidence intervals of the median, created with bootstrapping (R = 5000) and the BCa (Bias-Corrected and Accelerated Confidence Intervals) method, are shown as black error bars. Statistically significant differences were determined using two-sided unpaired Wilcoxon rank-sum tests. Sample size = ~60 (hermaphrodite) and ~27 (male) F2 animals from 5 F1 founders and 5 P0 founders per condition. Source data are available online for this figure.

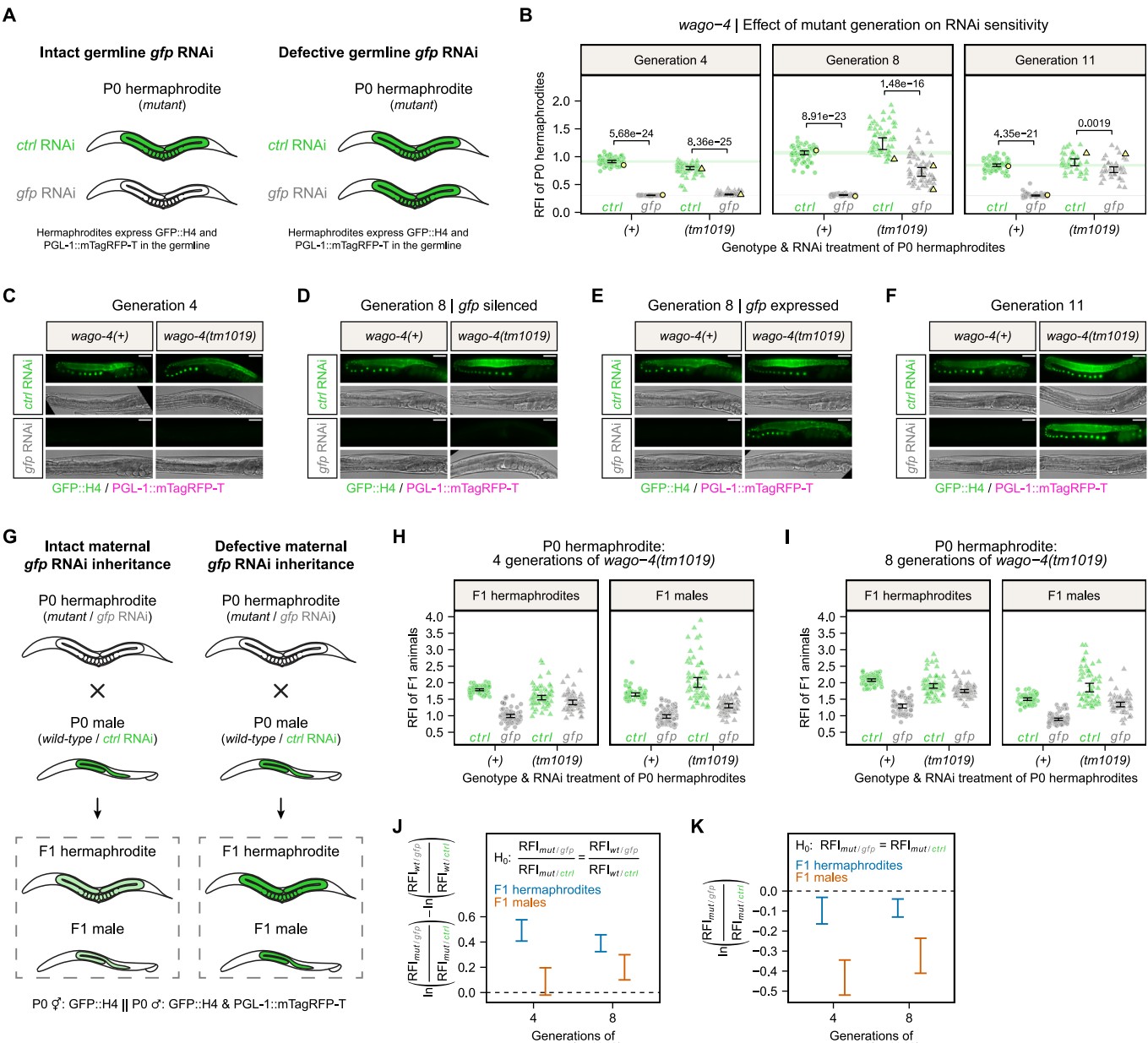

**Figure EV2.** *wago-4* mutant hermaphrodites become transgenerationally insensitive to germline RNAi.

(A) Schematic representations summarizing the *gfp* RNAi effect on mutant P0 hermaphrodites expressing GFP::H4 and PGL-1::mTagRFP-T in the germline. Germline color illustrates GFP::H4 expression: green—expressed, white—silenced. (B) Relative fluorescence intensity (RFI) of GFP::H4 in P0 hermaphrodites treated with either *control* RNAi (green) or *gfp* RNAi (gray). P0 hermaphrodites were either wild-type (circle) or mutant (triangle) for *wago-4*, with the latter being analyzed after 4, 8 and 11 generations of mutant homozygosity. Each dot represents an individual animal, with yellow dots referring to representative micrographs shown in (C–F). 95% confidence intervals of the median are shown as black error bars for all samples as well as green (*control* RNAi) and gray (*gfp* RNAi) lines for wild-type conditions. Statistically significant differences were determined using two-sided Wilcoxon rank-sum tests. Raw data of 'Generation 8' is identical to Fig. 2E. Sample size = ~60 animals per condition. (exception: ~40 *wago-4(tm1019)* animals for 'Generation 11'). (C–F) Widefield fluorescence micrographs of representative P0 hermaphrodites treated with either *control* RNAi or *gfp* RNAi, as indicated in (B). P0 hermaphrodites were either wild-type or mutant for *wago-4*, with latter being analyzed after 4 (C), 8 (D, E) and 11 (F) generations of mutant homozygosity. GFP::H4 and PGL-1::mTagRFP-T appear in green and magenta, respectively. Micrographs in (D) are the same as shown in Fig. 2J. Micrographs in (E) representing (i) *wago-4(+)|ctrl* RNAi, (ii) *wago-4(+)|gfp* RNAi, (iii) *wago-4(tm1019)|ctrl* RNAi are the same as shown in Figs. 2J and EV2D. Scale bars: 50 μm. (G) Crossing schemes summarizing how mutations in P0 hermaphrodites affect germline GFP::H4 expression in F1 animals after maternal *gfp* RNAi inheritance. Germline color illustrates GFP::H4 expression: green—expressed, light green—expressed at lower level, white—silenced. (H, I) Relative fluorescence intensity (RFI) of GFP::H4 in F1 hermaphrodites and F1 males after maternal RNAi inheritance. P0 hermaphrodites were treated with either *control* RNAi (green) or *gfp* RNAi (gray), and were either wild-type (circle) or mutant (triangle) for *wago-4*. Mutant P0 hermaphrodites carried the *wago-4(tm1019)* mutation for either 4 (H) or 8 (I) generations. P0 males were always wild-type for *wago-4* and treated with *control* RNAi. Each dot represents an individual animal. 95% confidence intervals of the mean are shown as black error bars. Raw data of (I) is identical to Fig. 3H. Sample size = ~60 F1 animals from 5 P0 founders per condition. (J) Comparison of the relative GFP::H4 fluorescence reduction after maternal *gfp* RNAi inheritance between F1 animals sired by *wago-4(+)* and *wago-4(tm1019)* P0 hermaphrodites. The plot summarizes the Gaussian models fitted in (H, I) and depicts 95% confidence intervals (CIs) of differences of mean log fold changes. The mean differences in log fold changes are in the center of the CIs. The null hypothesis (H$_0$) expresses equality of relative GFP::H4 fluorescence reduction between wild-type and mutant condition, meaning that the mutation does not cause an enhanced or defective maternal *gfp* RNAi inheritance. A 95% CI not including zero, is equivalent to a rejection of the null hypothesis at the 5% significance level and indicates that the mutation caused a defective (95% CI > 0) maternal *gfp* RNAi inheritance. Color of CIs indicates sex of F1 animals: blue—hermaphrodite, red—male. Sample size = ~60 F1 animals from 5 P0 founders per condition. (K) Comparison of the relative GFP::H4 fluorescence intensity after maternal RNAi inheritance between F1 animals sired by *wago-4(tm1019)* P0 hermaphrodites treated with either *control* RNAi or *gfp* RNAi. The plot summarizes the Gaussian models fitted in (H, I) and depicts 95% confidence intervals (CIs) of mean RFI log fold changes. The mean log fold changes are in the center of the CIs. The null hypothesis (H$_0$) expresses equality of relative GFP::H4 fluorescence intensity between *control* RNAi and *gfp* RNAi treatments, meaning that the mutation causes a completely defective maternal *gfp* RNAi inheritance. A 95% CI not including zero is equivalent to a rejection of the null hypothesis at the 5% significance level and indicates that the mutation does not cause a completely defective maternal *gfp* RNAi inheritance. Color of CIs indicates sex of F1 animals: blue—hermaphrodite, red—male. Sample size = ~60 F1 animals from 5 P0 founders per condition. Source data are available online for this figure.

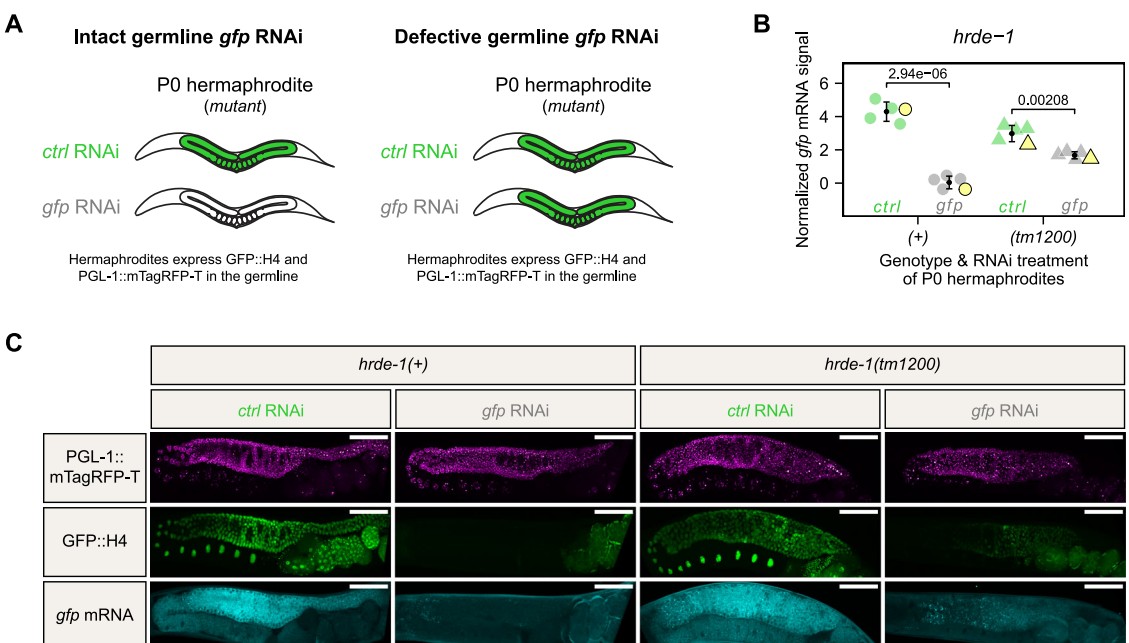

**Figure EV3. *hrde-1* mutant hermaphrodites show impaired RNAi sensitivity.**

(A) Schematic representations summarizing the *gfp* RNAi effect on mutant P0 hermaphrodites expressing GFP::H4 and PGL-1::mTagRFP-T in the germline. Germline color illustrates GFP::H4 expression: green—expressed, white— silenced. (B) Normalized smFISH signal of *gfp* mRNA in P0 hermaphrodites treated with either *control* RNAi (green) or *gfp* RNAi (gray). P0 hermaphrodites were either wild-type (circle) or mutant (triangle) for *hrde-1*. Each colored dot represents an individual animal, with yellow dots referring to representative micrographs shown in (C). The black dot and error bars represent the mean and standard deviation, respectively. *P* values were calculated using an unpaired two-tailed Student's *t* test. Sample size = 5 animals per condition. (C) Confocal maximum intensity projections of representative P0 hermaphrodites treated with either *control* RNAi or *gfp* RNAi, as indicated in (B). P0 hermaphrodites were either wild-type or mutant for *hrde-1*. GFP::H4, PGL-1::mTagRFP-T and *gfp* mRNA appear in green, magenta and cyan, respectively. Scale bars: 50 μm. Source data are available online for this figure.

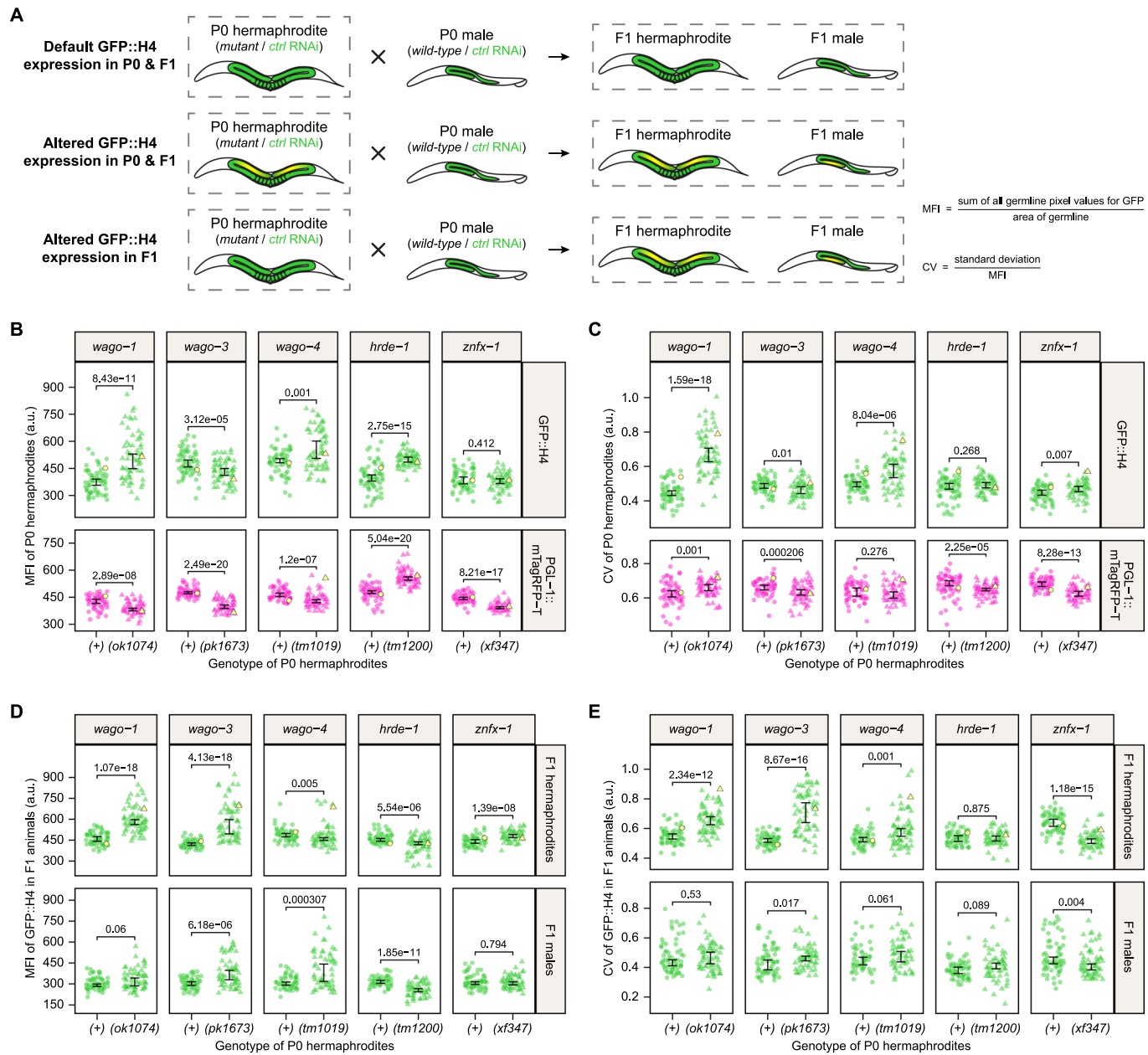

**Figure EV4. Global and local effects of mutant RNAi factors on germline GFP::H4 expression in hermaphrodites and their progeny.**

(A) Crossing schemes summarizing how mutations in P0 hermaphrodites affect germline GFP::H4 expression in F1 animals produced by allogamy. Germline color illustrates GFP::H4 expression: uniformly green—default expression, yellow-green – mutant expression. (B, C) Mean fluorescence intensity (MFI) (B) and coefficient of variation (CV) (C) of GFP::H4 (green) and PGL-1::mTagRFP-T (magenta) in P0 hermaphrodites treated with *control* RNAi. P0 hermaphrodites were either wild-type (circle) or mutant (triangle) for indicated genes. Each dot represents an individual animal, with yellow dots referring to representative animals shown in Fig. 2. 95% confidence intervals of the median are shown as black error bars. Statistically significant differences were determined using two-sided Wilcoxon rank-sum tests. Raw data is identical to Fig. 2. The CV is defined as the ratio of the standard deviation to the mean. Sample size = ~60 animals per condition. (D, E) Mean fluorescence intensity (MFI) (D) and coefficient of variation (CV) (E) of GFP::H4 in F1 hermaphrodites and F1 males after maternal RNAi inheritance. P0 hermaphrodites were treated with *control* RNAi, and were either wild-type (circle) or mutant (triangle) for indicated genes. P0 males were always wild-type for these genes and treated with *control* RNAi. Each dot represents an individual animal, with yellow dots referring to representative animals shown in Fig. 3. 95% confidence intervals of the median are shown as black error bars. Statistically significant differences were determined using two-sided Wilcoxon rank-sum tests. Raw data is identical to Fig. 3. The CV is defined as the ratio of the standard deviation to the mean. Sample size = ~60 F1 animals from 5 P0 founders per condition. Source data are available online for this figure.

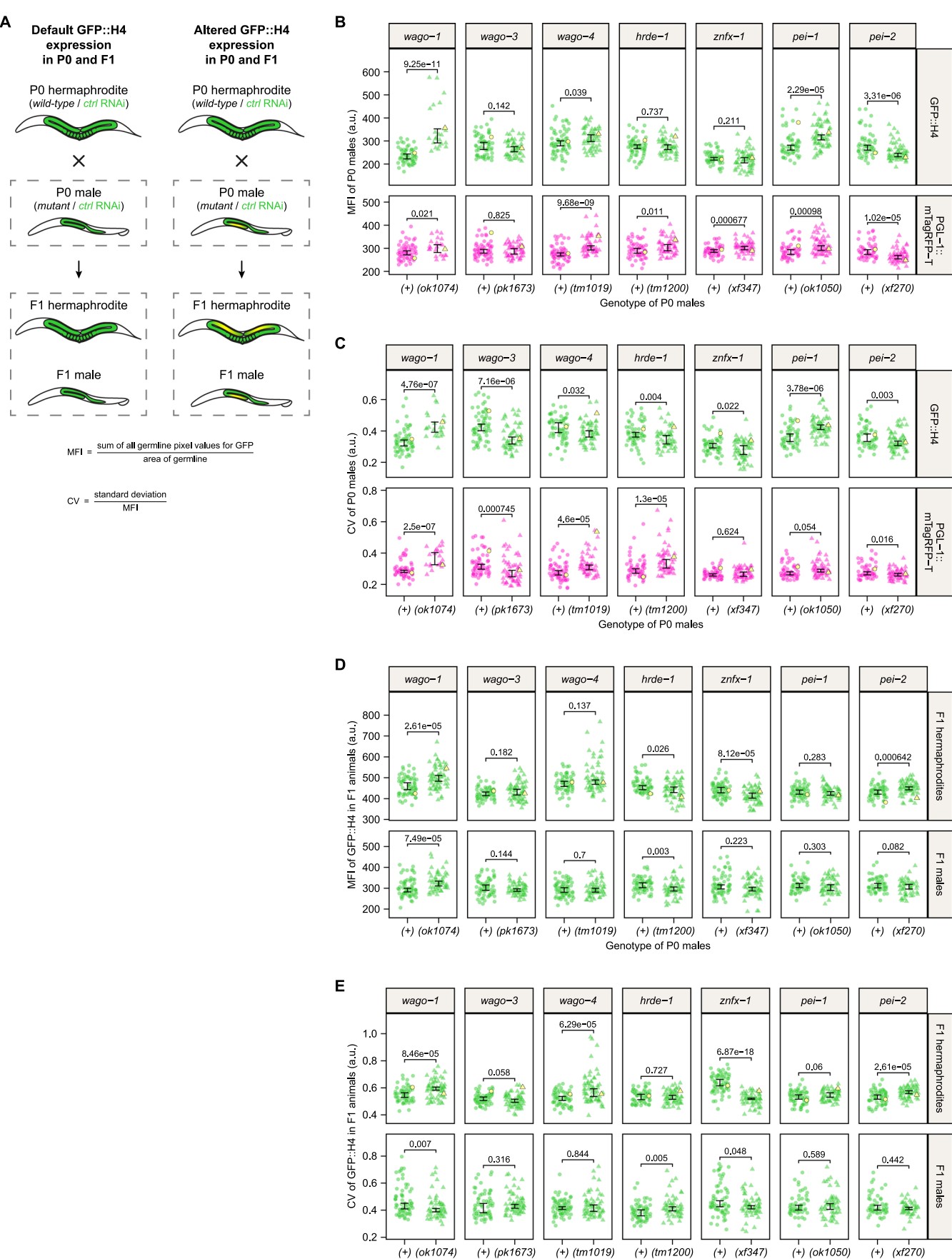

**Figure EV5. Global and local effects of mutant RNAi factors on germline GFP::H4 expression in males and their progeny.**

(A) Crossing schemes summarizing how mutations in P0 hermaphrodites affect germline GFP::H4 expression in F1 animals produced by allogamy. Germline color illustrates GFP::H4 expression: uniformly green—default expression, yellow-green—mutant expression. (B-C), Mean fluorescence intensity (MFI) (B) and coefficient of variation (CV) (C) of GFP::H4 (green) and PGL-1::mTagRFP-T (magenta) in P0 males treated with *control* RNAi. P0 males were either wild-type (circle) or mutant (triangle) for indicated genes. Each dot represents an individual animal, with yellow dots referring to representative animals shown in Fig. 4. 95% confidence intervals of the median are shown as black error bars. Statistically significant differences were determined using two-sided Wilcoxon rank-sum tests. Raw data is identical to Fig. 4. The CV is defined as the ratio of the standard deviation to the mean. Sample size = ~60 animals per condition (exception: ~23 *wago-1(ok1074)* animals). (D, E) Mean fluorescence intensity (MFI) (D) and coefficient of variation (CV) (E) of GFP::H4 in F1 hermaphrodites and F1 males after paternal RNAi inheritance. P0 males were treated with *control* RNAi, and were either wild-type (circle) or mutant (triangle) for indicated genes. P0 hermaphrodites were always wild-type for these genes and treated with *control* RNAi. Each dot represents an individual animal, with yellow dots referring to representative animals shown in Fig. 5. 95% confidence intervals of the median are shown as black error bars. Statistically significant differences were determined using two-sided Wilcoxon rank-sum tests. Raw data is identical to Fig. 5. The CV is defined as the ratio of the standard deviation to the mean. Sample size = ~60 F1 animals from 5 P0 founders per condition. Source data are available online for this figure.

