## [Peer Review File · EMBO Reports]

A genetic framework for RNAi inheritance in *Caenorhabditis elegans*

Jan Schreier, Fridolin Kielisch, Rene Ketting, and Lizaveta Pshanichnaya

Corresponding author(s): Rene Ketting (r.ketting@imb-mainz.de)

Review Timeline:

Submission Date:	1st Oct 24
Editorial Decision:	25th Oct 24
Revision Received:	30th Apr 25
Editorial Decision:	28th May 25
Revision Received:	11th Jun 25
Accepted:	20th Jun 25

Editor: Esther Schnapp

Transaction Report:

Dear Rene,

Thank you for the submission of your manuscript to EMBO reports. We have now received the full set of referee reports that is pasted below.

As you will see, the referees acknowledge that the findings are potentially interesting. However, they also have several suggestions for how the study could be improved and I think that all suggestions are good and should be addressed. Especially comment 1 from referee 1 should be addressed as proposed. Please let me know in case you disagree and we can discuss the exact revision requirements further, also in a video chat, if you like.

I would thus like to invite you to revise your manuscript with the understanding that the referee concerns must be fully addressed and their suggestions taken on board. Please address all referee concerns in a complete point-by-point response. Acceptance of the manuscript will depend on a positive outcome of a second round of review. It is EMBO reports policy to allow a single round of major revision only and acceptance or rejection of the manuscript will therefore depend on the completeness of your responses included in the next, final version of the manuscript.

We realize that it is difficult to revise to a specific deadline. In the interest of protecting the conceptual advance provided by the work, we recommend a revision within 3 months (25th Jan 2025). Please discuss the revision progress ahead of this time with the editor if you require more time to complete the revisions.

- 1) A data availability section providing access to data deposited in public databases is missing. If you have not deposited any data, please add a sentence to the data availability section that explains that.
- 2) Your manuscript contains statistics and error bars based on $n=2$. Please use scatter blots in these cases. No statistics should be calculated if $n=2$.

2) individual production quality figure files as .eps, .tif, .jpg (one file per figure). See https://wol-prod-cdn.literatumonline.com/pb-assets/embopress-site/EMBOPress_Figure_Guidelines_061115-1561436025777.pdf for more info on how to prepare your figures.

3) We replaced Supplementary Information with Expanded View (EV) Figures and Tables that are collapsible/expandable online. A maximum of 5 EV Figures can be typeset. EV Figures should be cited as 'Figure EV1, Figure EV2' etc... in the text and their respective legends should be included in the main text after the legends of regular figures.

<<https://www.embopress.org/page/journal/14693178/authorguide#expandedview>>

5) a complete author checklist, which you can download from our author guidelines

<<https://www.embopress.org/page/journal/14693178/authorguide>>. Please insert information in the checklist that is also reflected in the manuscript. The completed author checklist will also be part of the RPF.

6) Please note that all corresponding authors are required to supply an ORCID ID for their name upon submission of a revised manuscript (<<https://orcid.org/>>). Please find instructions on how to link your ORCID ID to your account in our manuscript tracking system in our Author guidelines

<<https://www.embopress.org/page/journal/14693178/authorguide#authorshipguidelines>>

10) Regarding data quantification (see Figure Legends:

<https://www.embopress.org/page/journal/14693178/authorguide#figureformat>)

12) All Materials and Methods need to be described in the main text using our 'Structured Methods' format, which is required for all research articles. According to this format, the Methods section includes a separate file Reagents and Tools Table (listing key reagents, experimental models, software and relevant equipment and including their sources and relevant identifiers) and a Methods and Protocols section describing the methods using a step-by-step protocol format. The aim is to facilitate adoption of the methodologies across labs. More information on how to adhere to this format as well as a downloadable template (.docx) for the Reagents and Tools Table can be found in our author guidelines:

An example of a Method paper with Structured Methods can be found here: <https://www.embopress.org/doi/full/10.1038/s44320-024-00037-6#sec-4>

I look forward to seeing a revised form of your manuscript when it is ready.

Best wishes,
Esther

Referee #1:

In this manuscript by Schreier et al., the authors aim to investigate the contribution of known RNAi factors in paternal and maternal RNAi initiation, inheritance, and re-establishment. Several previous studies have explored RNAi inheritance in GFP transgenes (multicopy or single-copy) or endogenous genes (namely oma-1) and identified factors required for initiation and maintenance. However, none of these studies have addressed the contribution of female and male gametes in heritable RNAi silencing. In this regard, this manuscript makes an important contribution to the field. The experimental design is clever, and the results are clear and well-presented.

The authors initiate RNAi in the female or male germline and observe heritable silencing in both cases, though the male contribution is weaker than the female. They also identify the factors participating in this process. Interestingly, the nuclear Argonaute HRDE-1 and the Argonaute WAGO-4 are fully required for RNAi initiation in the male germline. While WAGO-4 is also required for RNAi initiation in the female germline, HRDE-1 is only needed to establish silencing in the mitotic region. Another unexpected finding, compared to previous studies, is that ZNFX-1 and HRDE-1 are not required for oocyte RNAi inheritance, but their zygotic expression is essential for re-establishing silencing. Instead, WAGO-3 and WAGO-4 are required for oocyte RNAi inheritance, while WAGO-3 and PEI-1 are essential for RNAi inheritance from the male germline. Overall, these experiments provide a clearer picture of the paternal and maternal contributions of each factor in RNAi initiation, inheritance, and maintenance.

I have only a few comments that I hope the authors can address before the manuscript is ready for publication.

Specific comments:

1. Experimental approach:

Previous studies have used various approaches to investigate RNAi establishment and inheritance. Some have employed GFP multicopy transgenes, others have used single-copy transgenes or RNAi of endogenous genes. Additionally, the Seydoux lab has used smRNA FISH as a more precise readout of the silencing process compared to the GFP signal from very stable histone proteins. I suggest that the authors use smFISH probes for GFP RNAi and demonstrate that their key results, particularly regarding RNAi initiation, are consistent with these more sensitive methods. Similarly, it would be interesting to test whether RNAi targeting endogenous genes, such as oma-1 or mex-6, follows the same maternal and paternal inheritance patterns observed for GFP transgenes. This would help resolve some of the controversy in the field regarding the targets of RNAi (transgenes vs. endogenous genes).

2. Discussion of recent findings:

The authors might also comment on their results in light of the recent work by Rieger et al. (2023) from the Rechavi lab, which demonstrated the role of ZNFX-1 in nucleus-independent RNAi inheritance and showed that the nucleus is dispensable for heritable RNAi. For instance, in this manuscript, HRDE-1 appears to be fully required for the establishment and inheritance of RNAi in the male germline and necessary to re-establish silencing in the context of heritable RNAi from the female germline. It would be helpful for the authors to discuss whether they believe HRDE-1 functions in heritable RNAi in a nuclear-independent manner, in line with Rieger et al. (2023), or if the differing approaches might explain the contrasting findings.

Minor comment:

- On lines 74-75, the authors write, "We treated animals for one generation (embryo until adulthood)." Since RNAi by feeding does not work in embryos, it might be more accurate to state "from hatched L1 until adulthood."

Referee #2:

In this manuscript, the authors established a new assay to distinguish the roles of sperm and oocytes in RNAi inheritance. They also identified the functions of known RNAi factors in different steps that enable robust RNAi inheritance. Most surprisingly, HRDE-1 and ZNFX-1 were found to be unnecessary for RNAi inheritance (in contrast to previous studies using self-fertilizing hermaphrodites), but instead required for establishing gene silencing in the offspring. This study sheds new light on the molecular mechanisms that govern RNAi inheritance in *C. elegans*. I have raised a few comments for the authors to address:

Major comments:

1. Line 12-13: the role of ZNFX-1 balancing endo- and exo-siRNA is speculative. The authors will need to substantiate the claim experimentally by sequencing small RNA from RNAi-treated worms, for example, or tone it down/remove it from the abstract.
2. Line 32-33: there are other instances where inheritance is extended, not just in *prg-1* mutants. For example, *met-2(-)* (PMID: 28343968) and mating-induced epigenetic silencing (PMID: 34244495).
3. Please provide more background information on the different Argonates and their roles in RNAi inheritance in hermaphrodites to provide more context for readers who are unfamiliar with the worm small RNA field.
4. While the authors claimed that RNAi is removed after washing and cleaning the worms on OP50 plate, it's hard to be sure there's no transfer of RNAi bacteria. Since this experimental set up is used throughout the paper and is important for the claims, I suggest repeating some key experiments using streptomycin plate to inhibit the growth of HT115 and OP50-1 which is streptomycin resistant.
5. Related to point 3. Line 466-467 - how are the HT115 colonies detected?
6. Is oocyte- or sperm-mediated RNAi transgenerationally inherited?
7. Using F1 to assay the strength of RNAi inheritance is tricky, as they are directly exposed to the dsRNA trigger. For example, although RNAi is apparent in F1 of *hrde-1(-)* and *znfx-1(-)* hermaphrodite, this effect may or may not persist in the subsequent generations. So, it may not be fair to conclude that maternal inheritance of RNAi occurs independent of HRDE-1 and ZNFX-1. The same applies to WAGO-1, ZNFX-1, and PEI-2, which are found to be not required for sperm-mediated inheritance.
8. While the authors find that *gfp* transgene is robustly silenced by RNAi in male gonad/sperm, this phenomenon may be restricted to transgenes and not observed when endogenous genes are targeted (e.g., PMID: 24581041), as the authors briefly mentioned in line 64-67. Therefore, sperm-mediated transmission of RNAi the authors described may be restricted to transgenes. The authors should discuss this point further, along with its potential implication to help readers better interpret the findings.
9. I am curious about the potential differences between sperm provided by males and self-sperm from hermaphrodites. The authors observed that while *hrde-1(-)* oocytes can transmit RNAi, *hrde-1(-)* sperm do not. This raises the question of whether the RNAi inheritance defects previously observed in *hrde-1(-)* hermaphrodites could stem from defects in their self-sperm. Although the authors claim that HRDE-1 is required for establishing silencing in the progeny, it is possible that HRDE-1 plays distinct roles at different stages of RNAi inheritance.
10. Line 293-295: I don't follow the logic of the argument regarding the interactions between WAGO-3 and HRDE-1. The argument assumes that *hrde-1(-)* is inheritance defective but the authors' results suggest otherwise (males fail to establish RNAi)? Please rephrase.
11. Line 313-316: I again don't follow the logic of the argument. What is the evidence that HRDE-1 drives WAGO-3 activity in the male germline? Did the authors suggest redundancy between the different WAGOs? Please rephrase.

Minor comments:

1. Line 131: F1 "hermaphrodites"
2. Line 86-88: very confusing sentence. Can you please rephrase?
3. Just out of curiosity, does *wago-4(-)* males also show transgenerational loss of RNAi?
4. Line 199: what is the evidence that silencing is established in the embryo but not in the F1 germline?

Referee #3:

This manuscript examines inheritance patterns of RNA-mediated interference in *C. elegans*. It has been demonstrated previously that hermaphrodites exposed to double-stranded RNA by feeding (P0 generation) will silence the corresponding gene and pass on the silencing effect to their progeny (F1 generation). RNAi inheritance via a male parent has also been reported.

In this new study, the authors use a new transgene-based assay to directly compare RNAi inheritance to progeny (F1) from hermaphrodites vs males parents (P0) exposed to RNAi and to evaluate the role of several Argonates and the helicase ZNFX-1 in this process. The authors convincingly demonstrate that silencing can be propagated to F1 progeny from the paternal germline, although the effect appears weaker than when propagated through the maternal germline. Consistent with prior findings from this lab (<https://pubmed.ncbi.nlm.nih.gov/35132225/>), they demonstrate that WAGO-3, an Argonaute present in mature sperm, is required for paternal inheritance of RNAi. They also report that WAGO-3 is also required for maternal inheritance, but do not provide a molecular basis for this new finding. The authors also make several claims about HRDE-1 and ZNFX-1 which appear less solid and appear to contradict prior findings in the literature.

A potential issue with this new study is that the RNAi effect is evaluated solely by quantifying PROTEIN expression from a transgene targeted by RNAi. Given that RNAi targets mRNA not protein, this protein-focused assay may not be the most reliable

assay for quantifying silencing, especially since the dynamics of GFP::H4 protein synthesis and decay in the syncytial germline are not known. This issue is especially concerning given that the authors come to several conclusions that differ from those obtained in a recent study that used *in situ* hybridization to directly look at mRNA levels following exposure to dsRNA in P0 and F1 animals (Ouyang et al., <https://pubmed.ncbi.nlm.nih.gov/35739318/>). That study provided a genetic framework for RNAi establishment in P0s and transmission in F1s: Silencing in the P0 generation depends primarily on primary sRNAs (synthesized independently of HRDE-1 and ZNFX-1), whereas silencing in the F1 generation depends on secondary sRNAs synthesized by two redundant amplification pathways dependent on HRDE-1 and ZNFX-1. Those studies were done in selfing hermaphrodites and did not consider paternal vs maternal transmission, which is an added bonus of the present study. Nevertheless, the authors should discuss how the genetic model of Ouyang et al. compares with their own findings, and how assaying for protein versus RNA levels could cloud their interpretations (see below for specific comments).

Specific comments:

1. The authors use a transgene based on the *his-67* locus. *his-67* codes for a replication-dependent histone H4 which is predicted to produce a transcript that is only transcribed during S phase and rapidly degraded during mitosis. The transgene uses the histone promoter, the histone coding region fused to a GFP with introns, and replaces the endogenous *his-67* 3'UTR with a tubulin 3' UTR. Presumably this transgene is transcribed in the distal region of the germline (the only region where germ cells undergo S phase) but whether it is also transcribed and/or maintained as RNA/protein in the rest of the germline where germ cells are in meiosis and eventually become transcriptionally quiescent is not clear. The authors only present throughout the paper photomicrographs of adult hermaphrodites which show GFP::H4 protein in all germline nuclei. Without further experiments, it is not clear how this protein pattern reflects the RNA pattern, especially since the GFP::H4 protein could perdure in germ cells long after the RNA is destroyed.
2. Line 112: "HRDE-1 is specifically required for exogenous RNAi-mediated gene silencing in the distal germline". This conclusion does not consider the possibility that the transgene is ONLY transcribed in the distal germline, with the RNA and/or protein perduring throughout the germline. This is a reasonable possibility given that the transgene is driven by the *his-67* promoter, which is predicted to be active only in S phase (distal-most germ cells). If that is the case, the observation that *hrde-1* mutants retain GFP::H4 expression specifically in the distal region would be consistent with a defect in silencing transcription/nascent transcripts at the locus. This interpretation would be consistent with Ouyang et al., 2022 who showed that *hrde-1* mutants are defective in the silencing of nascent transcripts but retain the ability to degrade RNAi-targeted transcripts post-synthesis in the cytoplasm.
3. Paragraph starting on line 116: "HRDE-1 and ZNFX-1 are not required for inheritance". Ouyang et al., 2022 found that HRDE-1 and ZNFX-1 participate in two complementary branches of the RNAi amplification machinery and that both contribute to the production of secondary siRNAs that are necessary for silencing in F1 animals. Ouyang et al., 2022 found that single mutants had reduced but not zero silencing, and that the double mutant *hrde-1znfx-1* was completely deficient in silencing in the F1. The authors should explain why their conclusions contradict these prior findings.
4. Line 147: Could the apparent stronger RNAi defect of *hrde-1* males compared to hermaphrodites simply be due to longer perdurance of GFP::H4 in spermatogenic vs oogenic germlines?? It is possible that oogenic and spermatogenic germlines both require *hrde-1* to silence nascent transcripts to the same extent, but that only oogenic germlines turn over GFP::H4 protein synthesized in the mitotic zone?

Some general remarks on this revision:

We thank the reviewers for their constructive comments, and we apologize for the long delay in sending our work back. We know it is always hard to get back into a paper after a long period, but as our lead author had left the lab, we had organizational issues to get the revisions going.

Below you will find detailed responses to all the raised issues. We chose to remain focused on *gfp* RNAi in this manuscript, as the comparison between male and female inheritance will not work with f.i. *mex-6* RNAi. Also, detailed comparisons with previous work (such as Ouyang et al. 2022) are difficult because of the very different RNAi regime: we treat from L1-adult stage, while Ouyang et al treat only for a few hours. We find that this may cause significant differences, making any direct comparisons less meaningful than one may expect. Our resulting model, however, we believe brings together much of the published work. We realize that some aspects have not been proven yet, but we hope that you will agree that this is a useful conceptual model, driven by the available data, that may help forward the overall field of RNAi inheritance in *C. elegans*.

Referee #1:**1. Experimental approach:**

Previous studies have used various approaches to investigate RNAi establishment and inheritance. Some have employed GFP multicopy transgenes, others have used single-copy transgenes or RNAi of endogenous genes. Additionally, the Seydoux lab has used smRNA FISH as a more precise readout of the silencing process compared to the GFP signal from very stable histone proteins. I suggest that the authors use smFISH probes for GFP RNAi and demonstrate that their key results, particularly regarding RNAi initiation, are consistent with these more sensitive methods. Similarly, it would be interesting to test whether RNAi targeting endogenous genes, such as *oma-1* or *mex-6*, follows the same maternal and paternal inheritance patterns observed for GFP transgenes. This would help resolve some of the controversy in the field regarding the targets of RNAi (transgenes vs. endogenous genes).

Author reply:

RNAi against GFP, using GFP as read-out has been very useful in past studies to understand RNAi. Indeed, a more precise read-out is smFISH, and we do agree that smFISH experiments would be useful. To start to address this, we performed *gfp* smFISH experiments on wild-type P0 and F1 animals. This revealed the following findings:

smFISH against *gfp* in P0:

- 1) *gfp* mRNA is expressed in male and female gonads; diffuse signal throughout most of the tissue in hermaphrodites, whereas the signal is more restricted to early germ cell stages in the male. Some small granules detected at the end of pachytene in hermaphrodites.
- 2) Upon RNAi *gfp* mRNA is reduced, and perinuclear granules become more pronounced. These are near or overlap with PGL-1.
- 3) We do not detect strong *gfp* mRNA foci around nuclei in oocytes.

smFISH against *gfp* in F1 from RNAi parents (male of female):

- 1) *gfp* mRNA is reduced in male and female gonads from and we detect foci around nuclei around the pachytene zone. These are near or overlap with PGL-1. In males such foci are much weaker.
- 2) We do not detect clear *gfp* mRNA foci around nuclei in oocytes.

Overall, GFP protein silencing closely follows RNA levels in both male and female RNAi inheritance and hence. We conclude that our protein read-out is suitable to study RNAi inheritance. These data are shown in Figure 1 and are presented on page 5.

We did not follow-up on RNAi against an endogenous gene like *mex-6*. First, we could not identify a suitable gene that we could silence in the male, while still allowing mating and fertility. As our paper is on comparing male and female inheritance, this would cripple the experiment significantly. Second, we want to stress that using RNAi against GFP has been extremely useful in the past. In fact, with RNAi being a major mechanism recognizing foreign DNA/RNA, one could argue GFP may in fact be better suited than endogenous genes, which may have evolved to evade certain aspects of RNAi. Nevertheless, endogenous RNAi certainly would be great to include, but in the already long timeframe of revision, we opted not to pursue that angle.

2. Discussion of recent findings:

The authors might also comment on their results in light of the recent work by Rieger et al. (2023) from the Rechavi lab, which demonstrated the role of ZNFX-1 in nucleus-independent RNAi inheritance and showed that the nucleus is dispensable for heritable RNAi. For instance, in this manuscript, HRDE-1 appears to be fully required for the establishment and inheritance of RNAi in the male germline and necessary to re-establish silencing in the context of heritable RNAi from the female germline. It would be helpful for the authors to discuss whether they believe HRDE-1 functions in heritable RNAi in a nuclear-independent manner, in line with Rieger et al. (2023), or if the differing approaches might explain the contrasting findings.

Author reply:

We believe that our results actually match those described by Rieger et al. In our model WAGO-3 is responsible for inheritance, which is a cytoplasmic protein. Where we implicate HRDE-1 in inheritance (especially in the male) we proposed that it may function in loading WAGO-3 in the P0. These results do not contrast with the data from Rieger et al. We adapted the text to make this clear and included citation of Rieger et al. (2023) in the discussion, pages 14 and 15. We apologize for failing to cite this relevant work initially.

Minor comment:

• On lines 74-75, the authors write, "We treated animals for one generation (embryo until adulthood)." Since RNAi by feeding does not work in embryos, it might be more accurate to state "from hatched L1 until adulthood."

Author reply:

We agree and changed the sentence accordingly (Page 5).

Referee #2:

Major comments:

1. Line 12-13: the role of ZNFX-1 balancing endo- and exo-siRNA is speculative. The authors will need to substantiate the claim experimentally by sequencing small RNA from RNAi-treated worms, for example, or tone it down/remove it from the abstract.

Author reply:

We agree and rephrased the sentence accordingly (Page 2, abstract).

2. Line 32-33: there are other instances where inheritance is extended, not just in *prg-1* mutants. For example, *met-2(-)* (PMID: 28343968) and mating-induced epigenetic silencing (PMID: 34244495).

Author reply:

We included the mentioned studies (Page 3).

3. Please provide more background information on the different Argonautes and their roles in RNAi inheritance in hermaphrodites to provide more context for readers who are unfamiliar with the worm small RNA field.

Author reply:

We expanded our introduction and provided more information on Argonaute proteins and their roles in RNAi inheritance to provide more context for readers (Page 3-4). Note, however, that not much is known about specific roles of Argonautes in inheritance.

4. While the authors claimed that RNAi is removed after washing and cleaning the worms on OP50 plate, it's hard to be sure there's no transfer of RNAi bacteria. Since this experimental set up is used throughout the paper and is important for the claims, I suggest repeating some key experiments using streptomycin plate to inhibit the growth of HT115 and OP50-1 which is streptomycin resistant.

Author reply:

HT115 bacteria colonies are easily visible on OP50 bacteria lawns after 24 hours incubation at 20 degrees. We made sure to check every used plate for potential HT115 contamination and removed plates that showed such contamination. We note, however, that this hardly ever occurred and was never observed in our presented experiments. This is described in the Methods section. Furthermore, all experiments include three intrinsic controls which strongly indicate that no RNAi effects from accidental transfer of HT115 bacteria occurred. Finally, the fact that *wago-3* mutants have no defect in establishment, yet are fully inheritance defective, strongly argues that silencing in the F1 does not stem from contamination by HT115 bacteria.

5. Related to point 3. Line 466-467 - how are the HT115 colonies detected?

Author reply:

We used a conventional compound microscope that is capable of Nomarski and differential interference contrast optics to check OP50 plates for HT115 colonies. These are easily detected simply by their appearance. This has been added to the Methods section (Page 19).

6. Is oocyte- or sperm-mediated RNAi transgenerationally inherited?

Author reply:

We tested paternal *gfp* RNAi inheritance until the F2 generation (twice via the father only) and found that the *gfp* silencing hardly persisted until the F2 generation. We assume that the transgenerational aspect of RNAi inheritance may be either due to oocyte-mediated inheritance factors or due to the combined inheritance via both gametes. Since this manuscript focuses on the F1, we have restricted ourselves to commenting that sperm-mediated inheritance does not last beyond the F1 (Page 6, Figure EV1D-E). Whether this reflects a fundamental difference in mechanism beyond the F1 or simply reflects strength/abundance of the transmitted signal is an interesting question, but not one we address in this manuscript.

7. Using F1 to assay the strength of RNAi inheritance is tricky, as they are directly exposed to the dsRNA trigger. For example, although RNAi is apparent in F1 of *hrde-1(-)* and *znfx-1(-)* hermaphrodite, this effect may or may not persist in the subsequent generations. So, it may not be fair to conclude that maternal inheritance of RNAi occurs independent of HRDE-1 and ZNFX-1. The same applies to WAGO-1, ZNFX-1, and PEI-2, which are found to be not required for sperm-mediated inheritance.

Author reply:

If exposure of the F1 directly to dsRNA would be a significant factor in our assays, it should result in the same genetic requirements in P0 and F1. This is not the case. For instance, wild-type P0 responds to dsRNA, whereas the wild-type F1 of *wago-3* mutant parents does not. In addition, *znfx-1* mutant F1s do not silence, whereas *znfx-1* mutant P0 animals do. Furthermore, the fact that males can pass on silencing to embryos developing in unexposed mothers, in a *wago-3* dependent manner, is another strong indication the dsRNA exposure of the F1 is not significant in our assays. Hence, we feel that our conclusion that HRDE-1 and ZNFX-1 is needed for establishment of RNAi in both P0 and F1, but not for inheritance is valid. Note that ‘inheritance’ in this phrase is intended to indicate the process of molecules being transferred between parent and offspring. In the overall process of RNAi inheritance establishment in either P0 or F1 are of course essential steps.

8. While the authors find that *gfp* transgene is robustly silenced by RNAi in male gonad/sperm, this phenomenon may be restricted to transgenes and not observed when endogenous genes are targeted (e.g., PMID: 24581041), as the authors briefly mentioned in line 64-67. Therefore, sperm-mediated transmission of RNAi the authors described may be restricted to transgenes. The authors should discuss this point further, along with its potential implication to help readers better interpret the findings.

Author reply:

The indicated reference mentions inefficient RNAi against sperm genes, not that RNAi does not work. As the cited work also describes that endogenous RNAi works well in males, there may not be an intrinsic RNAi problem in males. We have now included reference to this work in the introduction (Page 5), but do not further discuss it.

9. I am curious about the potential differences between sperm provided by males and self-sperm from hermaphrodites. The authors observed that while *hrde-1(-)* oocytes can transmit RNAi, *hrde-1(-)* sperm do not. This raises the question of whether the RNAi inheritance defects previously observed in *hrde-1(-)* hermaphrodites could stem from defects in their self-sperm. Although the authors claim that HRDE-1 is required for establishing silencing in the progeny, it is possible that HRDE-1 plays distinct roles at different stages of RNAi inheritance.

Author reply:

First, the reviewer touches on an interesting point. In fact, we are convinced that differences between self-sperm and male sperm do exist, but we cannot go into this direction in this manuscript.

Second, regarding differential HRDE-1 function: while it is possible that HRDE-1 acts differently at different stages, the most parsimonious explanation that fits all data (published and ours) is that HRDE-1 acts in establishment in both P0 and F1. That establishment function was simply missed in the earlier studies because the assessed transgene only yields GFP expression in the oocytes, where *hrde-1* mutants indeed do not have an RNAi defect (this is what we also see: GFP expression of our transgene in oocytes is still silenced). This oocyte-resident RNAi in *hrde-1* mutants likely is post-transcriptional and could be linked to the generation of inheritance triggers, just like HRDE-1 is. As maternal inheritance in our assays depends on WAGO-3, this inheritance trigger generated by cytoplasmic RNAi may also be WAGO-3. This is now discussed (Page 11-15).

Finally, the initiation of the RNAi response in an animal by dsRNA or via inherited WAGO-3 will likely differ in many aspects, and the effect of HRDE-1 may likewise be different. However, in our model (Figure 7B, Page 15) we do not need to hypothesize differential functions of HRDE-1. We do not posit that this model is THE truth, but we do believe it is a useful framework that will hopefully be further tested, corrected and/or improved through future experimentation.

10. Line 293-295: I don't follow the logic of the argument regarding the interactions between WAGO-3 and HRDE-1. The argument assumes that *hrde-1(-)* is inheritance defective but the authors' results suggest otherwise (males fail to establish RNAi)? Please rephrase.

Author reply:

We have rephrased the discussion to better explain this idea. We also included a model to graphically depict it (Figure 7B).

11. Line 313-316: I again don't follow the logic of the argument. What is the evidence that HRDE-1 drives WAGO-3 activity in the male germline? Did the authors suggest redundancy between the different WAGOs? Please rephrase.

Author reply:

See reply to point 10 of this reviewer.

Minor comments:

1. Line 131: F1 "hermaphrodites"

Author reply:

We have changed the sentence accordingly.

2. Line 86-88: very confusing sentence. Can you please rephrase?

Author reply:

We rephrased the sentence.

3. Just out of curiosity, does *wago-4(-)* males also show transgenerational loss of RNAi?

Author reply:

We did not observe this phenomenon in *wago-4* mutant males; we mention this now on page 6. As we really do not understand the mechanisms behind this, we chose not to speculate about the underlying reasons.

4. Line 199: what is the evidence that silencing is established in the embryo but not in the F1 germline?

Author reply:

We cannot say when the silencing is established during development of the F1, as we only imaged adult F1 animals. Possibly, re-establishment might only occur in the PGCs (Z2 and Z3), but dedicated experiments following the development of F1 animals sired by RNAi treated parents will be necessary to address this question. This goes beyond this manuscript. We commented, however, on this (in our eyes) very important aspect, in the discussion (Page 11).

Referee #3:

The authors also make several claims about HRDE-1 and ZNFX-1 which appear less solid and appear to contradict prior findings in the literature.

We disagree with this general assessment. The main results on HRDE-1 and ZNFX-1 are on their requirements for establishment of silencing are supported by clear data. We can see this because of our experimental set-up, which was thus far never used. There is only one apparent contradiction to previous literature. This concerns our finding that loss of just HRDE-1 or ZNFX-1 resulted in a loss of silencing in the F1. This issue is addressed below (point 3).

A potential issue with this new study is that the RNAi effect is evaluated solely by quantifying PROTEIN expression from a transgene targeted by RNAi. Given that RNAi targets mRNA not protein, this protein-focused assay may not be the most reliable assay for quantifying silencing, especially since the dynamics of GFP::H4 protein synthesis and decay in the syncytial germline are not known. This issue is especially concerning given that the authors come to several conclusions that differ from those obtained in a recent study that used *in situ* hybridization to directly look at mRNA levels following exposure to dsRNA in P0 and F1 animals (Ouyang et al., <https://pubmed.ncbi.nlm.nih.gov/35739318/>). That study provided a genetic framework for RNAi establishment in P0s and transmission in F1s: Silencing in the P0 generation depends primarily on primary sRNAs (synthesized independently of HRDE-1 and ZNFX-1), whereas silencing in the F1 generation depends on secondary sRNAs synthesized by two redundant amplification pathways dependent on HRDE-1 and ZNFX-1.

The study referred to did not show that silencing in the P0 was independent of HRDE-1. It was shown that the 22G RNA levels did not largely depend on HRDE-1 or ZNFX-1, but HRDE-1 was shown to be required for silencing in the distal part of the gonad. This is what we also report.

Those studies were done in selfing hermaphrodites and did not consider paternal vs maternal transmission, which is an added bonus of the present study. Nevertheless, the authors should discuss how the genetic model of Ouyang et al. compares with their own findings, and how assaying for protein versus RNA levels could cloud their interpretations (see below for specific comments).

We now present a discussion on this aspect, including a model to depict a model that is consistent with our data and the data from Ouyang et al. (Page 11-15; Figure 7B).

Specific comments:

1. The authors use a transgene based on the *his-67* locus. *his-67* codes for a replication-dependent histone H4 which is predicted to produce a transcript that is only transcribed during S phase and rapidly degraded during mitosis. The transgene uses the histone promoter, the histone coding region fused to a GFP with introns, and replaces the endogenous *his-67* 3'UTR with a tubulin 3' UTR. Presumably this transgene is transcribed in the distal region of the germline (the only region where germ cells undergo S phase) but whether it is also transcribed and/or maintained as RNA/protein in the rest of the germline where germ cells are in meiosis and eventually become transcriptionally quiescent is not clear. The authors only present throughout the paper photomicrographs of adult hermaphrodites which show GFP::H4 protein in all germline nuclei. Without further experiments, it is not clear how this protein pattern reflects the RNA pattern, especially since the GFP::H4 protein could perdure in germ cells long after the RNA is destroyed.

Author reply:

We note that in our assays we expose the animals to dsRNA from L1 stage onwards (the eggs hatch on RNAi food). Hence, the silencing effects likely is established long before we analyze the adult animals, likely making turnover times of histone proteins less relevant compared to studying RNAi only hours after inducing the effect (also see point 4, below). Nevertheless, we have analyzed the expression of RNA from our transgene. In the female gonad we find it throughout the gonad as cytoplasmic signal. In males the RNA signal is more restricted to earlier germ cell development stages. Weak perinuclear signal is detectable. Following RNAi we see increased perinuclear signals around the pachytene zone, and a general decrease in cytoplasmic signal in both male and female gonads. Decrease of RNA levels and perinuclear foci can also be detected in the F1 animals. Overall, mRNA nicely correlates with GFP protein levels, indicating that GFP protein readout is a good proxy for mRNA levels in our assays. Only in the later spermatogenic stages do we see GFP signal in absence of mRNA, indicating that here protein turnover is slow in these cells. This does, however, not affect our conclusions. Data relevant to this issue are shown in Figure 1 (wild-type) and EV3 (*hrde-1* mutant).

2. Line 112: "HRDE-1 is specifically required for exogenous RNAi-mediated gene silencing in the distal germline". This conclusion does not consider the possibility that the transgene is ONLY transcribed in the distal germline, with the RNA and/or protein perduring throughout the germline. This is a reasonable possibility given that the transgene is driven by the his-67 promoter, which is predicted to be active only in S phase (distal-most germ cells). If that is the case, the observation that *hrde-1* mutants retain GFP::H4 expression specifically in the distal region would be consistent with a defect in silencing transcription/nascent transcripts at the locus. This interpretation would be consistent with Ouyang et al., 2022 who showed that *hrde-1* mutants are defective in the silencing of nascent transcripts but retain the ability to degrade RNAi-targeted transcripts post-synthesis in the cytoplasm.

Author reply:

We believe that what the reviewer depicts here is precisely what is happening. Our cited wording was not precise enough to reflect our meaning. Nevertheless, the fact that silencing in *hrde-1* mutants still occurs in the proximal region (also see point 4, below) implies that silencing there is HRDE-1 independent. Most likely, RNAi in that region is at the post-transcriptional level. We have adjusted our wording (Page 6).

3. Paragraph starting on line 116: "HRDE-1 and ZNFX-1 are not required for inheritance". Ouyang et al., 2022 found that HRDE-1 and ZNFX-1 participate in two complementary branches of the RNAi amplification machinery and that both contribute to the production of secondary siRNAs that are necessary for silencing in F1 animals. Ouyang et al., 2022 found that single mutants had reduced but not zero silencing, and that the double mutant *hrde-1znfx-1* was completely deficient in silencing in the F1. The authors should explain why their conclusions contradict these prior findings.

Author reply:

We have dedicated a specific section of our discussion to this issue (Page 14-15; Figure 7B).

In brief, experimental differences, including the time of feeding, but also the strength of the inherited RNAi response may be significant factors explaining the observed differences.

4. Line 147: Could the apparent stronger RNAi defect of *hrde-1* males compared to hermaphrodites simply be due to longer perdurance of GFP::H4 in spermatogenic vs oogenic germlines?? It is possible

that oogenic and spermatogenic germlines both require *hrde-1* to silence nascent transcripts to the same extent, but that only oogenic germlines turn over GFP::H4 protein synthesized in the mitotic zone?

Author reply:

Histone-GFP turnover in the female gonad is indeed likely to happen, otherwise we could not obtain loss of GFP signal in the proximal part only. Also, Buckley et al. (2012) showed that a transgene can express histone-GFP specifically in the oocytes, again pointing at turnover. Furthermore, turnover in the male germline indeed appears to be slow, as the GFP mRNA is produced only in the distal region, while GFP remains present throughout the later stages as well, including sperm (new Data Figure 1L). Hence, turnover may indeed contribute to the observed differences in RNAi between *hrde-1* mutant P0 males and hermaphrodites. We have added this to the discussion section, page 12.

We note, however, that if there is fast turnover in the female gonad, this implies that in the more proximal regions (i.e. the oocytes) histone-GFP protein is produced locally, as our transgenic strain expresses GFP all over the gonad. Loss of GFP specifically in the proximal region, like we observe, therefore is likely to result from interference with RNA locally. In other words, an HRDE-1 independent RNAi mechanism acting proximally. This possibility is fully supported by the paper originally identifying HRDE-1 (Buckley et al. 2012), where it was shown that an oocyte-expressed reporter was still silenced by RNAi in *hrde-1* mutants. Clearly, in this system no GFP turnover effect can affect the conclusion that an HRDE-1 independent RNAi mechanism is active in oocytes. We propose that this HRDE-1 independent mechanism drives the inheritance we observe from *hrde-1* mutant hermaphrodites, and that can only be observed upon crossing, to generate *hrde-1* heterozygous offspring. Given that *hrde-1* mutant males do not produce inheritance signals, we hypothesize that this HRDE-1-independent mechanism does not operate in males. This all is represented in Figure 7B.

Dear Rene,

Thank you for the submission of your revised manuscript. We have now received the enclosed reports from the referees as well as cross-comments. As you will see, referee 3 is not satisfied with the revised study but referees 1 and 2 agree that referee 3's comments can be addressed in the ms text. I would therefore like to ask you to do so.

A few editorial comments will also need to be addressed before we can proceed with the official acceptance of your manuscript:

- Please reduce the number of keywords to 5.
- The Data Availability Section needs to be placed before the Acknowledgments.
- The author credits need to be removed from the ms file. All credits need to be entered during online ms submission.
- These callouts need to be corrected in all places in the text: Figures S1, S2B, S3, S4; Fig. 6F, S1B, S3B-E, S4B-E. It should be Figure EV1, etc.
- The 4 Supplementary tables can all be part of the Reagents & Tools table that is currently missing. The Methods section should include a separate file called Reagents and Tools Table (listing key reagents, experimental models, software and relevant equipment and including their sources and relevant identifiers). A downloadable template (.docx) for the Reagents and Tools Table can be found in our author guidelines: <
<https://www.embopress.org/page/journal/14693178/authorguide#manuscriptpreparation>>.
- Materials and Methods should be just Methods.
- ORCIDs are not needed on the title page.

Our routine image analysis of to be accepted ms detected a possible reuse of cells between:

- * Figure 2J and Figure EV2 D&E (3 images).
- * Within Figure 5 O and Q. (different treatments - should be different)

Can you please clarify what happened?

Figure Legends - Comments

- Please note that the exact p values are not provided in the legends of figure 1H
- Please indicate the statistical test used for data analysis in the legends of figure EV2 B
- Please note that information related to n is missing in the legends of figures 3B, C; 5B, C
- Please note that the measure of center for the error bars needs to be defined in the legends of figures 3B, C; 5B, C; EV2 J, K

I would like to suggest a few minor changes to the abstract that needs to be written in present tense. Please let me know whether you agree with this:

Gene regulation by RNA interference (RNAi) is a conserved process driven by double-stranded RNA (dsRNA). It responds to exogenous cues and drives endogenous gene regulation. In *Caenorhabditis elegans*, RNAi can be inherited from parents to offspring. While a number of factors have been implicated in this inheritance process, we do not understand how and when they function. Using a new inheritance assay, we establish a hierarchy amongst previously identified inheritance factors. We show that the nuclear argonaute protein HRDE-1 is required for RNAi establishment in parents and offspring, but not for the inheritance process. In contrast, the cytoplasmic argonaute protein WAGO-3 is the only factor essential for inheritance, via sperm and oocyte, while not affecting establishment in either parent or offspring. We propose a cycle between nuclear and cytoplasmic argonaute proteins, where nuclear activity drives most of the silencing and cytoplasmic activity ensures inheritance. Finally, we implicate the RNA helicase ZNF-1 as a factor that links the inherited WAGO-3 protein to nuclear RNAi in the offspring.

EMBO press papers are accompanied online by A) a short (1-2 sentences) summary of the findings and their significance, B) 2-3 bullet points highlighting key results and C) a synopsis image that is exactly 550 pixels wide and 200-600 pixels high (the

height is variable). The synopsis image should provide a sketch of the major findings, like a graphical abstract. Please note that text needs to be readable at the final size. Please send us this information along with the final manuscript.

Referee #1:

The authors have satisfactorily addressed all of my previous concerns and have made a commendable effort to include additional experiments, even though the lead authors had already left the laboratory at the time of revision.

My main concern was related to using GFP protein levels as the sole readout for silencing effects. The authors now convincingly demonstrate that these effects are also observed at the mRNA level, providing a more comprehensive and robust analysis. Notably, the mRNA silencing observed in the gonad is consistent with previous reports, further supporting their conclusions.

I also appreciate the authors' inclusion of a discussion addressing prior work and clarifying a previously noted discrepancy. This addition significantly improves the manuscript's clarity and context.

In summary, the revised manuscript represents a valuable and rigorous contribution to the field of small RNA inheritance. I fully support its publication in EMBO Reports.

Referee #2:

The authors have done a great job addressing my questions. I only have a few remaining comments:

- 1) Since wago-4(-) exhibits generational effects, please indicate the generation number in panels 2E, 3H, 4G, and 5D for clarity. I assume that early-generation wago-4 mutants were used for the crosses in Figures 3 and 5. Please add the information to the figure and main text.
- 2) Do all the mutants used represent severe loss-of-function alleles? Please clarify the nature of each allele in the text and provide a summary in a supplemental table
- 3) Line 396: "explain ZNFX-1-independent..."
- 4) Line 375: typo. "...in the context of our model..."
- 5) Line 32: missing comma. "...in many cases,..."
- 6) Line 52-53: missing comma. "...Also, Argonuate proteins..."
- 7) Line 86-93: and thorough the paper: italicize *gfp* and keep the nomenclature consistent.
- 8) Line 270: "...When during development..." weird sentence. Please rephrase.
- 9) Line 273: typo. "...inheritance via the mother"
- 10) Line 324: typo. "...is functional in hrde-1 mutants..."
- 11) Line 362: missing commas. "...It is therefore feasible that, in the Mis-assay, loss..."
- 12) Line 367-368: remove comma. "...to better understand this is that we do..."
- 13) Line 376: typo. " initial RNAi response focused on..."
- 14) Line 414: typo. "...brought to RdRP substrates by..."

Referee #3:

This study examines pattern of RNAi inheritance in hermaphrodite and male *C. elegans*. A genetic framework for RNAi inheritance in selfing hermaphrodites has already been derived from many studies examining silencing of endogenous loci. These studies have yielded a working model for RNAi inheritance that involve parallel cycles of sRNA amplification that independently target transcripts in the nucleus and cytoplasm (described below). The authors use a flawed assay (which examines protein rather than RNA and consequently cannot distinguish between RNA silencing in the nucleus vs the cytoplasm) and claims to establish a new genetic hierarchy, which does not incorporate, and in some cases appear contrary to, prior findings. Their findings are also purely genetics and hence do not address molecular underpinnings. Because of the problems

inherent to their assay, it is difficult to understand how their data builds a conceptual framework that supersedes prior models, which incorporated molecular models of sRNA amplification etc..

Prior model: dsRNA ingested by *C. elegans* hermaphrodites can initiate silencing that is transmitted to the next generation. Prior studies have shown that "transgenerational RNAi" involves at least three independent sRNA amplification cycles that use the targeted transcript as a template to generate silencing small RNAs (sRNAs). In the first amplification cycle (RDE-1 dependent), primary sRNAs (generated from the dsRNA trigger) anneal to the targeted transcript, tagging it as a template for the generation of secondary sRNAs. Secondary sRNAs cause degradation of most of the targeted transcript in the cytoplasm of the animal exposed to the dsRNA trigger (P0). The secondary sRNAs also activate in the P0 animal two additional amplification cycles. These are required for maintenance of the silencing in the P0 and propagation of the silencing to the next generation. The HRDE-1-dependent cycle is a nuclear cycle that generates tertiary sRNAs from nascent transcripts in the nucleus. The HRDE-1 cycle reduces transcription at the locus but maintains a low level of nascent transcripts for continuous production of tertiary sRNAs. The ZNFX-1-dependent cycle also generates tertiary sRNAs but using mature transcripts in the cytoplasm as templates. These tertiary sRNAs ensure the continuous degradation of cytoplasmic transcripts. Like the HRDE-1 cycle, the ZNFX-1 cycle reduces does not eliminate all transcripts, but maintains low levels of mature transcripts (in the cytoplasmic P granules) to ensure continuous production of tertiary sRNAs. By maintaining a residual pool of transcripts to use as templates for sRNA amplification, the HRDE-1 and ZNFX-1 cycles "memorialize" the silencing for transmission across generations. Loss of either the HRDE-1 or the ZNFX-1 cycle impairs but does not eliminate the RNAi response, as the two cycles can operate independently in the nucleus (HRDE-1 cycle, which reduces nascent transcript production) and in the cytoplasm (ZNFX-1 cycle, which degrades mature transcripts). Only loss of both HRDE-1 and ZNFX-1 cycles at the same time eliminates silencing propagation completely. In *hrde-1;znfx-1* double mutants, only the RDE-1 cycle remains which transiently reduces cytoplasmic transcripts leading to apparent silencing in the P0 animal, but this effect is temporary and not transmitted to the next generation (Ouyang et al., 2022 and references therein).

New study: The new study by Schreier et al., proposes to extend prior studies by comparing RNAi inheritance patterns in males vs hermaphrodites and examining the roles of additional genes implicated in RNAi. Unlike prior studies which directly monitored the silencing of endogenous transcripts using in situ hybridization (monitoring both nuclear and cytoplasmic transcripts), this study utilizes a transgene expressed in the germline and follows protein (GFP) expression from the transgene. Consequently, the authors cannot determine whether the silencing patterns they are observing are due to silencing of nascent transcripts at the locus or silencing of mature transcripts in the cytoplasm. Additionally, the authors do not examine sRNA production as did prior studies. These concerns were brought up by reviewers in the last round of reviews, but only partially addressed in the revision. For example, although the authors present some in situ hybridization experiments, they do not establish where the transgene is first transcribed in the germline (by visualizing nascent transcripts in nuclei). In *hrde-1* hermaphrodites (P0), they find that silencing is defective in the distal germline but not in the proximal germline. They conclude that HRDE-1 is required for RNAi establishment, but not for "transmission" in hermaphrodites, presumably because they observe normal silencing in oocytes. This reasoning is flawed: silencing in oocytes does not necessarily imply transmission to the next generation. Rather it implies that the cytoplasmic cycle of silencing is working (oocytes are transcriptionally silent and only accumulate mature transcripts synthesized in younger germ cells). HRDE-1 is only required to silence nascent transcripts which in hermaphrodites are most prominent in the distal germline and absent in oocytes. The authors observed that *hrde-1* mutant hermaphrodites (mated to wild-type males) "contrary to expectations" still "passed on *gfp* RNAi to both F1 hermaphrodites and males". This is expected as per the model above, as the ZNFX-1 cytoplasmic cycle is still active in these animals. Furthermore, this observation does not mean that *hrde-1* is NOT required for transmission. If the authors had looked, they would have noticed that *hrde-1* F1 animals, despite silencing cytoplasmic transcripts, fail to silence nascent transcripts in nuclei. Similarly the findings that HRDE-1 and ZNFX-1 are required for re-establishment of RNAi in the F1s (line 220) is predicted from the model above, since each cycle is required to re-amplify the response in each new generation. If the authors had assayed RNA instead of GFP, they might have noticed that each SINGLE mutant only transmits and restart a partial silencing response due to the presence of the other cycle. This paper does present new data regarding patterns of male inheritance and the role of WAGO-3 and WAGO-4. However, without a more careful integration of their findings with prior models in the literature, and direct examination and quantification of cytoplasmic and nascent transcripts, it is difficult to interpret these data. The authors make several claims that appear at odds with prior data. For example, in the abstract: "nuclear activity drives most of the silencing and cytoplasmic activity ensures inheritance". This appears to contradict the model above where nuclear (HRDE-1) and cytoplasmic (ZNFX-1) cycles were shown to function independently: each are sufficient for partial silencing and transmission. Per the model above, SILENCING refers to a reduction in nascent transcript production (HRDE-1) and degradation of mature transcripts in the cytoplasm (ZNFX-1) and TRANSMISSION refers to cycles of sRNA amplification using residual nuclear transcripts (HRDE-1) or residual mature transcripts in P granules (ZNFX-1). The authors perhaps are using a different definition of silencing and transmission? In summary, while the authors have attempted to respond to the prior reviews with a few in situ hybridization, their findings remain poorly integrated with the prior literature and in some cases appear to have been miss-interpreted due to their inability to distinguish between silencing of nascent versus mature transcripts.

Cross-comments from referee 1:

At first glance, I would say that maybe the authors need to re-write the ms taking into consideration the comments by ref #3. Maybe they can additionally quantify their nuclear signal from smFISH to assess nuclear silencing (no requirement for additional experiment, just re-analysis), but overall their smFISH recapitulated the results from protein expression in my opinion.

Cross-comments from referee 2:

In my opinion textual edits would suffice.

Rebuttal

We thank the editorial team and the reviewers for having a second look at our manuscript. Below we will address the remaining issues, starting with the comments from the editors. Following that, we will address the reviewers' comments. In the manuscript text, major text changes have been labelled in cyan.

Responses to editorial issues:

A few editorial comments will also need to be addressed before we can proceed with the official acceptance of your manuscript:

- Please reduce the number of keywords to 5.
- The Data Availability Section needs to be placed before the Acknowledgments.
- The author credits need to be removed from the ms file. All credits need to be entered during online ms submission.
- These callouts need to be corrected in all places in the text: Figures S1, S2B, S3, S4; Fig. 6F, S1B, S3B-E, S4B-E. It should be Figure EV1, etc.
- The 4 Supplementary tables can all be part of the Reagents & Tools table that is currently missing. The Methods section should include a separate file called Reagents and Tools Table (listing key reagents, experimental models, software and relevant equipment and including their sources and relevant identifiers). A downloadable template (.docx) for the Reagents and Tools Table can be found in our author guidelines:
< <https://www.embopress.org/page/journal/14693178/authorguide#manuscriptpreparation>>.
- Materials and Methods should be just Methods.
- ORCID IDs are not needed on the title page.

Response:

These have all been corrected.

Our routine image analysis of to be accepted ms detected a possible reuse of cells between:

- * Figure 2J and Figure EV2 D&E (3 images).

Response:

We apologize for the confusion. We decided to present the images this way for better understanding and comparison of the presented data. Figure EV2B shows the effect of 4, 8 and 11 generations of wago-4 mutant homozygosity, with the data of "generation 8" being also shown in Figure 2E. Consequently, the representative micrographs for this dataset are shown twice for better understanding (Figure 2J and Figure EV2D-E), with Figure EV2D and Figure EV2E sharing the control images as indicated in Figure EV2B. In order to make this clearer for the reader, we included the wago-4 generation number in Figure 2, 3, 4, 5 (as also asked by Referee #2) and the main text, and also expanded the figure legend of Figure EV2 by clearly indicating the cross-reference to data shown in Figure 2. We hope this adequately addresses the raised point.

- * Within Figure 5 O and Q. (different treatments - should be different)

Response:

The images showing the wild-type/gfp RNAi condition in Figure 5O and Figure 5Q were indeed the same. Both genes, pei-1 and pei-2, were analyzed in parallel and shared their respective wild-type controls. We have now replaced the duplicated image in Figure 5O with another representative image and adjusted the plot in Figure 5H accordingly (yellow dot referring to representative micrographs shown in Figure 5O). We hope this resolves this issue.

- *Figure Legends - Comments*

- Please note that the exact p values are not provided in the legends of figure 1H

Response:

These are given in the figure.

- Please indicate the statistical test used for data analysis in the legends of figure EV2 B

Response:

This has been added.

- Please note that information related to n is missing in the legends of figures 3B, C; 5B, C

Response:

This has been added

- Please note that the measure of center for the error bars needs to be defined in the legends of figures 3B, C; 5B, C; EV2 J, K

Response:

We don't show point estimates + error bars in the plot, but we show confidence intervals. We have made the legends more precise, indicating to which quantities the confidence intervals belong ("confidence intervals (CIs) of mean differences in log fold changes", etc.). We also added the sentence "The mean differences in log fold changes are in the center of the CIs".

I would like to suggest a few minor changes to the abstract that needs to be written in present tense. Please let me know whether you agree with this:

Gene regulation by RNA interference (RNAi) is a conserved process driven by double-stranded RNA (dsRNA). It responds to exogenous cues and drives endogenous gene regulation. In *Caenorhabditis elegans*, RNAi can be inherited from parents to offspring. While a number of factors have been implicated in this inheritance process, we do not understand how and when they function. Using a new inheritance assay, we establish a hierarchy amongst previously identified inheritance factors. We show that the nuclear argonaute protein HRDE-1 is required for RNAi establishment in parents and offspring, but not for the inheritance process. In contrast, the cytoplasmic argonaute protein WAGO-3 is the only factor essential for inheritance, via sperm and oocyte, while not affecting establishment in either parent or offspring. We propose a cycle between nuclear and cytoplasmic argonaute proteins, where nuclear activity drives most of the silencing and cytoplasmic activity ensures inheritance. Finally, we implicate the RNA helicase ZNFX-1 as a factor that links the inherited WAGO-3 protein to nuclear RNAi in the offspring.

Response:

These changes have been adopted in the revised abstract.

EMBO press papers are accompanied online by A) a short (1-2 sentences) summary of the findings and their significance, B) 2-3 bullet points highlighting key results and C) a synopsis image that is exactly 550 pixels wide and 200-600 pixels high (the height is variable). The synopsis image should provide a sketch of the major findings, like a graphical abstract. Please note that text needs to be readable at the final size. Please send us this information along with the final manuscript.

Response:

These have been added to the submission.

Responses to Reviewers' issues:

Referee #1:

The authors have satisfactorily addressed all of my previous concerns and have made a commendable effort to include additional experiments, even though the lead authors had already left the laboratory at the time of revision.

My main concern was related to using GFP protein levels as the sole readout for silencing effects. The authors now convincingly demonstrate that these effects are also observed at the mRNA level, providing a more comprehensive and robust analysis. Notably, the mRNA silencing observed in the gonad is consistent with previous reports, further supporting their conclusions.

I also appreciate the authors' inclusion of a discussion addressing prior work and clarifying a previously noted discrepancy. This addition significantly improves the manuscript's clarity and context.

In summary, the revised manuscript represents a valuable and rigorous contribution to the field of small RNA inheritance. I fully support its publication in EMBO Reports.

Response:

We thank the reviewer for the supportive comments.

Referee #2:

The authors have done a great job addressing my questions. I only have a few remaining comments:

- 1) Since wago-4(-) exhibits generational effects, please indicate the generation number in panels 2E, 3H, 4G, and 5D for clarity. I assume that early-generation wago-4 mutants were used for the crosses in Figures 3 and 5. Please add the information to the figure and main text.
- 2) Do all the mutants used represent severe loss-of-function alleles? Please clarify the nature of each allele in the text and provide a summary in a supplemental table
- 3) Line 396: "explain ZNFX-1-independent..."
- 4) Line 375: typo. "...in the context of our model..."
- 5) Line 32: missing comma. "...in many cases,..."
- 6) Line 52-53: missing comma. "...Also, Argonuate proteins..."
- 7) Line 86-93: and thorough the paper: italicize gfp and keep the nomenclature consistent.
- 8) Line 270: "...When during development..." weird sentence. Please rephrase.
- 9) Line 273: typo. "...inheritance via the mother"
- 10) Line 324: typo. "...is functional in hrde-1 mutants..."
- 11) Line 362: missing commas. "...It is therefore feasible that, in the Mis-assay, loss..."
- 12) Line 367-368: remove comma. "...to better understand this is that we do..."
- 13) Line 376: typo. "initial RNAi response focused on..."
- 14) Line 414: typo. "...brought to RdRP substrates by..."

Response:

We thank the reviewer for the supportive comments. We have fixed the textual issues and added the requested information for the alleles used to the Reagents and Tools Table. In Figure 2, animals in generation 8 were used. This information has also been added.

Referee #3:

This study examines pattern of RNAi inheritance in hermaphrodite and male *C. elegans*. A genetic framework for RNAi inheritance in selfing hermaphrodites has already been derived from many studies examining silencing of endogenous loci. These studies have yielded a working model for RNAi inheritance that involve parallel cycles of sRNA amplification that independently target transcripts in the nucleus and cytoplasm (described below). The authors use a flawed

assay (which examines protein rather than RNA and consequently cannot distinguish between RNA silencing in the nucleus vs the cytoplasm) and claims to establish a new genetic hierarchy, which does not incorporate, and in some cases appear contrary to, prior findings. Their findings are also purely genetics and hence do not address molecular underpinnings. Because of the problems inherent to their assay, it is difficult to understand how their data builds a conceptual framework that supersedes prior models, which incorporated molecular models of sRNA amplification etc..

Response

*First, indeed, are data rest on genetic experiments. This is reflected in our title:
A genetic framework for RNAi inheritance in Caenorhabditis elegans*

Second, our data and assay are not flawed. We show that mRNA levels follow GFP protein levels very well. GFP intensity, as a proxy of mRNA expression, has been very successfully over the past years to arrive at valuable conclusions in the C. elegans RNAi field.

Third, genetics has very successfully been used to establish hierarchies. Our experimental set-up, involving crosses, is unique, and brings a different perspective to the inheritance field. We do not claim we have proven any molecular mechanisms. We do suggest how the process may work, and in fact we largely agree with the Ouyang et al model. See below. Finally, we do not see our model as 'superseding' anything. We see it as a refinement.

All in all, we are sorry to see this reviewer reacts in a dismissive manner to our work. We believe that different experimental set-ups are valuable to dissect this complex phenomenon of RNAi inheritance, and the field should embrace varied approaches, not dismiss them, and at the same time be aware of the pros and cons of the various approaches.

Prior model: dsRNA ingested by C. elegans hermaphrodites can initiate silencing that is transmitted to the next generation. Prior studies have shown that "transgenerational RNAi" involves at least three independent sRNA amplification cycles that use the targeted transcript as a template to generate silencing small RNAs (sRNAs). In the first amplification cycle (RDE-1 dependent), primary sRNAs (generated from the dsRNA trigger) anneal to the targeted transcript, tagging it as a template for the generation of secondary sRNAs. Secondary sRNAs cause degradation of most of the targeted transcript in the cytoplasm of the animal exposed to the dsRNA trigger (P0). The secondary sRNAs also activate in the P0 animal two additional amplification cycles. These are required for maintenance of the silencing in the P0 and propagation of the silencing to the next generation. The HRDE-1-dependent cycle is a nuclear cycle that generates tertiary sRNAs from nascent transcripts in the nucleus. The HRDE-1 cycle reduces transcription at the locus but maintains a low level of nascent transcripts for continuous production of tertiary sRNAs. The ZNFX-1-dependent cycle also generates tertiary sRNAs but using mature transcripts in the cytoplasm as templates. These tertiary sRNAs ensures the continuous degradation of cytoplasmic transcripts. Like the HRDE-1 cycle, the ZNFX-1 cycle reduces does not eliminate all transcripts, but maintains low levels of mature transcripts (in the cytoplasmic P granules) to ensure continuous production of tertiary sRNAs. By maintaining a residual pool of transcripts to use as templates for sRNA amplification, the HRDE-1 and ZNFX-1 cycles "memorialize" the silencing for transmission across generations. Loss of either the HRDE-1 or the ZNFX-1 cycle impairs but does not eliminate the RNAi response, as the two cycles can operate independently in the nucleus (HRDE-1 cycle, which reduces nascent transcript production) and in the cytoplasm (ZNFX-1 cycle, which degrade mature transcripts). Only loss of both HRDE-1 and ZNFX-1 cycles at the same time eliminates silencing propagation completely. In hrde-1;znfx-1 double mutants, only the RDE-1 cycle remains which transiently reduces cytoplasmic transcripts leading to apparent silencing in the P0 animal, but this effect is temporary and not transmitted to the next generation (Ouyang et al., 2022 and references therein).

Response:

We extensively discuss our model in relation to this model, and come to the conclusion that the two models are not conflicting.

The reviewer does not point out the following:

The P0 and F1 generations in the Ouyang et al. study have the same genotype, and hence, no statement can be made on the origin of any phenotype. Phenotypes in the F1 can, and likely will be affected by also by the fact that the P0 was already mutant. This is an aspect we address in our work and our discussion. For comments on the proposed ZNFX-1- and HRDE-1-independent aspects, we refer to our response below.

New study: The new study by Schreier et al., proposes to extend prior studies by comparing RNAi inheritance patterns in males vs hermaphrodites and examining the roles of additional genes implicated in RNAi. Unlike prior studies which directly monitored the silencing of endogenous transcripts using in situ hybridization (monitoring both nuclear and cytoplasmic transcripts), this study utilizes a transgene expressed in the germline and follows protein (GFP) expression from the transgene. Consequently, the authors cannot determine whether the silencing patterns they are observing are due to silencing of nascent transcripts at the locus or silencing of mature transcripts in the cytoplasm. Additionally, the authors do not examine sRNA production as did prior studies. These concerns were brought up by reviewers in the last round of reviews, but only partially addressed in the revision. For example, although the authors present some in situ hybridization experiments, they do not establish where the transgene is first transcribed in the germline (by visualizing nascent transcripts in nuclei).

Response:

Our smFISH experiments show that overall GFP protein and gfp mRNA correlate well. We acknowledge that we have not dissected transcription foci. The main reason was that we found it difficult to convince ourselves of the validity of any nuclear foci we detected. However, since we detect mRNA starting at the mitotic zone, this is the earliest the transgene is expressed.

We indeed use a transgene as an RNAi reporter, and Ouyang et al. used an endogenous gene as RNAi target. Both have their advantages, but also disadvantages. We write in our manuscript:

"We would like to emphasize here that any individual gene may respond slightly different to RNAi, depending on how strongly endogenous RNAi-pathways affect it. We would argue that such differences reflect relevant aspects of the RNAi mechanism, and one should not be valued above another. For instance, results from RNAi on transgenes may reflect better on RNAi's well-documented role on fighting foreign genetic material, while results on endogenous genes (which may have evolved to resist RNAi) may reflect more on the use of RNAi for controlling gene expression."

At present, the field is in no position to claim what would be the better approach to study RNAi inheritance, and hence we should not claim one approach to be superior over another.

The main potential discrepancy with our work, that this reviewer brings up, is the idea of two independent cycles, as proposed by Ouyang et al. One involving HRDE-1 and one involving ZNFX-1. Likely, we did not succeed in conveying the idea behind our model, which in many ways, fits very well with this general idea. We have therefore adjusted the discussion section. We hope it now clarifies better why the two scenarios are not as conflicting as this reviewer claims. Adjusted discussion section:

"First, we note that the exposure of P0 animals differs significantly between our studies: in our system the P0 is exposed from L1 to adulthood, while Ouyang et al. (Ouyang et al, 2022) expose only for a few hours. Another important difference in experimental setup is that we dissect parental from filial effects using crosses, while the Ouyang et al. study (Ouyang et al, 2022) does not: P0 and F1 genotypes were the same. This prevents any assignment of F1 effects to either P0 or F1. Third, in our assays we have not addressed mRNA levels in the F1. Even if GFP protein and gfp mRNA levels correlated well in the P0 generation, we cannot currently comment on effects on mRNA in the F1. Nevertheless, the model we propose (Figure 7) does not contradict the model proposed by Ouyang et al. (Ouyang et al, 2022). First, our model (Figure 7) contains a completely HRDE-1-independent, but ZNFX-1-dependent inheritance cycle, assuming that in the embryo a post-transcriptional response can be mounted, similar to what occurs in oocytes. Loss of HRDE-1 in the P0 may in fact enhance this mode of inheritance, as more

transcripts may become available for this post-transcriptional process in the oocytes of *hrde-1* mutants. Second, we state that ZNFX-1 stimulates tertiary 22G RNA biogenesis by cytoplasmic WAGOs, but not that it is essential for it. Thus, some level of ZNFX-1-independent inheritance is compatible with our model. Interestingly, Ouyang et al. (Ouyang et al, 2022) observed that in *znfx-1* mutants nuclear puncta accumulated more rapidly than in wild-type animals, suggesting that loss of ZNFX-1 enhances HRDE-1 mediated silencing dynamics. Intriguingly, we observed enhanced silencing in the F1 of *znfx-1* mutant mothers (Figure 3), possibly a consequence of the effect described by Ouyang et al. (Ouyang et al, 2022). Also, these observations are consistent with the ZNFX-1-independent silencing in the F1, as proposed by (Rieger et al, 2023). The precise molecular steps driven by ZNFX-1 remain presently unclear, but Ouyang et al.'s findings of ZNFX-1 association with pUG RNA (Ouyang et al, 2022) is fully consistent with our model."

In *hrde-1* hermaphrodites (P0), they find that silencing is defective in the distal germline but not in the proximal germline. They conclude that HRDE-1 is required for RNAi establishment, but not for "transmission" in hermaphrodites, presumably because they observe normal silencing in oocytes. This reasoning is flawed: silencing in oocytes does not necessarily imply transmission to the next generation. Rather it implies that the cytoplasmic cycle of silencing is working (oocytes are transcriptionally silent and only accumulate mature transcripts synthesized in younger germ cells). HRDE-1 is only required to silence nascent transcripts which in hermaphrodites are most prominent in the distal germline and absent in oocytes.

Response:

Our conclusion that HRDE-1 is not required for transmission does not rest on the difference between distal and proximal germline responses in *hrde-1* mutants. Based on these differences we state that "HRDE-1 is required for exogenous RNAi-mediated gene silencing in the distal germline."

Our conclusion that HRDE-1 does not act in transmission is based on the fact that *hrde-1* wild-type (heterozygous) offspring from *hrde-1* mutants can re-establish RNAi. There is nothing "flawed" about these experiments, statements or reasonings.

The authors observed that *hrde-1* mutant hermaphrodites (mated to wild-type males) "contrary to expectations" still "passed on *gfp* RNAi to both F1 hermaphrodites and males". This is expected as per the model above, as the ZNFX-1 cytoplasmic cycle is still active in these animals. Furthermore, this observation does not mean that *hrde-1* is NOT required for transmission. If the authors had looked, they would have noticed that *hrde-1* F1 animals, despite silencing cytoplasmic transcripts, fail to silence nascent transcripts in nuclei. Similarly the findings that HRDE-1 and ZNFX-1 are required for re-establishment of RNAi in the F1s (line 220) is predicted from the model above, since each cycle is required to re-amplify the response in each new generation. If the authors had assayed RNA instead of GFP, they might have noticed that each SINGLE mutant only transmits and restart a partial silencing response due to the presence of the other cycle.

Response:

Since HRDE-1 is still widely seen as the major inheritance Argonate, this was surprising to us. Furthermore, the complete independence of ZNFX-1 and HRDE-1 mechanisms, as portrayed by this reviewer is not in line with observations by Ouyang et al. (2022): they describe an effect on HRDE-1 silencing kinetics in the P0 upon loss of ZNFX-1, indicating these mechanisms are connected in some manner. Most importantly, however, whereas this reviewer repeatedly emphasizes how our work is in disagreement with previous work, we believe it is not (see above).

Possibly, the reviewer mis-understands our model and/or results, because the phrase "If the authors had looked, they would have noticed that *hrde-1* F1 animals, despite silencing cytoplasmic transcripts, fail to silence nascent transcripts in nuclei" appears strange to us. We see silencing in the F1 which is dependent on HRDE-1, and we would fully agree that nascent transcripts should fail to be reduced in *hrde-1* mutant F1 animals. We just did not see significant

silencing in absence of HRDE-1. Possibly this is due to us looking at protein and not mRNA; possibly this is due to other experimental conditions. This we include now in our discussion: "...in our assays we have not addressed mRNA levels in the F1. Even if GFP protein and gfp mRNA levels correlated well in the P0 generation, we cannot currently comment on effects on mRNA in the F1."

This paper does present new data regarding patterns of male inheritance and the role of WAGO-3 and WAGO-4. However, without a more careful integration of their findings with prior models in the literature, and direct examination and quantification of cytoplasmic and nascent transcripts, it is difficult to interpret these data. The authors make several claims that appear at odds with prior data. For example, in the abstract: "nuclear activity drives most of the silencing and cytoplasmic activity ensures inheritance". This appears to contradict the model above where nuclear (HRDE-1) and cytoplasmic (ZNF-1) cycles were shown to function independently: each are sufficient for partial silencing and transmission. Per the model above, SILENCING refers to a reduction in nascent transcript production (HRDE-1) and degradation of mature transcripts in the cytoplasm (ZNF-1) and TRANSMISSION refers to cycles of sRNA amplification using residual nuclear transcripts (HRDE-1) or residual mature transcripts in P granules (ZNF-1). The authors perhaps are using a different definition of silencing and transmission?

Response

We apologise for the inaccurate phrasing. We ourselves write about cytoplasmic silencing, notably in the oocytes. We have adjusted the sentence to:

"We propose a cycle in which nuclear and cytoplasmic Argonaute proteins interact to generate both a silencing response and a cytoplasmic factor that transmits the silencing between parent and offspring, WAGO-3. Finally, we implicate the RNA helicase ZNF-1 as a factor that allows the inherited WAGO-3 protein to trigger silencing in the offspring."

We would agree with the description of silencing by this reviewer. The term transmission we define more narrow. In fact, as this is not a generally used term in the field, we now explicitly define "transmission" in our manuscript:

"For clarity, we define transmission as the transfer of parental-derived factors that mediate gfp silencing to the offspring, e.g. small RNA-loaded Argonaute proteins. While these factors are required to initialize the re-establishment of gfp silencing in F1 animals, other parentally expressed factors are likely to act upstream of this, and zygotically expressed factors are likely to act downstream in this process without themselves being directly involved in the transmission process itself."

In summary, while the authors have attempted to respond to the prior reviews with a few in situ hybridization, their findings remain poorly integrated with the prior literature and in some cases appear to have been misinterpreted due to their inability to distinguish between silencing of nascent versus mature transcripts.

Response:

We politely, but firmly disagree with this statement. Besides our extensive reference to previous literature, our discussion section "Comparison to previous work" explicitly addresses this aspect. Nevertheless, we have adjusted this section of our discussion to try to better convey our model, and how this model is not as conflicting with previous results as this reviewer portrays.

Cross-comments from referee 1:

At first glance, I would say that maybe the authors need to re-write the ms taking into consideration the comments by ref #3. Maybe they can additionally quantify their nuclear signal from smFISH to assess nuclear silencing (no requirement for additional experiment, just re-analysis), but overall their smFISH recapitulated the results from protein expression in my opinion.

Response:

We have addressed the criticism of reviewer 3 textually, as indicated above. As we also mentioned in our response to reviewer 3, we are not confident enough that the nuclear signals we detect in fact reflect transcription foci.

Cross-comments from referee 2:

In my opinion textual edits would suffice.

Response:

We have addressed the criticism of reviewer 3 textually, as indicated above.

Prof. Rene Ketting
Institute of Molecular Biology
-
Ackermannweg 4
Mainz 55128
Germany

Dear Rene,

I am very pleased to accept your manuscript for publication in the next available issue of EMBO reports. Thank you for your contribution to our journal.

Best,
Esther
